# TOWARDS A COMPLETE LOGICAL FRAMEWORK FOR GNN EXPRESSIVENESS

**Tuo Xu**
Independent Researcher
`tourxu@live.com`

## ABSTRACT

Designing expressive Graph neural networks (GNNs) is an important topic in graph machine learning fields. Traditionally, the Weisfeiler-Lehman (WL) test has been the primary measure for evaluating GNN expressiveness. However, high-order WL tests can be obscure, making it challenging to discern the specific graph patterns captured by them. Given the connection between WL tests and first-order logic, some have explored the logical expressiveness of Message Passing Neural Networks. This paper aims to establish a comprehensive and systematic relationship between GNNs and logic. We propose a framework for identifying the equivalent logical formulas for arbitrary GNN architectures, which not only explains existing models, but also provides inspiration for future research. As case studies, we analyze multiple classes of prominent GNNs within this framework, unifying different subareas of the field. Additionally, we conduct a detailed examination of homomorphism expressivity from a logical perspective and present a general method for determining the homomorphism expressivity of arbitrary GNN models, as well as addressing several open problems.

## 1 INTRODUCTION

Graph Neural Networks (GNNs) are the dominant approaches for learning graph-structured data and have achieved remarkable success over the past few years. Among them, Message Passing Neural Networks (MPNNs) (Kipf & Welling, 2016b) are prominent GNN models that learn node and graph representations by aggregating information from neighbors. However, a noticeable drawback of GNNs lies in their limited expressive power. Xu et al. (2018); Morris et al. (2019) discovered that the separation power of MPNNs is inherently restricted by 1-dimensional Weisfeiler-Lehman (1-WL) test. Subsequently, many studies have focused on enhancing expressiveness and designing more powerful GNN models using the $k$-WL framework as a metric.

While the $k$-WL hierarchy offers a systematic measure of GNN expressiveness that increases with $k$, it remains somewhat limited. First, it lacks interpretability. Despite 1-WL being a relatively straightforward procedure which aggregates neighborhood information, it is hard to understand what $k$-WL actually learns and how it surpasses $(k-1)$-WL. Second, WL tests are arguably too coarse to evaluate the expressive power of GNN models (Zhang et al., 2024; Morris et al., 2022; Puny et al., 2023): many works (Qian et al., 2022; Frasca et al., 2022) only provide loose upper bounds of expressiveness of their proposed models in terms of $k$-WL and most efficient GNNs are only proved to be more expressive than 1-WL by constructing specific example graphs (Zhang & Li, 2021; Bevilacqua et al., 2021; Zhang et al., 2023).

Apart from the WL hierarchy, some works systematically study GNN expressivity from various perspectives. For instance, Zhang et al. (2024) identified all substructures captured by several popular GNN models. Xu & Zou (2024) examined the approximate inference capabilities of popular GNN models. These works, although provide novel insights about GNN expressivity, still lack *extendability*: they do not provide a general method for analyzing the expressiveness of *arbitrary* GNN models using their theoretical framework. Thus, considerable effort is required when considering novel GNN variants.

To address these limitations, this paper studies GNN expressivity from a logical perspective. Previous research, such as Barceló et al. (2020), investigated the logical expressivity of MPNNs, while

Huang et al. (2024) explored the logical expressivity of a specific class of GNN models for link prediction in knowledge graphs. However, these works study MPNNs and other GNN variants separately, leaving many popular models unexamined. Additionally, there still lacks a unified framework for assessing the logical expressivity of general GNN models.

**Contributions.** This paper presents a novel framework for assessing the logical expressivity of *arbitrary* GNN models, provided they can be represented through a series of combination and aggregation operations. We present a method for constructing the set of logical formulas captured by these GNN models. Using this framework, we describe the logical expressivity of popular GNN models in terms of graph-level, node-level, and link-level predictions. Furthermore, we demonstrate how several key topics in GNN expressivity, such as homomorphism expressivity, expressivity comparisons, and estimating WL upper bounds, can be effectively addressed by leveraging the logical expressivity results present in this study.

## 2 BACKGROUND

**Notations and definitions.** We use $\{\}$ to denote sets and use $\{\{\}\}$ to denote multisets. The index set is denoted as $[n] = \{1, ..., n\}$. In this paper we consider finite, directed graphs with node and edge labels. Let $G = (\mathcal{V}_G, \mathcal{E}_G)$ be a graph where $\mathcal{V}_G$ denotes the set of nodes in $G$ and $\mathcal{E}_G$ the set of edges. $\ell$ denotes the label function that maps nodes and edges to labels: $\ell(u)$ is the label of node $u$ and $\ell(u, v)$ the label of $(u, v)$ provided that edge $(u, v)$ exists. $\mathcal{N}(u)$ denotes the set of generalized neighbors of node $u$: certain GNNs might generalize the concept of neighbor in graphs in different manners. For example, in MPNNs the neighbors of a node $x$ is conventionally defined as $\mathcal{N}_{\text{MPNN}}(x) := \{y \mid E(x, y) \in \mathcal{E}\}$ where $\mathcal{E}$ is the set of edges; 2-FGNNs define the neighbor of a node pair $(x, y)$ to be $\mathcal{N}_{\text{2-FGNN}}(x, y) = \{((x, z), (z, y)) \mid z \in \mathcal{V}\}$ where $\mathcal{V}$ is the set of nodes. We use symbols $\varphi, \psi, \phi, ...$ to refer to logic formulas.

In this paper we focus on logic formulas and GNN models that operate on nodes and more generally node tuples. For instance, logic formulas $\psi(x_1, x_2, x_3)$ and GNNs that learn representations for node pairs. We define the *order* of logic formulas and GNNs to be the size of node tuples considered, e.g. the order of $\psi$ is 3 and the order of GNNs that compute representations for node pairs is 2. For notation brevity we use $\mathbf{u}, \mathbf{v}, \mathbf{x}, ... \in \mathcal{V}^k$ to refer to node tuples where $k \in \{0, 1, ...\}$ is the order of them, e.g. $\psi(x_1, x_2, x_3)$ is represented by $\psi(\mathbf{x})$ where $\mathbf{x}$ is a 3-order tuple $\mathbf{x} = (x_1, x_2, x_3)$. Specially, if $k = 1$, then $\mathbf{u} \in \mathcal{V}^1$ represents nodes in graphs; if $k = 0$, then $\mathbf{u} \in \mathcal{V}^0$ simple represents none.

**Graph isomorphism.** Two graphs $G = (\mathcal{V}_G, \mathcal{E}_G)$ and $H = (\mathcal{V}_H, \mathcal{E}_H)$ are isomorphic, denoted as $G \simeq H$, if $|\mathcal{V}_G| = |\mathcal{V}_H|$ and there exists a bijective permutation $\pi : \mathcal{V}_G \rightarrow \mathcal{V}_H$ satisfying: (1) $(u, v) \in \mathcal{E}_G \iff (\pi(u), \pi(v)) \in \mathcal{E}_H$ for $u, v \in \mathcal{V}_G$, (2) $\ell(u) = \ell(\pi(v))$ for $u \in \mathcal{V}_G$ and (3) $\ell(u, v) = \ell(\pi(u), \pi(v))$ for $(u, v) \in \mathcal{E}_G$. Such $\pi$ is an isomorphism from $G$ to $H$. Our work is also closely related to a family of necessary tests for graph isomorphism namely Weisfeiler-Lehman (WL) tests introduced in Appendix B.

**First-order Logic.** We briefly introduce first-order logic and its relation with graphs. Consider the following formula

$$\varphi(x) := \text{Red}(x) \land \exists y \, (E(x, y) \land \text{Blue}(y)) \, .$$

There are two variables $\mathbf{var}(\varphi) = \{x, y\}$ in the formulation of $\varphi$, and $\varphi$ has exactly one free variable $\mathbf{free}(\varphi) = \{x\}$ which is not bounded by any quantifiers $\exists, \forall$. $\varphi(x)$ is $\mathbf{true}$ *iff* $x$ is $\text{Red}$ and exists a $\text{Blue}$ $y$ such that $E(x, y)$ holds. It is straightforward to relate this formula with graphs: variables $x, y$ are corresponded nodes in graphs and the predicates $\text{Blue}, \text{Red}$ are corresponded to node labels while $E$ is corresponded to edges. Therefore, $\varphi(x)$ determines whether a node $x$ is $\text{Red}$ and has a $\text{Blue}$ neighbor.

In this paper we focus on a fragment of the first-order logic which allows the utilization of counting quantifiers $\exists^{\geq N}$. The semantic of the quantifier $\exists^{\geq N}$ where $N \in \{1, 2, ...\}$ is to describe "there exists no less than N variables such that". For example, consider

$$\psi(x) := \exists^{\geq 2} y \, (E(x, y) \land \text{Blue}(y)) \, .$$

$\psi(x)$ is **true** *iff* $x$ has 2 or more Blue neighbors. Such a family of logic formulas is called First-order Logic with Counting quantifiers (FOC) and possesses following property.

**Proposition 1.** *(Cai et al., 1992) For any graphs $G, H$, $k$-WL assigns the same color to $G$ and $H* iff *all FOC formulas with no more than $k$ variables classifies $G$ and $H$ the same.*

**Graph neural networks.** GNNs can be generally described as graph functions that are invariant under isomorphism. Most popular GNNs follow a *color refinement* paradigm Zhang et al. (2024) to achieve such invariance: they maintain a representation for each node (or more generally, node tuple) and iteratively updates these representations via combination and aggregation functions. Consider, for example, message passing neural networks (MPNNs) Morris et al. (2019), which maintains a representation $\chi^{(l)}(x)$ for node $x$ at layer $l$. The representations are updated using the following formula:

$$\chi^{(l+1)}(x) = \text{COM}^{(l)}\left(\chi^{(l)}(x), \text{AGG}^{(l)}\left(\left\{\left\{\chi^{(l)}(y) \mid y \in \mathcal{N}(x)\right\}\right\}\right)\right),$$

where $\text{COM}^{(l)}(\cdot, \cdot)$ represents an arbitrary function combining two representations and $\text{AGG}^{(l)}(\{\{\cdot\}\})$ represents an arbitrary permutation-invariant function that aggregates a multi-set of representations.

There are also many other popular GNNs, which are listed in Appendix C. Generally, these models maintain a representation $\chi^{(l)}(\mathbf{u})$ for node tuple $\mathbf{u}$ at layer $l$. Let $L$ be the total number of layers of a GNN model. Then the representation $\chi^{(L)}(\mathbf{u})$ for node tuple $\mathbf{u}$ at layer $L$ serves as the output of the GNN. Since this paper studies the relationship between GNNs and logic formulas, we focus on GNNs with binary outputs (i.e., **true** and **false**).

## 3 LOGICAL EXPRESSIVITY OF GRAPH NEURAL NETWORKS

### 3.1 EQUIVALENT LOGIC SETS

Given a GNN model $M$ and a logic formula $\varphi$, let $\chi(\mathbf{u})$ be the output of $M$ for node tuples $\mathbf{u} \in \mathcal{V}^k$. We say $M$ captures $\varphi$ if the results of $\varphi$ are reproduced by $M$, Concretely, $M$ captures $\varphi$ if the orders of $M$ and $\varphi$ are equal, and $\varphi(\mathbf{u}) = \chi(\mathbf{u})$ holds for arbitrary graph $G$ and $\mathbf{u} \in \mathcal{V}^k$. In this paper, we attempt to answer the question: *what logic formulas can GNN models capture?* This leads to the following definition of logical expressivity.

**Definition 2.** Given a family of functions $\mathcal{X}$ (e.g. a class of GNNs) where each function $\chi \in \mathcal{X}$ maps $k$-order node tuples to $\{$**true**, **false**$\}$, the equivalent logic set of $\mathcal{X}$ is a subset $\Phi$ of first order logic formulas satisfying:

- The order of each $\varphi \in \Phi$ is $k$;

- For all $\varphi \in \Phi$, there exists $\chi \in \mathcal{X}$ such that for arbitrary graphs $G$ and $\mathbf{u} \in \mathcal{V}_G^k$, $\varphi(\mathbf{u}) = $ **true** *iff* $\chi(\mathbf{u}) = $ **true**, and we say $\chi$ captures $\varphi$.

- A FOC formula is captured by $\mathcal{X}$ *iff* it is in $\Phi$.

- Given arbitrary positive integer $N$ and $\chi \in \mathcal{X}$, there exists $\varphi \in \Phi$ satisfying: for any graphs $G$ with no more than $N$ nodes and $\mathbf{u} \in \mathcal{V}_G^k$, $\varphi(\mathbf{u}) = $ **true** *iff* $\chi(\mathbf{u}) = $ **true**.

- Given any graphs $G, H$ and $\mathbf{u} \in \mathcal{V}_G^k, \mathbf{v} \in \mathcal{V}_H^k$, all $\chi \in \mathcal{X}$ cannot distinguish $\mathbf{u}, \mathbf{v}$ *iff* all logic formulas $\varphi \in \Phi$ classify $\mathbf{u}, \mathbf{v}$ the same.

In our definition, we emphasize the equivalence between logic formulas and GNNs by stating that the discriminating power of all $\varphi \in \Phi$ and all $\chi \in \mathcal{X}$ are equivalent, which follows the setting in Cai et al. (1989). Also, we attempt to study the one-to-one correspondence between each GNN model $\chi \in \mathcal{X}$ and each $\varphi \in \Phi$. The ideal statement would be: given arbitrary $\chi \in \mathcal{X}$, there exists $\varphi \in \Phi$ such that $\varphi(\mathbf{u}) = \chi(\mathbf{u})$ for all possible $\mathbf{u}$. However, there does exist GNN models which is not captured by any logic formulas, and thus we relax the statement to be the forth statement in Definition 2.

The equivalent logic sets therefore sufficiently describe the logical expressiveness of GNN models. Moreover, similar to the homomorphism expressivity (Zhang et al., 2024), the metric of equivalent logic sets is also *quantitative*, as we can identify distinct logic sets for different GNN models that precisely describe their expressiveness , making it finer than metrics based on graph isomorphism tests which only provide qualitative results. Moreover, the equivalent logic sets can also be used to compare the expressivity of different GNNs: a class of GNN models $M_1$ is more expressive than $M_2$ *iff* $\Phi_2 \subset \Phi_1$ where $\Phi_1, \Phi_2$ are the equivalent logic set of $M_1$ and $M_2$ respectively. Above all, the significance of logical expressivity lies in its interpretability: we can not only describe what patterns GNN models can capture, but also understand the expressivity gap of different models by studying the difference of the corresponding equivalent logic sets.

## 3.2 DESCRIBING LOGICAL EXPRESSIVITY FOR GNNS

It is evident that the equivalent logic set of GNNs can be *infinite*. We utilize a recursive construction procedure to describe such sets, similar to previous works. Consider the set of graded model logic $\Phi$ (de Rijke, 1996). $\Phi$ is defined by specifying how its elements are recursively constructed: to begin with, let $\mathrm{Col}(x) \in \Phi$ where $\mathrm{Col}$ represents node colors. Each element of $\Phi$ is either $\mathrm{Col}$, or one of the following:

$$\neg\varphi(x), \quad \varphi(x) \wedge \varphi'(x), \quad \exists^{\geq N}\left(E(x,y) \wedge \varphi(y)\right),$$

where $\varphi, \varphi' \in \Phi$ and $N$ is a positive integer. Therefore, we define $\Phi$ by specifying how its elements are constructed, starting from the input node colors $\mathrm{Col}$. For notation brevity we can abbreviate the definition into one line:

$$\varphi(x) := \exists^{\geq N}\left(\varphi'(x) \wedge E(x,y)\right) \mid \neg\varphi'(x) \mid \varphi'(x) \wedge \varphi''(x) \mid \mathrm{Col}(x), \tag{1}$$

with the convention that logic formulas $\varphi$ together with its superscript variants $\varphi', \varphi''$ belong to the same logic set $\Phi$. In Section 6.1 we discovered that Eq. 1 describes the equivalent logic set of MPNNs with undirected, homogeneous input graphs.

## 4 GENERAL AGGREGATE-COMBINE NETWORKS

To formally discuss the logical expressivity of arbitrary GNN models, we first summarize GNN models including Message Passing Neural Networks (Xu et al., 2018), Higher-order GNNs (Morris et al., 2018), Subgraph GNNs (Bevilacqua et al., 2021), via a unified design paradigm namely General Aggregate-Combine Neural Networks (GACNNs). The basic idea is straightforward: we decompose the structure of different GNN layers into the same, principled *aggregation* and *combination* function series, which further enable us to study the expressive power of different GNN models via a unified framework.

Formally, let $\chi^{(l)}(\mathbf{u})$ be the representation of a $k$-order node tuple $\mathbf{u} \in \mathcal{V}^k$ computed by the $l$-th GACNN layer. The $(l+1)$-th GACNN layer takes $\chi^{(l)}(\mathbf{u})$ as input and evaluates $\chi^{(l+1)}(\mathbf{u})$ for $\mathbf{u} \in \mathcal{V}^k$. $\chi^{(l+1)}(\mathbf{u})$ is computed by $\chi^{(l)}(\mathbf{u})$ via a sequence of two operations: *combination* (denoted by COM) and *aggregation* (denoted by AGG) . $\mathrm{COM}(\cdot,\cdot)$ represents an arbitrary function combining two representations, and $\mathrm{AGG}(\{\!\{\cdot\}\!\})$ represents a permutation-invariant function that aggregates a multi-set of representations. The evaluation of a GACNN layer is decomposed into a series of intermediate variables $\{\chi_1^{(l)}, ..., \chi_K^{(l)}\}$. Denoting $\chi_0^{(l)} := \chi^{(l)}$ and $\chi_{K+1}^{(l)} = \chi^{(l+1)}$, we define either

$$\chi_i^{(l)}(\mathbf{u}) = \mathrm{COM}_i^{(l)}\left(\chi_j^{(l)}(\mathbf{u}), \chi_k^{(l)}(\mathbf{u})\right), \quad \text{or} \quad \chi_i^{(l)}(\mathbf{u}) = \mathrm{AGG}_i^{(l)}\left(\{\!\{\chi_j^{(l)}(\mathbf{v}) \mid \mathbf{v} \in \mathcal{N}(\mathbf{u})\}\!\}\right), \tag{2}$$

where $1 \leq j, k < i \leq K+1$, $\mathrm{COM}_i^{(l)}$ is a combination function, $\mathrm{AGG}_i^{(l)}$ is an aggregation function and $\mathcal{N}(\mathbf{u})$ is the generalized neighbor of $\mathbf{u}$ defined by the GNN model. Specially, we denote by $\chi^{(0)} = \mathrm{INIT}(\mathbf{u})$ the initial representation of $\mathbf{u}$. The above definition generally expresses the aggregation and combination steps of GNN layers.

**Example.** To better introduce the idea of GACNNs we illustrate how MPNNs are described by the above GACNNs construction steps. Consider MPNNs Xu et al. (2018) whose layers are defined by

$$\chi^{(l+1)}(x) = \mathrm{COM}^{(l)}\left(\chi^{(l)}(x), \mathrm{AGG}^{(l)}\left(\{\!\{\chi^{(l)}(y) \mid y \in N(x)\}\!\}\right)\right),$$

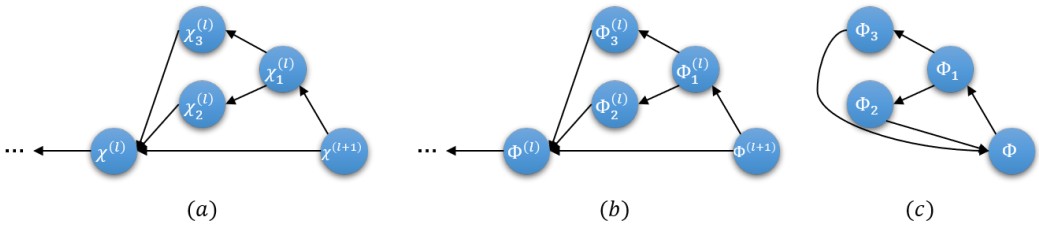

Figure 1: Left: the structure of the example GACNN. Middle: the structure of the equivalent logic sets, each corresponding to one of the nodes in the left GACNN. Right: the structure of the equivalent logic sets of the left GACNN, regardless of the number of layers $l$.

where $x, y$ denotes nodes in graphs. We can simply decompose one layer of MPNN into

$$\chi^{(l+1)}(x) = \text{COM}^{(l)}\left(\chi^{(l)}(x), \chi_1^{(l)}(x)\right),$$

$$\chi_1^{(l)}(x) = \text{AGG}^{(l)}\left(\left\{\left\{\chi^{(l)}(y) \mid v \in N(x)\right\}\right\}\right).$$

In this manner we describe the MPNN layers using the principled GACNN framework. For more examples, appendix G.6 illustrates how we build popular GNN variants using the GACNN framework.

**Structure of GACNNs.** The advantage of decomposing GNN layers into a series of the principled aggregation and combination procedures is that we can unify the study of complicated GNN models into the study of AGG and COM modules. Consider, for example, the Local 2-GNN model. The evaluation of this model can be illustrated as Figure 1 (a), where the node $\chi^{(l+1)}$ represents representations of all nodes at layer $l+1$ and is evaluated directly by $\chi_1^{(l)}$ and $\chi_2^{(l)}$, which are again evaluated by their children along the hierarchy until $\chi^{(l)}$. It is evident that by explicitly expanding the evaluation procedure of $\chi^{(l+1)}$, the whole and complicated GNN computation structure is broken down into small and simple pieces containing only two types of computation: let $\chi_p$ be a (parent) node, then if $\chi_p$ only has one child $\chi_l$, $\chi_p$ is evaluated in the form of $\chi_p = \text{AGG}_p\left(\{\{\chi_l\}\}\right)$; otherwise $\chi_p$ has two children $\chi_l, \chi_r$ and is evaluated by $\chi_p = \text{COM}_p\left(\chi_l, \chi_r\right)$. In this manner we break down the computation procedure of GNNs into principled aggregation and combination procedure, each corresponding to a parent-children pair in the computation graph. We next investigate the logical expressivity of GACNNs by studying the *local property* of such parent-children pairs.

## 5 ON THE EQUIVALENT LOGIC FRAGMENT OF GRAPH NEURAL NETWORKS

In this section we discuss the separation power and function approximation property of general graph neural networks by providing the equivalent logic set of arbitrary GACNNs.

### 5.1 EQUIVALENT LOGIC SETS FOR GENERAL COMPUTATION PROCEDURE

For now let us relax the utilization of GACNN models and focus purely on the two types of computation units AGG and COM proposed in Section 4. Concretely, suppose a set $\{\chi_1, \chi_2, ..., \chi_K\}$ each $\chi_i$ for $i \in [K]$ is defined by either $\chi_i(\mathbf{u}) = \text{AGG}_i(\{\{\chi_j(\mathbf{v}) \mid \mathbf{v} \in \mathcal{N}_i(\mathbf{u})\}\})$, $\chi_i(\mathbf{u}) = \text{COM}_i(\chi_j(\mathbf{u}), \chi_k(\mathbf{u}))$, or $\chi_i(\mathbf{u}) = \text{INIT}_i(\mathbf{u})$ where $i > j, k$. Moreover, $\chi_K$ maps a node tuple to binary outputs $\{\textbf{true}, \textbf{false}\}$. We next show that it is possible to find the equivalent logic sets of $\chi_K$.

**Theorem 3.** *Let $\mathcal{X}_i$ be the set of all possible $\chi_i$ defined above for $i \in [K]$. There exists $\{\Phi_1, ..., \Phi_K\}$ defined below, such that $\Phi_K$ is the equivalent logic set of $\chi_K$.*

- $\chi_i(\mathbf{u}) = \text{AGG}_i(\{\{\chi_j(\mathbf{v}) \mid \mathbf{v} \in \mathcal{N}_i(\mathbf{u})\}\})$
  $\iff \varphi_i(\mathbf{u}) := \exists^{\geq N}\mathbf{v}\left(\varphi_j(\mathbf{v}) \wedge \mathbf{1}_{\mathbf{v} \in \mathcal{N}_i(\mathbf{u})}\right) \mid \neg\varphi_i'(\mathbf{u}) \mid \varphi_i'(\mathbf{u}) \wedge \varphi_i''(\mathbf{u}),$

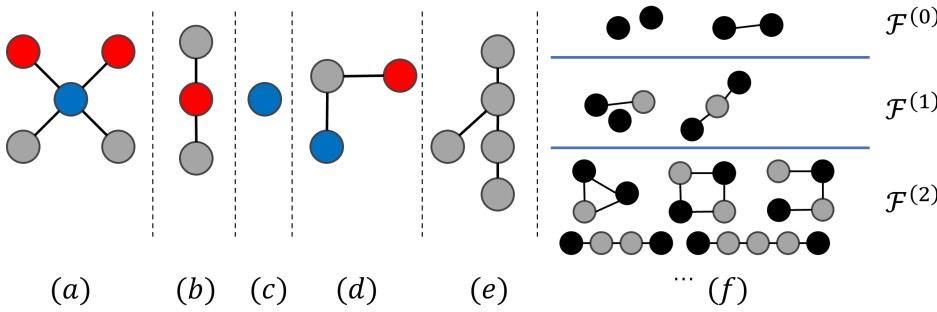

Figure 2: (a) - (d): An illustration of logic formulas and GNN behaviours. (e) - (f): An illustration of the subgraph used for construct the homogeneous expressivity, (e) corresponds to the construction of $\varphi := \exists xyzwu\,((E(x,y) \wedge E(y,z) \wedge E(y,w) \wedge E(w,u))$. (f) corresponds to 2-FGNNs.

- $\chi_i(\mathbf{u}) = \mathrm{COM}_i\,(\chi_j(\mathbf{u}), \chi_k(\mathbf{u}))$
$$\Longleftrightarrow \varphi_i(\mathbf{u}) := \varphi_j(\mathbf{u}) \mid \varphi_k(\mathbf{u}) \mid \neg\varphi_i'(\mathbf{u}) \mid \varphi_i'(\mathbf{u}) \wedge \varphi_i''(\mathbf{u}),$$

- $\chi_i(\mathbf{u}) = \mathrm{INIT}_i\,(\mathbf{u}) \iff \varphi_i(\mathbf{u}) := \mathbf{atp}(\mathbf{u}) \mid \neg\varphi_i'(\mathbf{u}) \mid \varphi_i'(\mathbf{u}) \wedge \varphi_i''(\mathbf{u}),$

*where $\varphi_i', \varphi_i'' \in \Phi_i, \varphi_j \in \Phi_j, \varphi_k \in \Phi_k$ for $i,j,k \in [K]$.*

Note that by denoting $\varphi_i(\mathbf{u}) := \mathbf{atp}(\mathbf{u})$, we mean that $\varphi_i \in \Phi_i$ is capable of capturing all structures of the subgraph induced by $\mathbf{u}$. Concretely, for each possible color Col of $\mathbf{atp}$, there exists a $\varphi^{\mathrm{Col}}(\mathbf{u})$ that is **true** if and only the color of $\mathbf{u}$ assigned by $\mathbf{atp}$ is Col. $\mathbf{1}$ is the indicator: $\mathbf{1}_{\mathrm{condition}}$ is **true** *iff* the condition is satisfied.

The results in Theorem 3 specifies the construction of equivalent logic sets for arbitrary computation procedure built upon aggregation and combination functions, which is not only confined to graphs and GACNN models. As the central finding of this paper, it enables the study of complicated models built upon multiple heterogeneous layers and graphs containing node and edge labels. Consider Figure 1 (b) for example: the equivalent logic set of each node in the computation graph in Figure 1 (a) only depends on its local neighbors. If the equivalent logic set $\Phi^{(l)}$ of $\chi^{(l)}$ is known, all logic sets in Figure 1 (b) are specified by Theorem 3. This indicates that once the input representations can be described by logic formulas, we are able to determine the logical expressivity of arbitrary functions over the input representations built upon aggregation and combination functions.

*Remark.* The logic formulas in Theorem 3 are described using operators $\exists^{\geq N}, \neg$ and $\wedge$, which follows the settings in previous works (Barceló et al., 2020). Each operator is corresponded to a specific property of GNNs. First, the major characteristic of GNNs as neural networks is that GNNs make predictions based on not only the current instance (e.g. a node), but also its relation with others (which is expressed by edges in graphs). This is realized via the aggregation of information among nodes. The logic operation $\exists^{\geq N}$, which expresses "exists at least $N$", exactly corresponds to this property: consider Figure 2 (a) for example. Suppose we want to color a node as blue if it has two red neighbors. GNNs achieve the target by directly explores the neighbors of the current node $x$ and decides whether it is Blue by aggregating the colors of its neighbors. Once a GNN found that there are 2 red nodes in its neighborhood, it decides the current node should be colored blue. Logic formulas achieve the same target by noticing that for a node $x$, if we can find two nodes $y$ that satisfies $\mathrm{Red}(y) \wedge E(x,y)$, the color of $x$ should be blue. This process can be expressed by $\mathrm{Blue}(x) := \exists^{\geq 2} y(\mathrm{Red}(y) \wedge E(x,y))$. Therefore, we can understand the aggregation procedures in GNNs via the logical operation $\exists^{\geq N}$. More generally, consider Figure 2 (b), and suppose we want to decide two nodes $(x,y)$ are linked if they are connected via a red node. This can also be done by aggregating the neighbors of $x$ and $y$: once there is a common neighbor of $x$ and $y$ which is colored red, $x$ and $y$ are linked. This can be expressed by the logic formula $\mathrm{Link}(x,y) := \exists z(E(x,z) \wedge \mathrm{Red}(z) \wedge E(z,y))$. In real world, this is useful for link prediction tasks: for example predicting whether a person $x$ is another person's $y$ grandparent in a kinship graph, which is realized by checking whether there exists $z$ who is $x$'s child and $y$'s parent. The procedure can be similarly expressed by $\mathrm{Grandparent}(x,y) := \exists z(\mathrm{Parent}(x,z) \wedge \mathrm{Parent}(z,y))$.

Second, the operators $\neg$ and $\wedge$ are used to recombine existing logic formulas to obtain more complex ones. This implies the ability of GNNs to recombine existing information and make complex predictions. This ability is crucial for practical tasks: for example in molecule classification tasks it is necessary to recombine the raw input features of each node to predict molecular properties. Consider Figure 2 (c), and suppose we want to color a node blue if it is neither red or gray. Obviously this can be done by a GNN layer without aggregating the neighborhood information. A logic formula $\mathrm{Blue}(x) := \neg \mathrm{Red}(x) \wedge \neg \mathrm{Gray}(x)$ exactly expresses this. Consider a more complex example Figure 2 (d), and suppose we want to color a node gray if it has a red neighbor and a blue neighbor. This can be realized by $\mathrm{Gray}(x) := \varphi_1(x) \wedge \varphi_2(x))$ where $\varphi_1(x) := \exists y(\mathrm{Blue}(y)), \varphi_2(x) := \exists z(\mathrm{Red}(z))$. In this manner, we recombine the information $\varphi_1, \varphi_2$ obtained by aggregating the neighbors of $x$.

## 5.2 EQUIVALENT LOGIC SETS FOR GACNNS

In this section we formally describe our results for GACNNs. We assume a $L$-layer GACNN layers defined in the form of Eq. 2. For layer $l \in [L]$, let $\chi^{(l)}$ be the output representation at layer $l$ and let $\left\{ \chi_1^{(l)}, ..., \chi_K^{(l)} \right\}$ be the set of intermediate representations when computing $\chi^{(l+1)}$ from $\chi^{(l)}$. Similarly, we denote $\Phi^{(l)}$ to be the equivalent logic set of $\chi^{(l)}$ and $\Phi_i^{(l)}$ the equivalent logic set of $\chi_i^{(l)}$ for $i \in [K]$. Obviously $\Phi^{(l)}$ and $\Phi_i^{(l)}$ for $i \in [K]$ and $l \in [L]$ are directly defined by Theorem 3, which directly leads to the following result:

**Corollary 4.** *The equivalent logic set of $L$-layer GACNNs defined above is given by $\Phi^{(L)}$.*

Corollary 4 requires to specify the number of GACNN layers $L$. To derive a general result for all GACNNs regardless of the number of layers, we propose the following proposition.

**Proposition 5.** *Denote $\chi_0^{(l)} = \chi^{(l)}, \chi_{K+1}^{(l)} = \chi^{(l+1)}$ and $\Phi_{K+1} = \Phi_0 = \Phi$, Let $\Phi^{(l)}, \{\Phi_i^{(L)}\}_{i \in [K]}$ be the equivalent logic sets defined above. Then, $\Phi = \bigcup_{L=0}^{\infty} \Phi^{(L)}$ and $\Phi_i = \bigcup_{L=0}^{\infty} \Phi_i^{(L)}$ for $i \in [K]$ are defined by*

- $$\chi_i^{(l)}(\mathbf{u}) = \mathrm{AGG}_i^{(l)} \left( \left\{ \left\{ \chi_j^{(l)}(\mathbf{x}) \mid \mathbf{v} \in \mathcal{N}_i(\mathbf{u}) \right\} \right\} \right) \text{ for } l \in [0, \infty)$$
  $$\iff \varphi_i(\mathbf{u}) := \exists^{\geq N} \mathbf{v} \left( \varphi_j(\mathbf{v}) \wedge \mathbf{1}_{\mathbf{v} \in \mathcal{N}_i(\mathbf{u})} \right) \mid \neg \varphi_i'(\mathbf{u}) \mid \varphi_i'(\mathbf{u}) \wedge \varphi_i''(\mathbf{u}) \mid \mathbf{atp}(\mathbf{u}),$$

- $$\chi_i^{(l)}(\mathbf{u}) = \mathrm{COM}_i^{(l)} \left( \chi_j^{(l)}(\mathbf{u}), \chi_k^{(l)}(\mathbf{u}) \right) \text{ for } l \in [0, \infty)$$
  $$\iff \varphi_i(\mathbf{u}) := \varphi_j(\mathbf{u}) \mid \varphi_k(\mathbf{u}) \mid \neg \varphi_i'(\mathbf{u}) \mid \varphi_i'(\mathbf{u}) \wedge \varphi_i''(\mathbf{u}) \mid \mathbf{atp}(\mathbf{u}),$$

*where $\varphi_i, \varphi_i', \varphi_i'' \in \Phi_i, \varphi_j \in \Phi_j, \varphi_k \in \Phi_k$ for $i \in \{0\} \cup [K+1]$.*

The difference between Proposition 5 and Theorem 3 is that we now allow the construction of arbitrary number of GACNN layers. As long as GACNN layers share identical structure (which holds for most GNN models), we can fully describe their logical expressivity by $\Phi$ regardless of the number of layers. The structure of logic sets in Proposition 5 is illustrated in Figure 1 (c), where each set is irrelevant to the number of layers $l$ and is constructed recursively. Given any class of GNN models, so long as we can break down a layer into a series of aggregation and combination operations, we can formally define its logical expressivity using Proposition 5.

**About graph-level readout.** Generally, GNNs compute graph representations by aggregating node tuple representations, i.e. $\chi_G = \mathrm{AGG} \left( \left\{ \left\{ \chi^{(L)}(\mathbf{u}) \mid \mathbf{u} \in \mathcal{V}^k \right\} \right\} \right)$ where $L$ is the output layer. We determine the equivalent logic set of $\chi_G$ below.

**Proposition 6.** *The equivalent logic set $\Phi_G$ of the graph representation $\chi_G$ defined above is given by*

$$\varphi_G := \exists^{\geq N} \left( \varphi(\mathbf{u}) \right) \mid \neg \omega_G' \mid \omega_G' \wedge \omega_G'',$$

*where $\varphi_G, \varphi_G', \varphi_G'' \in \Phi_G$, and $\varphi \in \Phi$ is the equivalent logic set of $\chi^{(L)}$.*

The results in this section provide a general method for determining the equivalent logic set of arbitrary GACNNs. In the remaining of this paper, we utilize these results to discuss several important topics implied by the logical expressiveness of GNNs.

## 6 IMPLICATIONS

With the complete description of logical expressivity for general GNN models in previous section, we now discuss how these results provide novel insights into understanding modern GNN frameworks. In this section, we highlight the significance of our theory by introducing important results induced by our theory.

### 6.1 REGARDING EXISTING GNN MODELS

First of all, we apply our results to formally describe the logical expressivity of popular GNNs including models for graph-level or node-level prediction MPNNs (Xu et al., 2018), Subgraph GNNs (Bevilacqua et al., 2021; Qian et al., 2022), Local GNNs (Zhang et al., 2024), Folklore-type GNNs (Zhang et al., 2024) and models for link prediction NBFNet (Zhu et al., 2021), SEAL (Zhang et al., 2020), etc. The details of these models are in Appendix C. For brevity we make the convention that the equivalent logic set of each class of GNNs is represented by $\Phi$ and denote by $\varphi, \varphi', ... \in \Phi$ the elements of $\Phi$. To properly express $\Phi$ it is sometimes convenient to also define an auxiliary logic set which helps with the explanation of $\Phi$. We denote $\psi, \psi', ... \in \Psi$ as such auxiliary logic sets. The result is summarized in Proposition 7. For notation brevity, since the terms in the form of $\varphi := \neg\varphi' \mid \varphi' \wedge \varphi'' \mid \textbf{atp}$ emerges in the definition of all logic formulas, they are omitted in the following description.

**Proposition 7.** *The equivalent logic sets of GNN models can be separately defined as:*

- **MPNN:** $\varphi(x) := \exists^{\geq N} y \left( \varphi'(y) \wedge E(y, x) \right)$, *where $E$ is the edge predicate.*

- **Subgraph GNN (weak):**
  $\varphi(x) := \exists^{\geq N} y \left( \psi(x, y) \right), \psi(x, y) := \exists^{\geq N} z \left( \psi'(x, z) \wedge E(z, y) \right)$.

- **Subgraph GNN (strong):**
  $\varphi(x) := \exists^{\geq N} y \left( \varphi'(y) \wedge \psi(y, x) \right), \psi(x, y) := \exists^{\geq N} z \left( \psi(x, z) \wedge E(z, y) \right) \mid \varphi(y)$.

- **NBFNet:** $\varphi(x, y) := \exists^{\geq N} z \left( \varphi'(x, z) \wedge E(z, y) \right)$.

- **Local 2-GNN:** $\varphi(x, y) := \exists^{\geq N} z \left( \varphi'(x, z) \wedge E(z, y) \right) \mid \exists^{\geq N} z \left( E(x, z) \wedge \varphi'(z, y) \right)$

- **2-FGNN:** $\varphi(x, y) := \exists^{\geq N} z \left( \varphi'(x, z) \wedge \varphi''(z, y) \right)$.

- **SEAL (MPNN):**
  $\varphi(x, y) := \exists^{\geq N} z \left( \psi(x, z, y) \right), \psi(x, z, y) := \exists^{\geq N} w \left( \psi(x, w, y) \wedge E(w, z) \right)$.

- **2-GNN:** $\varphi(x, y) := \exists^{\geq N} z \left( \varphi'(x, z) \right) \mid \exists^{\geq N} z \left( \varphi'(z, y) \right)$.

Proposition 7 gives a unified description of the logical expressivity of popular GNN models. The result of MPNNs follows Barceló et al. (2020). Following up, it is obvious that Subgraph GNNs (weak) surpasses MPNN by modeling more complex *relations* between nodes: rather than simply the edges $E$, they deploy the more general logic formulas $\psi$ for modeling relations between nodes, which is obviously more powerful. Continuing, Subgraph GNN (strong) further strengthen $\psi$ by not only allowing the single-source update pattern $\psi(x, y) := \exists^{\geq N} z \left( \psi'(x, z) \wedge E(z, y) \right)$, but also aggregating information across different sources $\psi(x, y) := \varphi(y)$ (since $\varphi(y)$ aggregates $\psi(z, y)$ with different sources $z$). The rest of GNNs compute node-pair representations. Starting from NBFNet, it models the relation between two nodes $\varphi(x, y)$ by checking intermediate nodes $z$ and its relation w.r.t. the two end nodes $\varphi'(x, z), E(z, y)$. This is useful for link prediction tasks, e.g. to predict whether two nodes are connected $\text{Connect}(x, y) := \exists z \left( \text{Connect}(x, y) \wedge E(y, z) \right) \mid E(x, y)$, which predicts the unknown $\text{Connect}$ relation by utilizing the known edges $E$. This pattern is similar with the Bellman-Ford algorithm Baras & Theodorakopoulos (2022), which is a single-source shortest path algorithm. The logic formulas $\varphi(x, y)$ corresponding to NBFNet are also constructed with the single source node $x$. Local 2-GNN extends NBFNet by considering two sources $x, y$ separately, which allows the construction of more complex logic formulas. 2-FGNNs further generalize by defining $\varphi(x, y)$ in a multi-source manner, analogous to Floyd shortest path algorithm. SEAL also defines $\varphi(x, y)$ in a multi-source manner, but it instead constructs $\psi(x, z, y)$ and uses the intermediate nodes $z$ to perceive the relation between $x, y$ simultaneously. 2-GNNs, although compute

node-pair representations, are not suitable for link prediction since it fails to even express the simple logic rule $\text{GrandParent}(x, y) := \exists z \, (\text{Parent}(x, z) \wedge \text{Parent}(z, y))$.

## 6.2 STRUCTURAL AWARENESS OF GNNS

There has been several works that study what graph structures different GNNs are aware of, such as cycles, cliques, etc. These concepts can be unified with logic formulas. For example, determining whether a node is in a 3-clique can be written as

$$\varphi_{\text{3-clique}}(x) := \exists y, z \, (E(x, y) \wedge E(y, z) \wedge E(z, x)) \, .$$

Therefore, whether GNN models can capture 3-clique patterns depends on whether it captures $\varphi_{\text{3-clique}}$. However, determining 3-clique is a trivial task, and in practice it is often necessary to study whether GNNs can capture more complex structural patterns. We consider the concept of homogeneous expressivity proposed by Dell et al. (2018); Zhang et al. (2024).

**Homomorphism expressivity.** Homomorphism expressivity is a theory developed to precisely describe the structures of graphs being captured by GNNs. Concretely, let $G = (\mathcal{V}_G, \mathcal{E}_G), H = (\mathcal{V}_H, \mathcal{E}_H)$ be two graphs. A homomorphism from $F$ to $G$ is a mapping $\pi : \mathcal{V}_G \to \mathcal{V}_F$ that preserves labels (if any) and edges, i.e. $(\pi(u), \pi(v)) \in \mathcal{E}_H$ for all $(u, v) \in \mathcal{E}_G$ and $\ell(u) = \ell(\pi(u))$ for all nodes $u, \ell(u, v) = \ell(\pi(u), \pi(v))$ if there are node labels or edge labels respectively. $\textbf{Hom}(F, G)$ is defined to be the number of homomorphisms from $F$ to $G$. The crux is, to find all subgraphs $F$ for GNNs such that, for all pairs of graphs $G, H$, GNNs distinguish $G, H \iff \textbf{Hom}(F, G) \neq \textbf{Hom}(F, H)$. Such a set of subgraphs is referred as the *homomorphism expressivity* of GNNs. Dell et al. (2018) gives the homomorphism expressivity for 1-WL (MPNNs), while Zhang et al. (2024) extends the results to several popular GNN models.

Similar as previous discussions, in this section we aim at providing a general method to determine the homomorphism expressivity of GACNNs, based on our findings about equivalent logic sets. Suppose we are given a class of GACNNs whose equivalent logic set is $\Phi$. To simplify the discussion, we first assume no node / edge labels. We assume that the concept of neighbors in GACNNs is described by composition of edges: for example in MPNNs the neighbors of a node $x$ is defined by $\mathbf{1}_{y \in \mathcal{N}(x)} := E(x, y)$ where $E$ is the edge predicate; similarly in NBFNet $\mathbf{1}_{(x,z) \in \mathcal{N}_1(x,y)} := E(y, z)$. The homomorphism expressivity $\mathcal{F}$ can be constructed from the logic formulas in $\Phi$ via the following procedure.

1. Remove all formulas in $\Phi$ that contains negation $\neg$ or $\exists^{\geq n}$ where $n \geq 2$;

2. For each formula $\varphi \in \Phi$, add a graph $F$ into $\mathcal{F}$ which is defined below:

    (a) There exists a bijective mapping $\tau$ from $\textbf{var}(\varphi)$ to $\mathcal{V}_F$, i.e. from the variables in $\varphi$ (we avoid the reuse of variables)[1] to the nodes in $F$.

    (b) For any variables $x, y \in \textbf{var}(\varphi)$, $E(x, y)$ is a term in $\varphi$ iff $E(\tau(x), \tau(y))$ is an edge of $F$.

A discussion about the reuse of variables and why we avoid this technique is in Appendix E. We now explain the procedure. Consider for example constructing a subgraph $F$ for the logic formula $\varphi := \exists xyzwu \, ((E(x, y) \wedge E(y, z) \wedge E(y, w) \wedge E(w, u))$. The construction of $F$ is illustrated in Figure 2 (e), where $F$ possesses a node corresponding to each variable $x, y, z, w, u$ in $\varphi$ and contains edges $E(x, y), E(y, z), E(y, w)$ and $E(w, u)$. We have the following result:

**Theorem 8.** *Given a class of GACNN models and suppose $\Phi$ be the equivalent logic set. Let $\mathcal{F}$ be the homomorphism expressivity constructed by $\Phi$ as discussed above. For all pairs of graphs $G, H$. the following statements are equivalent:*

1. $\textbf{Hom}(F, G) = \textbf{Hom}(F, H)$ *for all* $F \in \mathcal{F}$.

2. *All GACNNs do not distinguish $G$ and $H$.*

---

[1] E.g. $\varphi(x) := \exists y(E(x, y) \wedge \exists x(E(x, y)))$. The variable $x$ is reused. This is a technique often used in the context of logic to reduce the number of used symbols. In the construction of homogeneous expressivity we avoid this technique and write $\varphi(x)$ as $\varphi(x) := \exists y(E(x, y) \wedge \exists z(E(z, y)))$ so that all variables are explicitly expressed. As a result, there are 3 variables $x, y, z$ in total.

Theorem 8 validates the effectiveness of our construction procedure. Together, we provide a general method to identify the homomorphism expressivity for arbitrary GACNNs, which extends the known results in previous works (Dell et al., 2018; Zhang et al., 2024). Meanwhile, we have solved a conjecture in Zhang et al. (2024), i.e. when a GNN can be described by a GACNN, its homomorphism expressivity exists and is given by Theorem 8.

**Example.** We illustrate the strategy of recursively constructing homogeneous expressivity $\mathcal{F}$ by investigating 2-FGNNs, whose equivalent logic set $\Phi$ (removed negation and $\exists^{\geq N}$ for $N \geq 2$) is given by $\varphi(x,y) := \exists z \left( \varphi'(x,z) \wedge \varphi''(z,y) \right) \mid \varphi'(x,y) \wedge \varphi''(x,y) \mid \mathbf{atp}(x,y)$. Let $\Phi^{(l)}$ be the equivalent logic set of $l$-layer 2-FGNNs. Let $\mathcal{F}^{(l)}$ be the homogeneous expressivity constructed at iteration $l$. For $\mathcal{F}^{(0)}$ at beginning, there are only two graphs in $\mathcal{F}^{(0)}$ corresponding to $\varphi^{(0)} \in \Phi^{(0)}$ where $\varphi^{(0)}(x,y) := \mathbf{atp}(x,y)$, as illustrated in top of Figure 2 (f). At the next iteration, we consider the more complex $\Phi^{(1)}$, which is given by

$$\varphi^{(1)}(x,y) := \exists z \left( \varphi_1^{(0)}(x,z) \wedge \varphi_2^{(0)}(z,y) \right) \mid \varphi_1^{(0)}(x,y) \wedge \varphi_2^{(0)}(x,y) \mid \mathbf{atp}(x,y).$$

We can simply construct $\mathcal{F}^{(1)}$ by reusing the known results about $\mathcal{F}^{(0)}$. Specially, to construct $\varphi^{(1)}(x,y) := \exists z \left( \varphi_1^{(0)}(x,z) \wedge \varphi_2^{(0)}(z,y) \right)$, we start from an empty graph $F$ and add three nodes $v_x, v_y, v_z$ corresponding to variables $x, y, z$ in $\varphi^{(1)}$. Then, we replace $(v_x, v_z)$ and $(v_z, v_y)$ with the known subgraphs in $\mathcal{F}^{(0)}$, as illustrated in middle of Figure 2 (f). By continuing this procedure, the homomorphism expressivity $\mathcal{F}$ is constructed, as illustrated in bottom of Figure 2 (f).

## 6.3 Expressivity Comparison

It is also convenient to obtain the upper bounds with regard to WL tests thanks to the relation of logic and WL studied in Cai et al. (1992), as well as comparing the expressive power of different GNN models:

**Proposition 9.** *Suppose the equivalent logic set of a class of GNN models is $\Phi$. Then, the expressive power of the GNN models is bounded by $k$-WL, iff all logic formulas in $\Phi$ can be expressed with at most $k$ variables.*

Note that it is trival to check the number of variables in our setting: recall that in Proposition 5 the equivalent logic sets are defined by specifying the grammar of logic formulas. This implies that we can simply check the number of variables emerged in the grammar. For example, consider Subgraph GNN (weak). There are 2 free variables $\{x,y\}$ in $\varphi(x) := \exists^{\geq N} y \left( \varphi'(y) \wedge \psi(x,y) \right) \mid \neg\varphi'(x) \mid \varphi'(x) \wedge \varphi''(x) \mid \mathbf{atp}(x)$ and 3 variables $\{x,y,z\}$ in $\psi(x,y) := \exists^{\geq N} z \left( \psi'(x,z) \wedge E(z,y) \right) \mid \neg\psi'(x) \mid \psi'(x) \wedge \psi''(x) \mid E(x,y)$. Therefore, the expressive power of Subgraph GNN (weak) is bounded by 3-WL. We summarize the section by introducing following results for popular GNN models.

**Corollary 10.** *The expressivity of GNN models satisfies: MPNNs = 1-WL < Subgraph GNNs (weak) = NBFNet < Subgraph GNNs (strong) < Local 2-FGNN < 2-FGNN = 3-WL, 1-WL < SEAL < 4-WL.*

## 7 Limitation and Conclusion

**Limitation.** The results of this paper are applicable to GNNs that can be expressed by GACNNs. Our framework is not applicable for GNNs which do not consist sole of aggregation and combination operations. For example, Graphormer-GD (Zhang et al., 2023) which injects distance information into node pairs and cannot be described by aggregation or combination layers.

**Conclusion.** In this paper we present a novel framework for systematically describe the logical expressivity of GNN models built upon combination and aggregation operations. We analyze the logical expressivity of popular GNN models and provide insight about many important topics in graph representation learning including expressivity comparison, structural awareness of GNNs, estimating WL expressivity, etc. Our framework serves as a toolbox to understand both existed and new GNN architectures: with new GNNs being designed, one can easily obtain the logical expressivity, study the substructures captured by them and bound these models with WL tests.

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

## A    RELATED WORKS

In this section we discuss related works that studies the expressive power of GNNs and several GNN models investigated in this paper.

**Expressivity of GNNs.**    Studying the expressive power of GNNs has been a hot topic in graph machine learning community. Xu et al. (2018) investigate the expressive power of GNNs by relating MPNNs with 1-WL tests, making it possible to utilize many know results about WL tests for the analysis of GNNs. Barceló et al. (2020) study the logical expressiveness of MPNNs which is close to our work. Compared with them, we successfully design a method to describe the logical expressivity for arbitrary aggregation-combination networks and analyzed several important implications brought by our work, including homomorphism expressivity, comparison of expressive power, etc. Zhang et al. (2024) investigate several popular GNN models and study their expressive power in the perspective of homomorphisms. Compared with their work, we propose a general method to determine the homomorphism expressivity for arbitrary aggregation-combination networks while solving a conjecture in Zhang et al. (2024) in the meantime.

To summarize, the major advantages of our works are:

(1) Our study establishes a deeper connection between GNNs and logic. Briefly speaking, previous works (Barceló et al., 2020) only study a set $\Phi$ of logic formulas for MPNNs such that:

- All logic formulas $\varphi \in \Phi$ can be captured by MPNNs.
- A logic formula can be captured by a MPNN *iff* it is in $\Phi$.

It is worth noting that the above statements only guarantees that the logic formulas can be captured by MPNNs. Intuitively, this does not establish an equivalence between logic and GNNs. Instead, in our work, we study a set $\Phi$ of logic formulas for GNNs such that:

- All logic formulas $\varphi \in \Phi$ can be captured by GNNs.
- The GNNs are equivalently expressive with $\Phi$ in distinguishing non-isomorphic graphs.

Therefore, our theory reveals deeper connection between GNNs and logic (i.e. the equivalence between GNNs and logic.)t

(2) Our study specifies the logical expressiveness of *arbitrary* GNNs, provided that their layers are built using aggregation and combination functions.

(3) In this paper, we establish a connection between logic and the homomorphism expressivity of GNNs.

**Higher order GNNs.**    Since the works of Xu et al. (2018); Morris et al. (2018) that relate GNNs with the 1-WL tests, it is straightforward to extend GNNs by imitating higher-order WL tests. Precisely, $k$-order WL tests assign colors for $k$-tuples of nodes and perform color aggregation between different tuples. Similarly, instead of learning representations for nodes, many works choose to apply the message passing paradigm in higher-order WL tests to GNNs and directly learn representations for node tuples (Morris et al., 2018; Maron et al., 2019a; 2018; 2019b; Keriven & Peyré, 2019; Azizian & Lelarge, 2020; Geerts & Reutter, 2022).

**Subgraph GNNs.**    Since the higher order GNNs are often too expensive for larger graphs, many works try to find cheaper ways to design more expressive GNNs. A variety of works feed subgraphs to MPNNs. At each layer, a set of subgraphs is generated according to some predefined permutation-invariant policies, including node deletion (Cotta et al., 2021), edge deletion Bevilacqua et al. (2021), node marking (Papp & Wattenhofer, 2022), ego-networks (Zhao et al., 2021; Zhang & Li, 2021; You et al., 2021). We will focus on the unified ESAN framework proposed by Bevilacqua et al. (2021). Qian et al. (2022); Frasca et al. (2022) studied the expressive power of different branches of subgraph GNNs.

**Substructure counting GNNs.**    There is another way to design GNNs that surpass 1-WL by constructing structural features for GNNs. Chen et al. (2020) showed that regular MPNNs cannot capture simple patterns such as cycles, cliques and paths. Bouritsas et al. (2020); Barcel'o et al. (2021)

proposed to apply substructure counting as pre-processing, and add substructure information into node features. Bodnar et al. (2021b;a); Thiede et al. (2021); Horn et al. (2021) further designed novel WL variants and proposed fully-neural approaches that captures complex substructures.

**GNNs for link prediction.** Standard GNNs learn representations for each node. Early methods such as GAE Kipf & Welling (2016a) use GNN as an encoder and decode link representations as a function over node representation pairs. These methods are problematic in capturing complex graph structures, and might lead to poor performance. Later on, labeling trick was introduced by SEAL Zhang & Chen (2018) and adopted by GraIL Teru et al. (2019), IGMC Zhang & Chen (2020), INDIGO Liu et al. (2021), etc. These methods encode source and target nodes to mark them differently from the rest of the graph, and are proved to be more powerful than GAE. ID-GNN You et al. (2021) and NBFNet Zhu et al. (2021) both augments GNNs with the identity of the source nodes. Besides, All-path Toutanova et al. (2016) encodes relations as linear projections and proposes to efficiently aggregate all paths with dynamic programming. However, All-Path is restricted to bilinear models, has limited link prediction capability and is also not inductive. EdgeTransformer Bergen et al. (2021a) utilizes attention mechanism to learn representations for nodes and links. While it also follows the 2-FWL message passing procedure, it operates directly on fully-connected graphs and have no proposals for simplifications as we do, thus it is not scalable to larger graphs. ELPH and BUDDY (Chamberlain et al., 2023) incorporate neighbor counting into node features to enhance the link prediction performance of MPNNs.

## B    WEISFEILER-LEHMAN TESTS

In this section we introduce the Weisfeiler-Lehman (WL) tests and their variants. Weisfeiler-Lehman (WL) tests are a family of necessary tests for graph isomorphism. Apart from some corner cases (Cai et al., 1992), they are effective and computationally efficient tests for graph isomorphism. Its 1-dimensional variant iteratively aggregates the colors of nodes and their neighbors and then injectively hashes them into new colors. The algorithms decides two graphs non-isomorphic if the colors of two graphs are different.

Extending from classic WL tests, $k$-dimensional WL test ($k$-WL) refines colors for node tuples. At beginning, the color of a node tuple $\mathbf{u}$ is set to be injective w.r.t. the structure of $\mathbf{u}$, denoted as $\mathbf{atp}(\mathbf{u})$. That is, for arbitrary two tuples $\mathbf{u} = (u_1, ..., u_k)$ and $\mathbf{v} = (v_1, ..., v_k)$, $\mathbf{atp}(\mathbf{u}) = \mathbf{atp}(\mathbf{v})$ *iff* there exists an isomorphism $\pi$ for the subgraphs induced by nodes in $\mathbf{u}, \mathbf{v}$ and $\pi(u_i) = v_i$ for $i = 1, ..., k$. $k$-WL then recursively refines these colors until convergence.

### B.1    1-WL (COLOR REFINEMENT)

The classic 1-WL test (Weisfeiler & Leman, 1968) maintains a color for each node which is refined by aggregating the colors of their neighbors. It can be easily applied on node-featured graphs (Xu et al., 2018) as in Algorithm 1.

---

**Algorithm 1:** The 1-WL test (color refinement)

**Input** : $G = (\boldsymbol{A}, \boldsymbol{X})$

1  $l \leftarrow 0$;
2  $c_v^0 \leftarrow \mathrm{hash}(\boldsymbol{x}_v)$ for all $v \in \mathcal{V}_G$;
3  **while** *not converge* **do**
4  $\quad$ $c_v^{l+1} \leftarrow \mathrm{hash}(c_v^l, \{\!\{c_u^l \mid u \in \mathcal{N}(v)\}\!\})$;
5  $\quad$ $l \leftarrow l + 1$;
6  **end**
7  **return** $\{\!\{c_v^l \mid v \in \mathcal{V}_G\}\!\}$;

---

The iteration converges when the partitions of nodes no longer changes. The 1-WL test decides two graphs are non-isomorphic if the multisets of colors of the two graphs are different. The WL algorithm successfully distinguishes most pairs of graphs, apart from some special examples such as regular graphs. Similarly, given a subset of nodes $C$, 1-WL define its color as $\{\!\{c_v^l \mid v \in C\}\!\}$, and 1-WL distinguishes two set of nodes if the colors of them are differernt.

### B.2   $k$-WL

The $k$-WL tests extend 1-WL to coloring $k$-tuples of nodes as in Algorithm 2, where we use $\boldsymbol{v}$ to denote a tuple of nodes, $G[\boldsymbol{v}]$ for ordered subgraphs. The neighbors $\mathcal{N}^k(\boldsymbol{v})$ are defined as follows: assume $\boldsymbol{v} = (v_1, ..., v_k)$, then $\mathcal{N}^k(\boldsymbol{v}) = (\mathcal{N}_1^k(\boldsymbol{v}), \mathcal{N}_1^k(\boldsymbol{v}), ..., \mathcal{N}_k^k(\boldsymbol{v}))$, where

$$\mathcal{N}_i^k(\boldsymbol{v}) = \{\{(v_1, ..., v_{i-1}, u, v_{i+1}, ..., v_k) \mid u \in \mathcal{V}\}\}.$$

---

**Algorithm 2:** The $k$-WL tests

**Input** : $G = (\boldsymbol{A}, \boldsymbol{X})$

1  $l \leftarrow 0$;
2  $c_{\boldsymbol{v}}^0 \leftarrow \mathrm{hash}(G[\boldsymbol{v}])$ for all $\boldsymbol{v} \in \mathcal{V}_G^k$;
3  **while** *not converge* **do**
4      $c_{\boldsymbol{v}}^{l+1} \leftarrow \mathrm{hash}(c_{\boldsymbol{v}}^l, \{\{c_{\boldsymbol{u}}^l \mid \boldsymbol{u} \in \mathcal{N}^k(\boldsymbol{v})\}\})$;
5      $l \leftarrow l + 1$;
6  **end**
7  **return** $\{\{c_{\boldsymbol{v}}^l \mid v \in \mathcal{V}_G \text{ for all } v \in \boldsymbol{v}\}\}$;

---

### B.3   $k$-FWL

The $k$-FWL (Cai et al., 1989) test is equally expressive with the $(k+1)$-WL test. It has the same initialization with $(k+1)$-WL. The neighbors $\mathcal{N}^k(\boldsymbol{v})$ are defined as follows: assume $\boldsymbol{v} = (v_1, ..., v_k)$, then $\mathcal{N}^k(\boldsymbol{v}) = \{\{\mathcal{N}_u^k(\boldsymbol{v}) \mid u \in \mathcal{V}\}\}$, where

$$\mathcal{N}_u^k(\boldsymbol{v}) = ((u, v_2, ..., v_k), (v_1, u, ..., v_k), ..., (v_1, ..., u, v_k)).$$

---

**Algorithm 3:** The $k$-FWL tests

**Input** : $G = (\boldsymbol{A}, \boldsymbol{X})$

1  $l \leftarrow 0$;
2  $c_{\boldsymbol{v}}^0 \leftarrow \mathrm{hash}(G[\boldsymbol{v}])$ for all $\boldsymbol{v} \in \mathcal{V}_G^k$;
3  **while** *not converge* **do**
4      $c_{\boldsymbol{v}}^{l+1} \leftarrow \mathrm{hash}(c_{\boldsymbol{v}}^l, \{\{c_{\boldsymbol{u}}^l \mid \boldsymbol{u} \in \mathcal{N}^k(\boldsymbol{v})\}\})$;
5      $l \leftarrow l + 1$;
6  **end**
7  **return** $\{\{c_{\boldsymbol{v}}^l \mid v \in \mathcal{V}_G \text{ for all } v \in \boldsymbol{v}\}\}$;

---

### B.4   COLORS OF $k$-WL / $k$-FWL

From the previous discussions $k$-WL and $k$-FWL both assign colors for $k$-tuples of nodes. The color of the graph $G$ is defined by

$$c_G = \mathrm{Hash}(\{\{c_{\boldsymbol{v}} \mid \boldsymbol{v} \in \mathcal{V}^k\}\}).$$

Similarly, given any subset of nodes $\mathcal{S} \subseteq \mathcal{V}$, we also define its color as

$$c_{\mathcal{S}} = \mathrm{Hash}(\{\{c_{\boldsymbol{v}} \mid \boldsymbol{v} \in \mathcal{S}^k\}\}).$$

## C   ABOUT GNN MODELS

In this section we decompose several popular GNN models using the GACNN framework. First, MPNNs Xu et al. (2018) whose layers are defined by

$$\chi^{(l+1)}(x) = \mathrm{COM}^{(l)}\left(\chi^{(l)}(x), \mathrm{AGG}^{(l)}\left(\{\{\chi^{(l)}(y) \mid y \in N(x)\}\}\right)\right),$$

where $x, y$ denotes nodes in graphs, can be simply decomposed into

$$\chi^{(l+1)}(x) = \mathrm{COM}^{(l)}\left(\chi^{(l)}(x), \chi_1^{(l)}(x)\right),$$

$$\chi_1^{(l)}(x) = \mathrm{AGG}^{(l)}\left(\left\{\!\!\left\{\chi^{(l)}(y) \mid v \in N(x)\right\}\!\!\right\}\right).$$

In this manner we describe the MPNN layers using the principled GACNN framework. To better explain our framework, consider Local 2-GNN Zhang et al. (2024) whose layers are defined by

$$\chi^{(l+1)}(x, y) = \mathrm{COM}^{(l)}\left(\chi^{(l)}(x, y), \mathrm{AGG}_1^{(l)}\left(\left\{\!\!\left\{\chi^{(l)}(z, y) \mid z \in \mathcal{N}(x)\right\}\!\!\right\}\right),\right.$$
$$\left.\mathrm{AGG}_2^{(l)}\left(\left\{\!\!\left\{\chi^{(l)}(x, z) \mid z \in \mathcal{N}(y)\right\}\!\!\right\}\right)\right). \tag{3}$$

Similarly, we can decompose one layer of Local 2-GNN into

$$\chi^{(l+1)}(x, y) = \mathrm{COM}_1^{(l)}\left(\chi^{(l)}(x, y), \chi_1^{(l)}(x, y)\right), \qquad \chi_1^{(l)}(x, y) = \mathrm{COM}_2^{(l)}\left(\chi_2^{(l)}(x, y), \chi_3^{(l)}(x, y)\right),$$

$$\chi_2^{(l)}(x, y) = \mathrm{AGG}_1^{(l)}\left(\left\{\!\!\left\{\chi^{(l)}(z, y) \mid z \in \mathcal{N}(x)\right\}\!\!\right\}\right), \chi_3^{(l)}(x, y) = \mathrm{AGG}_2^{(l)}\left(\left\{\!\!\left\{\chi^{(l)}(x, z) \mid z \in \mathcal{N}(y)\right\}\!\!\right\}\right),$$
$$\tag{4}$$

where $\mathrm{COM}_1^{(l)}, \mathrm{COM}_2^{(l)}$ are combination functions satisfying $\mathrm{COM}_1^{(l)}\left(\chi, \mathrm{COM}_2^{(l)}(\chi', \chi'')\right) = \mathrm{COM}^{(l)}(\chi, \chi', \chi'')$ for arbitrary representations $\chi, \chi', \chi''$.

**MPNN.**
$$\chi^{(l+1)}(x) = \mathrm{COM}\left(\chi^{(l)}(x), \mathrm{AGG}\left(\left\{\!\!\left\{\chi^{(l)}(y) \mid y \in \mathcal{N}(x)\right\}\!\!\right\}\right)\right).$$

**Subgraph GNN (weak).**
$$\chi^{(l+1)}(x) = \mathrm{AGG}\left(\left\{\!\!\left\{\chi^{(l+1)}(x, y) \mid y \in \mathcal{V}\right\}\!\!\right\}\right),$$
$$\chi^{(l+1)}(x, y) = \mathrm{COM}\left(\chi^{(l)}(x, y), \mathrm{AGG}\left(\chi^{(l)}(x, z) \mid z \in \mathcal{N}(y)\right)\right).$$

**Subgraph GNN (strong).**
$$\chi^{(l+1)}(x, y) = \mathrm{COM}\left(\chi^{(l)}(x, y), \mathrm{AGG}\left(\left\{\!\!\left\{\chi^{(l)}(x, z) \mid z \in \mathcal{N}(y)\right\}\!\!\right\}\right), \chi^{(l)}(y), \mathrm{AGG}\left(\left\{\!\!\left\{\chi^{(l)}(z) \mid z \in \mathcal{N}(y)\right\}\!\!\right\}\right)\right),$$
$$\chi^{(l+1)}(x) = \mathrm{AGG}\left(\chi^{(l+1)}(y, x) \mid y \in \mathcal{V}\right).$$

**NBFNet.**
$$\chi^{(l+1)}(x, y) = \mathrm{COM}\left(\chi^{(l)}(x, y), \mathrm{AGG}\left(\left\{\!\!\left\{\chi^{(l)}(x, z) \mid z \in \mathcal{N}(y)\right\}\!\!\right\}\right)\right).$$

**Local 2-GNN.**
$$\chi^{(l+1)}(x, y) = \mathrm{COM}\left(\chi^{(l)}(x, y), \mathrm{AGG}\left(\left\{\!\!\left\{\chi^{(l)}(z, y) \mid z \in \mathcal{N}(x)\right\}\!\!\right\}\right), \mathrm{AGG}\left(\left\{\!\!\left\{\chi^{(l)}(x, z) \mid z \in \mathcal{N}(y)\right\}\!\!\right\}\right)\right).$$

**2-FGNN.**
$$\chi^{(l+1)}(x, y) = \mathrm{COM}\left(\chi^{(l)}(x, y), \mathrm{AGG}\left(\left\{\!\!\left\{\mathrm{COM}\left(\chi^{(l)}(x, z), \chi^{(l)}(z, y)\right) \mid z \in \mathcal{V}\right\}\!\!\right\}\right)\right).$$

**SEAL (MPNN).**
$$\chi^{(l+1)}(x, z, y) = \mathrm{COM}\left(\chi^{(l)}(x, z, y), \mathrm{AGG}\left(\left\{\!\!\left\{\chi^{(l)}(x, w, y) \mid w \in \mathcal{N}(z)\right\}\!\!\right\}\right)\right),$$
$$\chi^{(l+1)}(x, y) = \mathrm{AGG}\left(\left\{\!\!\left\{\chi^{(l+1)}(x, z, y) \mid z \in \mathcal{N}\right\}\!\!\right\}\right).$$

**2-GNN**

$$\chi^{(l+1)}(x,y) = \text{COM}\left(\chi^{(l)}(x,y), \text{AGG}\left(\left\{\left\{\chi^{(l)}(z,y) \mid z \in \mathcal{V}\right\}\right\}\right), \text{AGG}\left(\left\{\left\{\chi^{(l)}(x,z) \mid z \in \mathcal{V}\right\}\right\}\right)\right).$$

## D    NUMERICAL EXPERIMENTS

In this section we perform numerical experiments to empirically validate our theoretical results. The motivation of this section is to test the logical expressivity of several GNN models by checking whether they are able to express specific logic formulas on synthetic data. Generally, the experiments are divided into two parts: one to test the *node classification* capacities of GNNs and the other to test the *link prediction* capacities of GNNs.

The experiments are designed as below. We conduct two types of tasks. First, we would like to test the GNNs' ability to learn logic formulas. In this part, we specify the target logic formula $\varphi$ we want to learn. Then, we randomly generate graphs with initial node colors encoded by zero-one vectors. After that, we apply $\varphi$ on the generated graphs and label each node $x$ (or node pair $(x, y)$) where $\varphi(x)$ (or $\varphi(x, y)$) is **true** with target label 1. We then train each GNN model on the graphs and observe the results. For each task the GNNs are divided into two classes: one for node classification and the other for link prediction. Second, we would like to test the GNNs' ability to distinguish non-isomorphic graphs. In this part, we manually construct several pairs of graphs and show that they can be distinguished by different logic formulas. Then, we test whether GNNs can separate the graphs. The detail of each experiment is presented below.

**Experiment setting.**    For all GNNs we choose the aggregation AGG to be sum, and the combination function COM to be
$$\text{COM}(\boldsymbol{x}, \boldsymbol{y}) = \boldsymbol{W}[\boldsymbol{x}^T, \boldsymbol{y}^T]^T,$$
where $\boldsymbol{W}$ is a parameter matrix. We refer to GNN-$k$ as the GNN model with $k$ layers, i.e. MPNN-1 refers to a MPNN which has one aggregation-combination layer. For the last layer, the sigmoid function is selected as the activation function. For the rest of the layers, the ReLU function is selected as the activation function.

We randomly generate graphs as follows. We consider Erdös-Renyi graphs, which are random graphs by specifying $N$ the number of nodes and $p$ the possibility for each edge to exist. We then randomly color each node with a specified probability. After that, we apply the logic formulas on the generated graphs and obtain the prediction targets for the GNNs. Each train and test graph contains 500 nodes. Each test-larger graph contains 1000 nodes.

### D.1    LEARNING LOGICAL FORMULAS

#### D.1.1    NODE CLASSIFICATION

**Target logic formulas.**    For node classification we consider three target logic formulas with increasing complexity:
$$\varphi_1(x) := \text{Red}(x) \wedge \exists y(E(y,x) \wedge \exists z(E(z,y) \wedge \text{Blue}(z))),$$
$$\varphi_2(x) := \exists y \exists z(E(x,y) \wedge E(x,z) \wedge E(y,z)),$$
$$\varphi_3(x) := \exists y(E(y,x) \wedge \varphi_2(y)).$$

We pick GNN models by analyzing the above formulas. $\varphi_1$ is expressed by MPNNs. $\varphi_2$ is expressed by Subgraph GNNs (weak) but not MPNNs. $\varphi_3$ is expressed by 2-FGNNs but not Subgraph GNNs (weak). Hence, we choose MPNNs, Subgraph GNNs (weak) and 2-FGNNs. For Subgraph GNNs, we follow the node marking policy (Bevilacqua et al., 2021) and set the initial color of $(x, x)$ to be the color of $x$. For 2-FGNNs, the node representation of node $x$ is obtained by the representation of $(x, x)$. The results are shown in Table 1. We can see that the results meet our expectation.

**Results about $\varphi_1$.**    $\varphi_1$ decides whether a node is red and is connected to a blue node via 2 edges. It can be decomposed by $\varphi_1(x) := \text{Red}(x) \wedge \exists y(E(y,x) \wedge \psi(y))$ and $\psi(y) := \text{Red}(y) \wedge \exists z(E(z,y) \wedge \text{Blue}(z))$, thus it is simple and can be expressed by MPNNs. Also, according to the decomposition two MPNN layers are required to express $\varphi_1$. Similarly, two Subgraph GNN layers are required to express $\varphi_1$. The results in Table 1 satisfy this.

Table 1: Results on node classification.

| Algorithm | $\varphi_1$ | | | $\varphi_2$ | | | $\varphi_3$ | | |
|---|---|---|---|---|---|---|---|---|---|
| | Train | Test | Test-larger | Train | Test | Test-larger | Train | Test | Test-larger |
| MPNN-1 | 0.907 | 0.903 | 0.909 | 0.867 | 0.865 | 0.666 | 0.866 | 0.865 | 0.527 |
| MPNN-2 | 1.000 | 1.000 | 1.000 | 0.863 | 0.863 | 0.556 | 0.868 | 0.878 | 0.657 |
| MPNN-3 | 1.000 | 1.000 | 1.000 | 0.855 | 0.864 | 0.601 | 0.877 | 0.877 | 0.477 |
| Subgraph-1 | 0.905 | 0.903 | 0.899 | 0.867 | 0.872 | 0.668 | 0.882 | 0.859 | 0.524 |
| Subgraph-2 | 1.000 | 1.000 | 1.000 | 1.000 | 1.000 | 1.000 | 0.914 | 0.906 | 0.682 |
| Subgraph-3 | 1.000 | 1.000 | 1.000 | 1.000 | 1.000 | 1.000 | 0.901 | 0.905 | 0.582 |
| 2-FGNN-1 | 1.000 | 1.000 | 1.000 | 0.865 | 0.862 | 0.666 | 0.857 | 0.847 | 0.531 |
| 2-FGNN-2 | 1.000 | 1.000 | 1.000 | 1.000 | 1.000 | 1.000 | 0.891 | 0.896 | 0.639 |
| 2-FGNN-3 | 1.000 | 1.000 | 1.000 | 1.000 | 1.000 | 1.000 | 1.000 | 1.000 | 1.000 |

**Results about $\varphi_2$.** $\varphi_2$ decides whether the node is in a 3-clique and requires at least three variables $, y, z$ and thus cannot be expressed by MPNNs. However, it can be expressed by Subgraph GNNs: we can decompose $\varphi_2$ into $\varphi_2(x) := \exists y(\psi_1(x,y)), \psi_1(x,y) := E(x,y) \wedge \exists z(\psi_2(x,z) \wedge (E(y,z)),$ $\psi_2(x,z) := E(x,z)$, which directly corresponds to the equivalent logic set of Subgraph GNNs. Also, it can be seen from the decomposition that two Subgraph GNN layers are required to express $\varphi_2$, since there are two nested formulas $\psi_1, \psi_2$. For 2-FGNNs, two layers are required: we have $\varphi_2(x) := \psi(x,x)$ and $\psi(x,x) := \exists y(\psi_1(x,y), E(y,x)), \psi_1(x,y) := \exists z(E(x,z) \wedge E(z,y))$. The results in Table 1 meet the expectation.

**Results about $\varphi_3$.** $\varphi_3$ decides whether the node is adjacent to a 3-clique, which cannot be expressed by Subgraph GNNs. For 2-FGNNs, we can decompose $\varphi_3$ into $\varphi_3(x) := \psi(x,x), \psi(x,x) := \exists y(\psi_1(x,y) \wedge E(y,x)), \psi_1(x,y) := \exists z(\psi_2(y,z)), \psi_2(y,z) := E(y,z) \wedge \exists w(E(y,w) \wedge E(w,z))$, thus it can be expressed by 2-FGNNs with 3 layers. The results in Table 1 meet the expectation.

### D.1.2 Link Prediction

**Target logic formulas.** For link prediction we consider two target logic formulas with increasing complexity:

$$\varphi_1(x,y) := \mathrm{Red}(x) \wedge \mathrm{Blue}(y),$$
$$\varphi_2(x,y) := \exists z(\mathrm{Blue}(z) \wedge E(x,z) \wedge E(z,y)).$$

We pick MPNNs and NBFNet. For MPNNs, the representation of a node pair $(x,y)$ is obtained by combining the representation of node $x$ and node $y$. The results are presented in Table 2.

**About $\varphi_1$.** $\varphi_1$ is extremely simple: it decides two nodes are connected if one of them is red and the other is blue. This logic formula does not consider the relation between the two nodes, and can be expressed by MPNNs. The results in Table 2 meet the expectation.

**About $\varphi_2$.** Compared with $\varphi_1$, $\varphi_2$ is more complex and decides two nodes are connected if they are connected via a blue node. Such form of $\varphi_2$ is more practical and considers the relation between the two node: for example, deciding whether one person is another one's grandparent in a kinship graph can be done by checking whether they are connected via another person with the predicate "parent". However, MPNNs fail to express $\varphi_2$. NBFNet is sufficient for expressing $\varphi_2$, as we can decompose it into $\varphi_2(x,y) := \exists z(\psi(x,z) \wedge E(z,y))$ and $\psi(x,z) := E(x,z) \wedge \mathrm{Blue}(z)$. The results in Table 2 meet the expectation, where we can see that there exists a large margin between MPNNs and NBFNet for learning $\varphi_2$.

Table 2: Results on link prediction.

| Algorithm | $\varphi_1$ | | | $\varphi_2$ | | |
|---|---|---|---|---|---|---|
| | Train | Test | Test-larger | Train | Test | Test-larger |
| MPNN-1 | 1.000 | 1.000 | 1.000 | 0.818 | 0.819 | 0.586 |
| MPNN-2 | 1.000 | 1.000 | 1.000 | 0.817 | 0.817 | 0.328 |
| NBFNet-1 | 1.000 | 1.000 | 1.000 | 1.000 | 1.000 | 1.000 |
| NBFNet-2 | 1.000 | 1.000 | 1.000 | 1.000 | 1.000 | 1.000 |
| 2-FGNN-1 | 1.000 | 1.000 | 1.000 | 1.000 | 1.000 | 1.000 |
| 2-FGNN-2 | 1.000 | 1.000 | 1.000 | 1.000 | 1.000 | 1.000 |

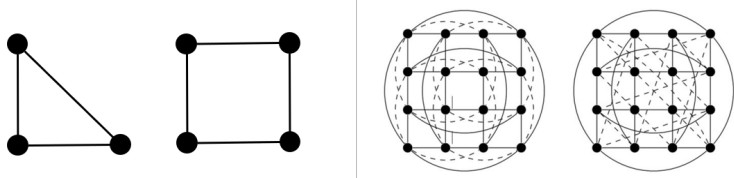

Figure 3: An illustration of several graph pairs.

## D.2 SEPARATION POWER

We design several non-isomorphic graph pairs and test whether GNNs can distinguish the nodes / node tuples in them. This includes:

- A cycle with 3 nodes and a cycle with 4 nodes, as illustrated in Figure 3 left. They are 2-regular graphs. The task is to separate the nodes. The nodes can be separated by $\varphi(x) := \exists y(\psi(x,y)), \psi(x,y) := \exists z(E(x,z) \wedge E(z,y) \wedge E(z,y))$. It cannot be expressed by MPNNs, but can be expressed by Subgraph GNNs (weak) and more powerful models.
- Shrikhande Graph and Rook's 4x4 graph as illustrated in Figure 3 right. The task is to separate the graph. They are strongly regular graphs which cannot be separated by 2-FWL, and thus 2-FGNNs.

The results are presented in Table 3.

## E ABOUT VARIABLES IN LOGIC FORMULAS

Consider the formula

$$\varphi(x) := \mathrm{Red}(x) \wedge \exists y(E(y,x) \wedge \exists z(E(z,y))).$$

The formula has a *free variable* $x$ which is not bounded by the quantifier $\exists$ and two *quantified variables* $y, z$ which are bounded by $\exists$. Therefore, the formula has 3 variables in total. Given a graph $G$, a *grounding* of $\varphi(x)$ in $G$ is a mapping $\eta$ from the variables in $\varphi(x)$ to the nodes in $G$. For example consider the graph in Figure 4. There are two groundings $\eta_1, \eta_2$ from $\varphi(x)$ to it, with $\eta_1(x) = v_1, \eta_1(y) = v_2, \eta_1(z) = v_3$ and $\eta_2(x) = v_1, \eta_2(y) = v_2, \eta_2(z) = v_5$.

To reduce the number of symbols used in logic formula, there is a trick which is to *reuse* the variable $x$ and replace every occurrence of $z$ in $\varphi$ with $x$, leading to:

$$\varphi'(x) := \mathrm{Red}(x) \wedge \exists y(E(y,x) \wedge \exists x(E(x,y))).$$

To ground $\varphi'(x)$ on $G$, one still needs to substitute the variables in $\varphi'(x)$ with the nodes in $G$. This indicates that in Figure 4, we need to substitute the outer variable $x$ in $\mathrm{Red}(x)$ with $v_1$ and the inner variable $x$ in $\exists x(E(x,y))$ with $v_3$ or $v_5$. Therefore, when the variables are reused, the grounding is no longer a well-defined mapping from variables to nodes, and the essentially different variables $x, z$ in $\varphi(x)$ are expressed by the same symbol $x$ in $\varphi'(x)$. To avoid such clunky situations, we avoid the reuse of variables.

Table 3: Results on separation power.

| Graph | MPNN | Subgraph | 2-FGNN | SEAL |
|---|---|---|---|---|
| 2-regular | ✗ | ✓ | ✓ | ✓ |
| Shrikhande & Rook's 4x4 | ✗ | ✗ | ✗ | ✓ |

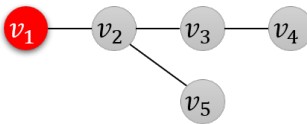

Figure 4: An example graph which has two groundings for $\varphi(x) := \mathrm{Red}(x) \wedge \exists y (E(y,x) \wedge \exists z(E(z,y)))$.

**The properties of $F$ constructed by $\varphi$.** Recall that to construct the homomorphism expressivity, we construct a graph $F$ for $\varphi$ which is defined below:

1. There exists a bijective mapping $\tau$ from the variables in $\varphi$ to the nodes in $F$.

2. For any variables $x, y$ in $\varphi$, $E(x, y)$ is a term in $\varphi$ *iff* $E(\tau(x), \tau(y))$ is an edge of $F$.

We define the concept of injective grounding:

**Definition 11.** An injective grounding from a logic formula $\varphi$ to a graph $G$ is a grounding from $\varphi$ to $G$ that maps different variables in $\varphi$ to different nodes in $G$ (without the reuse of variables).

It is now obvious that $F$ is the minimum graph that contains an injective grounding from $\varphi$.

## F  LINK PREDICTION AND LOGIC

In this section we briefly discuss how the logical expressivity of GNNs affects their link prediction capabilities. We also attempt to explain, why compared to node classification or graph classification tasks, there often exists a larger margin between MPNNs and other advanced GNN variants (see, e.g. Zhu et al. (2021)).

**Beginning with MPNNs.** The logic formulas expressed by MPNNs are of the form

$$\varphi(x) := \exists^{\geq N} y \left( \varphi'(y) \wedge E(y, x) \right) \mid \neg \varphi' \mid \varphi' \wedge \varphi''.$$

For MPNNs, to predict whether a pair of nodes $(x, y)$ are linked, it is common to combine the representation of $x$ and $y$, e.g.

$$\mathrm{pred}(x, y) = \mathrm{MLP}(\boldsymbol{h}_x, \boldsymbol{h}_y),$$

where $\boldsymbol{h}_x, \boldsymbol{h}_y$ are the representations of $x$ and $y$ respectively. Using Theorem 3, it is evident that pred is expressed by

$$\psi(x, y) := \varphi(x) \mid \varphi(y) \mid \neg \psi' \mid \psi' \wedge \psi''. \tag{5}$$

Intuitively, this can be understood as gathering the information of $x$ and $y$ respectively, which does not consider the correlation between $x$ and $y$ existed in the structure of the graph. To explain this more concisely, consider Figure 5 (a). For arbitrary $\varphi$ defined above, $\varphi(v_1) = \varphi(v_2) = \varphi(v_3)$ since the nodes $v_1, v_2, v_3$ are isomorphic. This implies that for arbitrary $\psi$ defined above, $\psi(v_1, v_2) = \psi(v_1, v_3)$, and thus $\psi$ fail to separate $(v_1, v_2)$ between $(v_1, v_3)$. However, it is obvious that the structure between $(v_1, v_2)$ and $(v_1, v_3)$ are different: $v_1$ and $v_2$ are connected via a node while $v_1$ and $v_3$ are not connected. This illustrates that for the link prediction of $(x, y)$, MPNNs can only check each node $x, y$ separately, and fails to consider the correlation between them. This is also reflected in the definition of $\psi$ in Eq. 5: $\psi(x, y)$ is constructed by $\varphi(x)$ and $\varphi(y)$ separately, and there does not exist logic formulas besides $\psi$ that jointly consider $x$ and $y$.

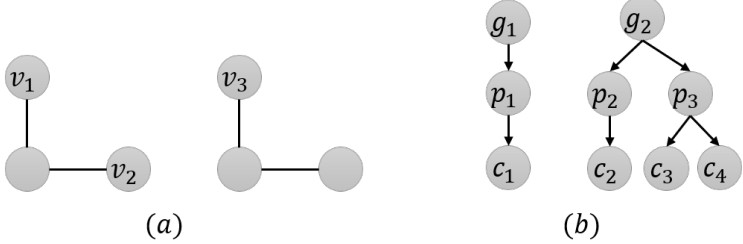

Figure 5: An example graph for link prediction tasks.

**More powerful variants.** To explain what is crucial for link prediction, consider the simple task: given a graph of family kinship, the target is to predict whether one person $x$ is another person $y$'s grandparent. In Figure 5 (b) where each node represents a person and the directed edges represent the relation "parent". This is a typical link prediction task, which cannot be done by MPNNs: as discussed above, MPNNs are confused by $(g_1, c_1)$ and $(g_2, c_1)$ and thus can mistakenly considers $g_2$ as the grandparent of $c_1$, also they are not relatives. The correct logic formula for this task can be written as

$$\text{Grandparent}(x, y) := \exists z(\text{Parent}(x, z) \land \text{Parent}(z, y)). \tag{6}$$

We believe this simple formula reflects the key logical patterns that should be considered in link prediction tasks, which can be written in the form of

$$\varphi(x, y) := \exists^{\geq N} z(\varphi'(x, z) \land \varphi''(z, y)) \mid \neg\varphi' \mid \varphi' \land \varphi'', \tag{7}$$

which is used by 2-FGNNs. Compared with MPNNs, this logic formula directly models the correlation between $x, y$ with $\varphi(x, y)$. Moreover, the structures between $x$ and $y$ is perceived by checking the relation between each node $z$ and $(x, y)$ via $\exists^{\geq N} z(\varphi'(x, z) \land \varphi''(z, y))$. For example, Figure 5 (b), the relation between $g_1, c_1$ is judged by checking whether there is an intermediate node $p_1$ connected to both of them via $\exists^{\geq N} p_1(\text{Parent}(g_1, p_1) \land \text{Parent}(p_1, c_1))$. We also note that there is a variant weaker than Eq. 7 but also empirically performs well for link prediction:

$$\varphi(x, y) := \exists^{\geq N} z(\varphi'(x, z) \land E(z, y)) \mid \neg\varphi' \mid \varphi' \land \varphi'', \tag{8}$$

which is used by NBFNet. This variant also expresses the Grandparent predicate.

To summarize, we believe the reason that MPNNs are relatively weak for link prediction tasks is that they cannot express the relation between arbitrary pair of nodes as Eq. 7 and Eq. 8 do. Intuitively, MPNNs can only express concepts like "$x$ is a grandparent, $y$ is a grandchild, then $x$ is the grandparent of $y$", which can be described by the following logic formula within the expressiveness of MPNNs:

$$\text{Grandparent}(x, y) := \text{IsGrandparent}(x) \land \text{IsChild}(y),$$
$$\text{IsGrandparent}(x) := \exists y(\text{Parent}(x, y) \land \exists z(\text{Parent}(y, z))),$$
$$\text{IsGrandchild}(x) := \exists y(\text{Parent}(y, x) \land \exists z(\text{Parent}(z, y))).$$

Therefore, the ability of expressing Eq. 7 or Eq. 8 is required for describing the Grandparent predicate. In reality, we notice that many advanced GNN models designed for link prediction (Zhu et al., 2021; Liu et al., 2021; Zhang & Chen, 2018; Teru et al., 2019; You et al., 2021; Bergen et al., 2021b) follows this intuition.

## G  PROOF

### G.1  PROOF OF THEOREM 3

**Theorem 3.** *Let $\mathcal{X}_i$ be the set of all possible $\chi_i$ defined above for $i \in [K]$. There exists $\{\Phi_1, ..., \Phi_K\}$ defined below, such that $\Phi_K$ is the equivalent logic set of $\chi_K$.*

- $\chi_i(\mathbf{u}) = \text{AGG}_i\left(\{\{\chi_j(\mathbf{v}) \mid \mathbf{v} \in \mathcal{N}_i(\mathbf{u})\}\}\right)$
  $\iff \varphi_i(\mathbf{u}) := \exists^{\geq N} \mathbf{v}\left(\varphi_j(\mathbf{v}) \land \mathbf{1}_{\mathbf{v} \in \mathcal{N}_i(\mathbf{u})}\right) \mid \neg\varphi'_i(\mathbf{u}) \mid \varphi'_i(\mathbf{u}) \land \varphi''_i(\mathbf{u}),$

- $$\chi_i(\mathbf{u}) = \mathrm{COM}_i\left(\chi_j(\mathbf{u}), \chi_k(\mathbf{u})\right)$$
  $$\iff \varphi_i(\mathbf{u}) := \varphi_j(\mathbf{u}) \mid \varphi_k(\mathbf{u}) \mid \neg\varphi_i'(\mathbf{u}) \mid \varphi_i'(\mathbf{u}) \wedge \varphi_i''(\mathbf{u}),$$

- $$\chi_i(\mathbf{u}) = \mathrm{INIT}_i(\mathbf{u}) \iff \varphi_i(\mathbf{u}) := \mathbf{atp}(\mathbf{u}) \mid \neg\varphi_i'(\mathbf{u}) \mid \varphi_i'(\mathbf{u}) \wedge \varphi_i''(\mathbf{u}),$$

*where $\varphi_i', \varphi_i'' \in \Phi_i, \varphi_j \in \Phi_j, \varphi_k \in \Phi_k$ for $i, j, k \in [K]$.*

*Proof sketch.* The theorem represents a major technical contribution, so we present a proof sketch below. Our proof is divided into two parts, presented in Appendix G.1. First, we show that each logic formula $\varphi_i \in \Phi_1$ can be captured by $\chi_i$. Obviously, for $i = 1$, we only have $\varphi_1(\mathbf{u}) := \mathbf{atp}(\mathbf{u}) \mid \neg\varphi_1'(\mathbf{u}) \mid \varphi_1'(\mathbf{u}) \wedge \varphi_1''(\mathbf{u})$, which can be directly described by $\chi_1(\mathbf{u}) = \mathrm{INIT}_1(\mathbf{u})$. By induction on $i$, we suppose all $\varphi_j \in \Phi_j$ for $j < i$ can be captured by $\chi_j$. Then, given arbitrary $\varphi_i$, we provide a method to explicitly construct the corresponding $\{\mathrm{COM}_j\}_{j\in[i]}, \{\mathrm{AGG}_j\}_{j\in[i]}$ and $\{\mathrm{INIT}_j\}_{j\in[i]}$ functions so that $\varphi_i$ is captured by $\chi_i$.

In the next step, we prove that for any graphs $G, H$ and $\mathbf{u} \in \mathcal{V}_G^k, \mathbf{v} \in \mathcal{V}_H^k$, $\chi_i$ cannot distinguish $\mathbf{u}, \mathbf{v}$ *iff* all $\varphi_i \in \Phi_i$ classify $\mathbf{u}, \mathbf{v}$ the same. By utilizing the fact that each $\varphi_i \in \Phi_i$ is captured by $\chi_i$, the first direction is proved. It is therefore only necessary to show $\chi_i(\mathbf{u}) \neq \chi_i(\mathbf{v}) \Rightarrow$ there exists $\varphi_i \in \Phi_i$ satisfying $\varphi_i(\mathbf{u}) \neq \varphi_i(\mathbf{v})$. Again, we prove by induction on $i$. Suppose for all $j < i$ the statement holds. We then enumerate all possible cases where $\chi_i(\mathbf{u}) \neq \chi_i(\mathbf{v})$: for example if $\chi_i(\mathbf{u}) = \mathrm{COM}_i(\chi_j(\mathbf{u}), \chi_k(\mathbf{u}))$, then there are three cases: (1) $\chi_j(\mathbf{u}) \neq \chi_j(\mathbf{v}), \chi_k(\mathbf{u}) = \chi_k(\mathbf{v})$; (2) $\chi_j(\mathbf{u}) = \chi_j(\mathbf{v}), \chi_k(\mathbf{u}) \neq \chi_k(\mathbf{v})$, and (3) $\chi_j(\mathbf{u}) \neq \chi_j(\mathbf{v}), \chi_k(\mathbf{u}) \neq \chi_k(\mathbf{v})$. For each case, we provide a method to construct $\varphi_i \in \Phi_i$ satisfying $\varphi_i(\mathbf{u}) \neq \varphi_i(\mathbf{v})$, thus concluding the proof. $\square$

*Proof.* Recall that the definition of equivalent logic set $\Phi_K$ of a family of graph functions $\mathcal{X}_K$ is defined as:

1. The order of each $\varphi_K \in \Phi_K$ matches that of $\chi_K \in \mathcal{X}_K$;

2. For all $\varphi_K \in \Phi_K$, there exists $\chi_K \in \mathcal{X}_K$ such that for arbitrary graphs $G$ and $\mathbf{u} \in \mathcal{V}_G^k$, $\varphi_K(\mathbf{u}) = \mathbf{true}$ *iff* $\chi_K(\mathbf{u}) = \mathbf{true}$;

3. A logic formula is captured by $\mathcal{X}$ *iff* it is in $\Phi$;

4. Given arbitrary positive integer $N$ and $\chi_K \in \mathcal{X}_K$, there exists $\varphi_K \in \Phi_K$ satisfying: for any graphs $G$ with no more than $N$ nodes and $\mathbf{u} \in \mathcal{V}_G^k$, $\varphi_K(\mathbf{u}) = \mathbf{true}$ *iff* $\chi_K(\mathbf{u}) = \mathbf{true}$.

5. Given any graphs $G, H$ and $\mathbf{u} \in \mathcal{V}_G^k, \mathbf{v} \in \mathcal{V}_H^k$, all $\chi_K \in \mathcal{X}_K$ cannot distinguish $\mathbf{u}, \mathbf{v}$ *iff* all logic formulas $\varphi_K \in \Phi_K$ classify $\mathbf{u}, \mathbf{v}$ the same.

The proof is divided into four parts, each corresponding to one of the statement.

**Statement 1. The order of each $\varphi_K \in \Phi_K$ matches that of $\chi_K \in \mathcal{X}_K$.** The statement naturally holds.

**Statement 2. For all $\varphi_K \in \Phi_K$, there exists $\chi_K \in \mathcal{X}_K$ such that for arbitrary graphs $G$ and $\mathbf{u} \in \mathcal{V}_G^k$, $\varphi_K(\mathbf{u}) = \mathbf{true}$ *iff* $\chi_K(\mathbf{u}) = \mathbf{true}$.** We prove this by manually constructing the aggregation and combination functions such that $\varphi_K$ is captured by $\chi_K$. To do this, we first preset the following definition.

**Definition 12.** Given a finite set of logic formulas $\Phi$, we define a graph function $\chi$ to fully expresses $\Phi$, if the output of $\chi$ is a vector of binary values whose dimension is $|\Phi|$. Moreover, for each $\varphi \in \Phi$, there exists a dimension $d$ of the output of $\chi$ such that for arbitrary graph $G$ and node tuple $\mathbf{u}$, $\chi(\mathbf{u})[d] = \varphi(\mathbf{u})$.

Using the definition above, we have the following lemmas.

**Lemma 13.** *Let $\Phi_1$ be a set of logic formulas which is fully expressed by $\chi_1$. Let $\Phi$ be defined by*

$$\varphi(\mathbf{u}) := \exists^{\geq n}\mathbf{v}\left(\varphi_1(\mathbf{v}) \wedge \mathbf{1}_{\mathbf{v}\in\mathcal{N}(\mathbf{u})}\right) \mid \neg\varphi'(\mathbf{u}) \mid \varphi'(\mathbf{u}) \wedge \varphi''(\mathbf{u}),$$

*where* $\varphi, \varphi', \varphi'' \in \Phi$, *and we restrict* $n < N$ *where* $N$ *is an arbitrary positive integer. Then, there exists an aggregation function* $\mathrm{AGG}$ *such that* $\chi(\mathbf{u}) = \mathrm{AGG}(\{\!\{\chi_1(\mathbf{v}) \mid \mathbf{v} \in \mathcal{N}(\mathbf{u})\}\!\})$ *fully expresses* $\Phi$.

**Lemma 14.** *Let* $\Phi_1, \Phi_2$ *be a set of logic formulas which is fully expressed by* $\chi_1, \chi_2$. *Let* $\Phi$ *be defined by*

$$\varphi(\mathbf{u}) := \varphi_1(\mathbf{u}) \mid \varphi_2(\mathbf{u}) \mid \neg\varphi_2'(\mathbf{u}) \mid \varphi_2'(\mathbf{u}) \wedge \varphi_2''(\mathbf{u}),$$

*where* $\varphi, \varphi', \varphi'' \in \Phi$. *Then, there exists a combination function* $\mathrm{COM}$ *such that* $\chi(\mathbf{u}) = \mathrm{COM}(\chi_1(\mathbf{u}), \chi_2(\mathbf{u}))$ *fully expresses* $\Phi$.

With Lemma 13, 14, we are now ready to show that for any $\varphi_K \in \Phi_K$, there is a $\chi_K \in \mathcal{X}_K$ such that $\varphi_K(\mathbf{u}) = \chi_K(\mathbf{u})$ holds for arbitrary $\mathbf{u}$. We prove this by induction.

Let $\{\Phi_1, ..., \Phi_K\}$ be sets of logic formulas defined in Theorem 3. For $i \in [K]$, we denote $\Phi_i^N$ to be the fragment of $\Phi_i$ where the superscripts $n$ of quantifiers $\exists^{\geq n}$ are less than $N$. Moreover, we define $N$ to be the maximum superscript of the quantifiers in $\varphi_K$. Then, at beginning for $\Phi_1$, we only have

$$\varphi_1(\mathbf{u}) := \mathbf{atp}(\mathbf{u}) \mid \neg\varphi_1'(\mathbf{u}) \mid \varphi_1'(\mathbf{u}) \wedge \varphi_1''(\mathbf{u}),$$

where $\varphi_1, \varphi_1', \varphi_1'' \in \Phi_1$. Obviously, $\Phi_1$ is finite, and there exists an init function $\mathrm{INIT}_1$ such that $\chi_1(\mathbf{u}) = \mathrm{INIT}_1(\mathbf{u})$ fully expresses $\Phi_1$. Next, suppose at some iteration $i$, there exists a function $\chi_i$ that fully expresses $\Phi_i^N$. Then for $\Phi_{i+1}^N$, by directly utilizing the results in Lemma 13, 14, it is obvious that there still exists $\chi_{i+1}$ that fully expresses $\Phi_{i+1}^N$. Finally, at iteration $K$, let $\chi_K'$ be the function that fully expresses $\Phi_K^N$. From Definition 12 it is evident that the output of $\chi_K'$ has a dimension for each formula in $\Phi_K^N$. By only preserving the dimension corresponding to $\varphi_K$ (denoted as dimension $d$), there exists $\chi_K(\mathbf{u}) = \chi_K'(\mathbf{u})[d]$ that satisfies: for arbitrary graphs $G$ and $\mathbf{u} \in \mathcal{V}_G^k$, $\varphi_K(\mathbf{u}) = \mathbf{true}$ *iff* $\chi_K(\mathbf{u}) = \mathbf{true}$.

**Statement 3. A logic formula is captured by $\mathcal{X}$ *iff* it is in $\Phi$.** To prove the statement, we need to show that

**Lemma 15.** *Given arbitrary* $\mathcal{X}$, *let* $\Phi$ *be constructed as in Theorem 3. Then, all logic formulas captured by a* $\chi \in \mathcal{X}$ *is in* $\Phi$.

The lemma is proved via a reduction from GACNNs to MPNNs in Appendix G.14.

**Statement 4. Given arbitrary positive integer $N$ and $\chi_K \in \mathcal{X}_K$, there exists $\varphi_K \in \Phi_K$ satisfying: for any graphs $G$ with no more than $N$ nodes and $\mathbf{u} \in \mathcal{V}_G^k$, $\varphi_K(\mathbf{u}) = \mathbf{true}$ *iff* $\chi_K(\mathbf{u}) = \mathbf{true}$.** We first introduce the following lemma.

**Lemma 16.** *Given a series of functions* $\{\chi_1, ..., \chi_K\}$ *where each function* $\chi_i$ *maps a node tuple to a discrete color and is defined by either* $\chi_i(\mathbf{u}) = \mathrm{AGG}_i(\{\!\{\chi_j(\mathbf{v}) \mid \mathbf{v} \in \mathcal{N}_i(\mathbf{u})\}\!\}), \chi_i(\mathbf{u}) = \mathrm{COM}_i(\chi_j(\mathbf{u}), \chi_k(\mathbf{u}))$ *or* $\chi_i(\mathbf{u}) = \mathrm{INIT}_i(\mathbf{u})$. *Let* $\mathcal{X}_i$ *be the set of all possible* $\chi_i$ *defined in the above manner for* $i \in [K]$. *Let* $\Phi_i$ *be the set specified by Theorem 3 of* $\mathcal{X}_i$. *Then, for arbitrary* $i \in [K]$ *and each possible color* $\mathbf{col}$ *of* $\chi_i$, *there exists a logic formula* $\varphi_i^{\mathbf{col}} \in \Phi_i$ *satisfying:*

$$\varphi_i^{\mathbf{col}}(\mathbf{u}) = \mathbf{true} \iff \chi_i(\mathbf{u}) = \mathbf{col}$$

*for arbitrary graph $G$ with no more than $N$ nodes and $\mathbf{u} \in \mathcal{V}_G^k$.*

Therefore, given $\{\chi_1, ..., \chi_K\}$ and $\{\Phi_1, ..., \Phi_K\}$ the sets of logic formulas defined in Theorem 3, Lemma 16 indicates that for arbitrary $N$, there exists a logic formula $\varphi_i^{\mathbf{col}} \in \Phi_i$ for each $\chi_i$ and $\mathbf{col}$ a possible color of $\chi_i$ such that $\varphi_i^{\mathbf{col}}(\mathbf{u}) = \mathbf{true} \iff \chi_i(\mathbf{u}) = \mathbf{col}$. It is evident that for $\chi_K$, the statement: $\varphi_K(\mathbf{u}) = \mathbf{true}$ *iff* $\chi_K(\mathbf{u}) = \mathbf{true}$ is just a special case where $\chi_K$ only has binary colors. Therefore, the proof completes.

**Statement 5. Given any graphs $G, H$ and $\mathbf{u} \in \mathcal{V}_G^k, \mathbf{v} \in \mathcal{V}_H^k$, all $\chi_K \in \mathcal{X}_K$ cannot distinguish $\mathbf{u}, \mathbf{v}$ *iff* all logic formulas $\varphi_K \in \Phi_K$ classify $\mathbf{u}, \mathbf{v}$ the same.** This result is a direct corollary of the statements 2, 3. Let $N = \max(|\mathcal{V}_G|, |\mathcal{V}_H|)$. It is obvious that if there is a $\chi_K \in \mathcal{X}_K$ that $\chi_K(\mathbf{u}) \neq \chi_K(\mathbf{v})$, there also exists a $\varphi_K \in \Phi_k$ where $\varphi_K$ is specified in statement 3 that satisfies $\varphi_K(\mathbf{u}) \neq \varphi_K(\mathbf{v})$, and vice versa. $\qquad\square$

## G.2  PROOF OF THEOREM 8

**Theorem 8.** *Given a class of GACNN models and suppose $\Phi$ be the equivalent logic set. Let $\mathcal{F}$ be the homomorphism expressivity constructed by $\Phi$ as discussed above. For all pairs of graphs $G, H$. the following statements are equivalent:*

1. $\mathbf{Hom}(F, G) = \mathbf{Hom}(F, H)$ *for all $F \in \mathcal{F}$.*
2. *All GACNNs do not distinguish $G$ and $H$.*

*Proof sketch.* Theorem 8 represents another major technical contribution of the paper, so we present a proof sketch below. Given the homomorphism expressivity $\mathcal{F}$ and for any graph $G$, the intuition is that we can use logic formulas to count the number of homomorphisms from each $F \in \mathcal{F}$ to $G$. Consider the graph in Figure 2 (a) for example: the number of homomorphisms from it to $G$ is $N$ *iff* $\varphi := \exists^{=N} \mathbf{v}(E(v_x, v_y) \wedge E(v_y, v_w) \wedge E(v_y, v_z) \wedge E(v_w, v_u))$ evaluates **true** in $G$, where $\mathbf{v} = (v_x, v_y, v_z, v_w, v_u)$. The next step of our proof is involved and shows that such $\varphi$ can be expressed by logic formulas in the equivalent logic set and vice versa, which is presented in Appendix G.2, thus concluding the proof. $\qquad\square$

*Proof.* From Proposition 5 it is evident that statement 2 is equivalent to: All $\varphi \in \Phi$ do not distinguish $G$ and $H$. We therefore instead prove:

All $\varphi \in \Phi$ do not distinguish $G$ and $H$ $\iff$ $\mathbf{Hom}(F, G) = \mathbf{Hom}(F, H)$ for all $F \in \mathcal{F}$.

We first prove the direction left to right. Given each $\varphi \in \Phi$ that does not contain negation $\neg$ or $\exists^{\geq N}$ where $N \geq 2$, recall that the corresponding $F$ is constructed by:

1. Start from an empty graph $F$;

2. Construct the nodes of $F$: Add a node $v_x$ for each variable $x$ emerged in $\varphi$ (we avoid the reuse of variables);

3. Construct the structure of $F$: Add an edge $(v_x, v_y)$ for each edge term $E(x, y)$ in $\varphi$. Add $F$ to $\mathcal{F}$.

Before we start, we first introduce some useful quantifiers $\exists^{=N}, \exists^{\leq N}$ which express "there exists exactly $N$" and "there exists no more than $N$" respectively. Note that the two quantifiers can be directly deduced by $\exists^{\geq N}$: $\exists^{\leq N} := \neg \exists^{\geq N+1}$ and $\exists^{=N} := \exists^{\geq N} \wedge \exists^{\leq N}$.

Note that all $\varphi$ satisfying the constraint (i.e. without negation or $\exists^{\geq N}$ for $N > 1$) can be flattened into the form of:
$$\varphi := \exists \mathbf{x}_1 \exists \mathbf{x}_2 ... \exists \mathbf{x}_K E(x_{i_1}, x_{j_1}) \wedge ... \wedge E(x_{i_M}, x_{j_M}), \tag{9}$$
where $i_p, j_p \in [K]$ for all $p \in [M]$. We now prove that there is a logic formula $\psi \in \Phi$ that captures $\mathbf{Hom}(F, G) = 1$, i.e. given arbitrary graph $G$, $\psi$ is **true** *iff* $\mathbf{Hom}(F, G) = 1$. Suppose $F$ is constructed by $\varphi$ as:

$$\varphi := \exists \mathbf{x}_1 \exists \mathbf{x}_2 ... \exists \mathbf{x}_K E(x_{i_1}, x_{j_1}) \wedge ... \wedge E(x_{i_M}, x_{j_M}).$$

Then by letting

$$\psi := \exists^{=1} \mathbf{x}_1 \exists^{=1} \mathbf{x}_2 ... \exists^{=1} \mathbf{x}_K E(x_{i_1}, x_{j_1}) \wedge ... \wedge E(x_{i_M}, x_{j_M}).$$

We now show that for arbitrary graph $G$, $\psi$ is **true** *iff* $\mathbf{Hom}(F, G) = 1$. By the construction of $F$ each variable $\mathbf{x}_1, ..., \mathbf{x}_K$ is corresponded to a distinct node tuple in $F$ and each term $E(x_{i_1}, x_{j_1}), ..., E(x_{i_m}, x_{j_m})$ is corresponded to an distinct edge in $F$. If $\psi$ is **true** on $G$, then there exists a grounding $\mathbf{x}_1 \to \mathbf{u}_1, ..., \mathbf{x}_K \to \mathbf{u}_K$ such that all $E(x_{i_1}, x_{j_1}), ..., E(x_{i_m}, x_{j_m})$ are **true**. Then the mapping $\pi : F \to G$ that $\pi(v_{x_l}) = u_l$ for $l \in [K]$ where $v_{x_l}$ is the node in $F$ corresponding to the variable $x_l$ in $\psi$. obviously $\pi$ is a homomorphism from $F$ to $G$, therefore $\mathbf{Hom}(F, G) \geq 1$. Further more, suppose $\mathbf{Hom}(F, G) > 1$, then there exists another $\pi' \neq \pi$ that is also a homomorphism from $F$ to $G$, which indicates that the grounding for $E(x_{i_1}, x_{j_1}), ..., E(x_{i_M}, x_{j_M})$ to be **true** is not unique. In this case, $\psi$ is not **true** because there exists not only one $\mathbf{x}_1, ..., \mathbf{x}_K$ such that $E(x_{i_1}, x_{j_1}), ..., E(x_{i_M}, x_{j_M})$ is **true**, violating the quantifiers $\exists^{=1}$ in $\psi$.

We next prove that there is also a logic formula $\psi \in \Phi$ that captures $\textbf{Hom}(F, \cdot) = N$ for arbitrary $N$: that is, for any graphs $G, H$, $\textbf{Hom}(F, G) \neq \textbf{Hom}(F, H) \Rightarrow$ there exists $\psi$ such that $\psi$ evaluates to different values on $G$ and $H$. We prove by contradiction and assume all $\psi \in \Phi$ evaluates to the same value on $G$ and $H$. We denote $\psi_{n_1 n_2 \dots n_K} := \exists^{=n_1} \mathbf{x}_1 \exists^{=n_2} \mathbf{x}_2 \dots \exists^{=n_K} \mathbf{x}_K \left( \bigwedge_{m \in [M]} E(x_{i_m}, x_{j_m}) \right)$.

By assumption $\psi_{n_1 n_2 \dots n_K}$ evaluates to the same value on $G, H$ for any $n_1, n_2, \dots, n_K$. We now construct a $\psi$ such that $\psi$ evaluates to $\textbf{true}$ on $G$.

Let $F$ be constructed by $varphi := \exists \mathbf{x}_1 \exists \mathbf{x}_2 \dots \exists \mathbf{x}_K E(x_{i_1}, x_{j_1}) \wedge \dots \wedge E(x_{i_M}, x_{j_M})$ such that $\textbf{Hom}(F, G) = N \neq \textbf{Hom}(F, H)$. Let $\mathbf{X}$ be the tuple of all variables in $\varphi$: $\mathbf{X} = (\mathbf{x}_1, \mathbf{x}_2, \dots, \mathbf{x}_k)$. For all homomorphisms from $F$ to $G$, let $\mathbb{V}$ be the set of images of $\mathbf{X}$ from $F$ to $G$. We first pick all $\psi_{n_1 n_2 \dots n_K}$ that evaluates to $\textbf{true}$ on $G$. First, we have the following result:

**Lemma 17.** *If $n_k > N$ for some $k \in [K]$, then $\psi_{n_1 n_2 \dots n_K} = \textbf{false}$ on $G$.*

Therefore, there are only finite $\psi_{n_1 n_2 \dots n_K}$ that evaluates to $\textbf{true}$ on $G$, and $n_k \leq N$ for all $k \in [K]$. For some $\psi_{n_1 n_2 \dots n_K} = \textbf{true}$ on $G$, we refer to its *grounding* as a mapping from the variables $\mathbf{X}$ in $\psi$ to the corresponding tuple of nodes $\mathbf{V}$ in $G$, and $\mathbf{V}$ is the *grounding result*. We then have the following result:

**Lemma 18.** *For $\psi_{n_1 n_2 \dots n_K}$ and $\psi_{m_1 m_2 \dots m_K}$, if $n_k \neq m_k$ for some $k \in [K]$, then then grounding results of $\psi_{n_1 n_2 \dots n_K}$ and $\psi_{m_1 m_2 \dots m_K}$ are different, i.e. there exists no $\mathbf{V}$ that is both a grounding result of $\psi_{n_1 n_2 \dots n_K}$ and $\psi_{m_1 m_2 \dots m_K}$.*

Let $\mathbb{S} = \{ \psi_{n_1 n_2 \dots n_K} \mid n_k \leq N \text{ for } k \in [K], \psi_{n_1 n_2 \dots n_K} = \textbf{true} \text{ on } G \} = \{ \psi_{n_1^l n_2^l \dots n_K^l} \mid l \in [L] \}$ where $L = |\mathbb{S}|$. Then we have the following result:

$$\phi := \bigwedge_{l \in [L]} \psi_{n_1^l n_2^l \dots n_K^l} \in \Phi \text{ evaluates } \textbf{true} \text{ on } G.$$

Moreover, according to Lemma 18 it is evident that the total number of different grounding results can be evaluated as

$$\sum_{l \in [L]} \text{Number of grounding results of } \psi_{n_1^l n_2^l \dots n_K^l}$$

$$= \sum_{l \in [L]} \prod_{k \in [K]} n_k^l$$

Since a grounding is also exactly a homomorphism from $F$ to $G$, we have

$$\sum_{l \in [L]} \text{Number of grounding results of } \psi_{n_1^l n_2^l \dots n_K^l}$$

$$= \textbf{Hom}(F, G) = N.$$

By assumption, $\phi$ also evaluates to $\textbf{true}$ on $H$, which indicates that $\psi_{n_1^l n_2^l \dots n_K^l}$ evaluates to $\textbf{true}$ for $l \in [L]$ on $H$. As a result,

$$\textbf{Hom}(F, H)$$

$$\geq \sum_{l \in [L]} \text{Number of grounding results of } \psi_{n_1^l n_2^l \dots n_K^l}$$

$$= N.$$

This yields a contradiction where we assume $\textbf{Hom}(F, H) < \textbf{Hom}(F.G) = N$. Thus the proof completes.

We next prove the other direction, i.e. for any graphs $G, H$, there exists $\psi$ such that $\psi$ evaluates to different values on $G$ and $H \Rightarrow \textbf{Hom}(F, G) \neq \textbf{Hom}(F, H)$. Without loss of generality we assume $\psi$ evaluates to $\textbf{true}$ on $G$ and $\textbf{false}$ on $H$. We first introduce the following lemma:

**Lemma 19.** *If there is $\psi \in \Phi$ such that $\psi$ evaluates to different values on $G$ and $H$, then there exists $\psi \in \Phi$ in the form of $\psi_{n_1 n_2 \dots n_K} := \exists^{=n_1} \mathbf{x}_1 \exists^{=n_2} \mathbf{x}_2 \dots \exists^{=n_K} \mathbf{x}_K \left( \bigwedge_{p \in [P]} E(x_{i_p}, x_{j_p}) \bigwedge_{q \in [Q]} \neg E(x_{i_q}, x_{j_q}) \right)$ that also evaluates to different values on $G$ and $H$.*

Without loss of generality, we may now assume that there exists $\psi$ in the form described by Lemma 19 that evaluates to **true** on $G$ and **false** on $H$. We now prove that there must exists a $F \in \mathcal{F}$ such that $\textbf{Hom}(F, G) \neq \textbf{Hom}(F, H)$.

Given

$$\phi := \exists \mathbf{x}_1 \exists \mathbf{x}_2 ... \exists \mathbf{x}_K \left( \bigwedge_{p \in [P]} E(x_{i_p}, x_{j_p}) \bigwedge_{q \in [Q]} \neg E(x_{i_q}, x_{j_q}) \right),$$

let $\textbf{grd}(\phi, G)$ be the number of groundings from the variables in $\phi$ to $G$. Obviously $\textbf{grd}$ is a extension to $\textbf{hom}$ which allows negative edges $\neg E(x_{i_q}, x_{j_q})$ for $q \in [Q]$. We have the following result:

**Lemma 20.** *If* $\textbf{Hom}(F, G) = \textbf{Hom}(F, H)$ *for all* $F \in \mathcal{F}$*, then every* $\phi \in \Phi$ *of the above form satisfies:*

$$\textbf{grd}(\phi, G) = \textbf{grd}(\phi, H).$$

We define

$$\phi_{n_1 n_2 ... n_K} := \exists \mathbf{x}_1^1 \exists \mathbf{x}_1^2 ... \exists \mathbf{x}_{x1}^{n_1} \exists \mathbf{x}_2^{11} \mathbf{x}_2^{12} ... \exists \mathbf{x}_2^{1 n_2} \exists \mathbf{x}_2^{21} \exists \mathbf{x}_2^{22} ... \exists \mathbf{x}_2^{n_1 n_2} ... \exists \mathbf{x}_K^{11...1} ... \exists \mathbf{x}_K^{n_1 n_2 ... n_K}$$
$$\left( \bigwedge_{\substack{p \in [P],}} E(x_{i_p}^{l_1 l_2 ... l_{i_p}}, x_{j_p}^{l_1 l_2 ... l_{j_p}}) \bigwedge_{q \in [Q]} \neg E(x_{i_q}^{l_1 l_2 ... l_{i_q}}, x_{j_q}^{l_1 l_2 ... l_{j_q}}) \right).$$

Note that $\phi_{n_1 n_2 ... n_K} \in \Phi$ for arbitrary $n_1, n_2, ..., n_K$. We now prove by contradiction. Assume $\textbf{Hom}(F, G) = \textbf{Hom}(F, H)$ for arbitrary $F \in \mathcal{F}$. By Lemma 20 it is obvious that $\textbf{grd}(\phi_{n_1 n_2 ... n_K}, G) = \textbf{grd}(\phi_{n_1 n_2 ... n_K}, H)$ for all $n_1, n_2, ..., n_K$. First consider the case where $\psi_{11...1} = \textbf{true}$ on $G$. Obviously we have $\textbf{grd}(\phi_{11...1}, G) = 1 = \textbf{grd}(\phi_{11...1}, H)$, which indicates that there exists exactly one grounding from $\phi_{11...1}$ to $H$ and thus $\psi_{11...1} = \textbf{true}$ on $H$, which yields a contradiction. We now assume $\psi_{N_1 N_2 ... N_K}$ evaluates to **true** on $G$. We prove that the evaluation of $\psi_{N_1 N_2 ... N_K}$ can be determined by the number of groundings from $\{ \phi_{n_1 n_2 ... n_K} \mid n_k \in [N_k] \text{ for } k \in [K] \}$ to $G$, i.e.

$$\left( \textbf{grd}(\phi_{n_1 n_2 ... n_K}, G) \right)_{n_k \in [N_k] \text{ for } k \in [K]}.$$

By proof by induction, we already know that $\psi_{11...1}$ evaluates to **true** on $G$ exactly when $\textbf{grd}(\phi_{11...1}, G) = 1$ and thus $\psi_{11...1}$ is captured by $(\textbf{grd}(\phi_{11...1}, G))$. Now consider we are to prove $\psi_{N_1 N_2 ... N_K}$ is captured by $(\textbf{grd}(\phi_{n_1 n_2 ... n_K}, G))_{n_k \in [N_k] \text{ for } k \in [K]}$. Let us consider the groundings from $\phi_{N_1 N_2 ... N_K}$ to $G$. Obviously we can divide the groundings into two parts:

1. The non-injective groundings, i.e. groundings that maps different variables in $\phi_{N_1 N_2 ... N_K}$ to the same node in $G$.

2. The injective groundings, i,e, groundings that maps different variables in $\phi_{N_1 N_2 ... N_K}$ to different nodes in $G$.

Obviously the number of non-injective groundings can be computed by $\textbf{grd}(\phi_{n_1 n_2 ... n_K}, G)$ where for all $k \in [K]$ $n_k \leq N_k, k \in [K]$ and there exists $k \in [K]$ $n_k < N_k$. Thus, the number of injective homomorphisms can be evaluated. If the following constraints hold:

- The number of injective groundings from $\phi_{n_1 n_2 ... n_K}$ to $G$ is larger than $0$,

- The numbers of injective groundings from $\phi_{n_1 + 1 n_2 ... n_K}, \phi_{n_1 n_2 + 1 ... n_K} ... \phi_{n_1 n_2 ... n_K + 1}$ to $G$ are $0$,

then obviously $\psi_{n_1 ... n_K}$ evaluates to **true**. Therefore this yields a contradiction and the proof completes.

$\square$

### G.3 PROOF OF COROLLARY 4

**Corollary 4.** *The equivalent logic set of $l$-layer GACNNs defined above is given by $\Phi^{(l)}$.*

*Proof.* This is a direct result derived from Theorem 3 when we explicitly write down the computation procedure of a $l$-layer GACNNs. $\qquad\square$

### G.4 PROOF OF PROPOSITION 5

**Proposition 5.** *The equivalent logic set of all GACNNs defined above is given by $\Phi = \bigcup_{l=0}^{\infty} \Phi^{(l)}$. Moreover, let $\Phi_i = \bigcup_{l=0}^{\infty} \Phi_i^{(l)}$ for $i \in [K]$, then $\Phi$ and $\{\Phi_i\}_{i\in[K]}$ exist and is defined by a similar procedure as Theorem 3. For the brevity of notation we denote $\chi^{(l)}$ as $\chi_{K+1}^{(l)}$, $\chi^{(l+1)}$ as $\chi_0^{(l)}$ and $\Phi$ as $\Phi_0$ in the following description.*

- $$\chi_i^{(l)}(\mathbf{u}) = \mathrm{AGG}\left(\left\{\!\left\{\chi_j^{(l)}(\mathbf{x}) \mid \mathbf{v} \in \mathcal{N}_i(\mathbf{u})\right\}\!\right\}\right)$$
  $$\iff \varphi_i(\mathbf{u}) := \exists^{\geq N} \mathbf{v}\left(\varphi_j(\mathbf{v}) \wedge \mathbf{1}_{\mathbf{v}\in\mathcal{N}_i(\mathbf{u})}\right) \mid \neg\varphi_i'(\mathbf{u}) \mid \varphi_i'(\mathbf{u}) \wedge \varphi_i''(\mathbf{u}) \mid \mathbf{atp}(\mathbf{u}),$$

- $$\chi_i^{(l)}(\mathbf{u}) = \mathrm{COM}\left(\chi_j^{(l)}(\mathbf{u}), \chi_k^{(l)}(\mathbf{u})\right)$$
  $$\iff \varphi_i(\mathbf{u}) := \varphi_j(\mathbf{u}) \mid \varphi_k(\mathbf{u}) \mid \neg\varphi_i'(\mathbf{u}) \mid \varphi_i'(\mathbf{u}) \wedge \varphi_i''(\mathbf{u}) \mid \mathbf{atp}(\mathbf{u}),$$

*where $\varphi_i, \varphi_i', \varphi_i'' \in \Phi_i, \varphi_j \in \Phi_j, \varphi_k \in \Phi_k$ for $i \in \{0\} \cup [K]$.*

*Proof.* Obviously, to consider all numbers of layers $l$ simultaneously, the equivalent logic set is given by $\Phi = \bigcup_{l=0}^{\infty} \Phi^{(l)}$. Since for layer $l$ the corresponding equivalent logic set $\Phi_l$ is given by

- $$\chi_i^{(l)}(\mathbf{u}) = \mathrm{AGG}\left(\left\{\!\left\{\chi_j^{(l)}(\mathbf{x}) \mid \mathbf{v} \in \mathcal{N}_i(\mathbf{u})\right\}\!\right\}\right)$$
  $$\iff \varphi_i^{(l)}(\mathbf{u}) := \exists^{\geq N} \mathbf{v}\left(\varphi_j^{(l)}(\mathbf{v}) \wedge \mathbf{1}_{\mathbf{v}\in\mathcal{N}_i(\mathbf{u})}\right) \mid \neg\varphi^{(l)}{}_i'(\mathbf{u}) \mid \varphi^{(l)}{}_i'(\mathbf{u}) \wedge \varphi^{(l)}{}_i''(\mathbf{u}),$$

- $$\chi_i^{(l)}(\mathbf{u}) = \mathrm{COM}\left(\chi_j^{(l)}(\mathbf{u}), \chi_k^{(l)}(\mathbf{u})\right)$$
  $$\iff \varphi_i^{(l)}(\mathbf{u}) := \varphi_j^{(l)}(\mathbf{u}) \mid \varphi_k^{(l)}(\mathbf{u}) \mid \neg\varphi^{(l)}{}_i'(\mathbf{u}) \mid \varphi^{(l)}{}_i'(\mathbf{u}) \wedge \varphi^{(l)}{}_i''(\mathbf{u}),$$

where $\varphi_i^{(l)}, \varphi_i^{(l)\prime}, \varphi_i^{(l)\prime\prime} \in \Phi_i^{(l)}, \varphi_j^{(l)} \in \Phi_j^{(l)}, \varphi_k^{(l)} \in \Phi_k^{(l)}$ for $i \in \cup[K]$, and
$$\varphi^0(\mathbf{u}) := \mathbf{atp}(\mathbf{u}).$$

It is obvious that the construction of $\Phi_i$ in Proposition 5 is a union of all $\Phi_i^{(l)}$ for $l \in [0, \infty)$: at beginning $\varphi_i := \mathbf{atp}(\mathbf{u})$ thus at this moment $\Phi_i = \Phi_i^{(0)}$. Suppose at some iteration $\Phi_i = \Phi_i^{(l)}$. Then in next iteration we add

$$\varphi_i(\mathbf{u}) := \exists^{\geq N} \mathbf{v}\left(\varphi_j(\mathbf{v}) \wedge \mathbf{1}_{\mathbf{v}\in\mathcal{N}_i(\mathbf{u})}\right) \mid \varphi_j(\mathbf{u}) \mid \varphi_k(\mathbf{u}) \mid \neg\varphi_i'(\mathbf{u}) \mid \varphi_i'(\mathbf{u}) \wedge \varphi_i''(\mathbf{u}) \mid \mathbf{atp}(\mathbf{u})$$

to $\Phi_i$, and we still have $\Phi_i = \Phi_i^{(l+1)}$. Therefore $\Phi = \bigcup_{l=0}^{\infty} \Phi^{(l)}$ is given by Proposition 5. $\qquad\square$

### G.5 PROOF OF PROPOSITION 6

**Proposition 6.** *The equivalent logic set $\Psi$ of the graph representation $\chi_G$ defined above is given by*
$$\psi := \exists^{\geq N}\left(\varphi(\mathbf{u})\right) \mid \neg\psi' \mid \psi' \wedge \psi'',$$
*where $\psi, \psi', \psi'' \in \Psi, \varphi \in \Phi$.*

*Proof.* This is a direct result derived from Theorem 3. Since
$$\chi_G = \mathrm{AGG}\left(\left\{\!\left\{\chi(\mathbf{u}) \mid \mathbf{u} \in \mathcal{V}^k\right\}\!\right\}\right)$$
where $\mathcal{V}$ is the set of nodes in $G$ and $k$ is the order of $\mathbf{u}$, $\Psi$ is specified by Theorem 3 as above. $\qquad\square$

### G.6 PROOF OF PROPOSITION 7

**Proposition 7.** *The equivalent logic sets of GNN models can be separately defined as:*

- **MPNN:** $\varphi(x) := \exists^{\geq N} x \left( \varphi'(y) \wedge E(x, y) \right)$, *where $E$ is the edge predicate.*
- **Subgraph GNN (weak):** $\varphi(x) := \exists^{\geq N} y \left( \psi(x, y) \right)$, *and* $\psi(x, y) := \exists^{\geq N} z \left( \psi'(x, z) \wedge E(z, y) \right)$.
- **Subgraph GNN (strong):** $\varphi(x) := \exists^{\geq N} y \left( \varphi'(y) \wedge \psi(y, x) \right), \psi(x, y) := \exists^{\geq N} z \left( \psi(x, z) \wedge E(z, y) \right) \mid \varphi(y)$.
- **NBFNet:** $\varphi(x, y) := \exists^{\geq N} z \left( \varphi'(x, z) \wedge E(z, y) \right)$.
- **Local 2-GNN:** $\varphi(x, y) := \exists^{\geq N} z \left( \varphi'(x, z) \wedge E(z, y) \right) \mid \exists^{\geq N} z \left( E'(x, z) \wedge \varphi'(z, y) \right)$
- **2-FGNN:** $\varphi(x, y) := \exists^{\geq N} z \left( \varphi'(x, z) \wedge \varphi''(z, y) \right)$.
- **SEAL (MPNN):** $\varphi(x, y) := \exists^{\geq N} z \left( \psi(x, z, y) \right), \psi(x, z, y) := \exists^{\geq N} w \left( \psi(x, w, y) \wedge E(w, z) \right)$.
- **2-GNN:** $\varphi(x, y) := \exists^{\geq N} z \left( \varphi'(x, z) \right) \mid \exists^{\geq N} z \left( \varphi'(z, y) \right)$.

*Proof sketch.* The proposition directly utilize the results in Proposition 5 to derive the equivalent logic sets of popular GNNs. We present a proof sketch for Local 2-GNNs and illustrate the pipeline for determining the expressivity of certain GNN models. First, we explicitly write down the GNN layers as Eq. 3. Then, we transform the GNN layers into GACNN layers by decomposing each layer into a sequence of AGG and COM functions, as in Eq. 3. By utilizing Proposition 5, we can directly obtain the equivalent logic set of Local 2-GNNs as below. (Again, we omit the terms in the form of $\varphi := \neg \varphi' \mid \varphi' \wedge \varphi'' \mid$ **atp** for notation brevity.)

$$\varphi(x, y) := \varphi'(x, y) \mid \varphi_1(x, y), \qquad \varphi_1(x, y) := \varphi_2(x, y) \mid \varphi_3(x, y),$$
$$\varphi_2(x, y) := \exists^{\geq N} z (\varphi(z, y) \wedge E(x, z)), \quad \varphi_3(x, y) := \exists^{\geq N} z (\varphi(x, z) \wedge E(z, y)),$$

where $\varphi, \varphi' \in \Phi$ is the equivalent logic set of Local 2-GNNs, and $\varphi_1 \in \Phi_1, \varphi_2 \in \Phi_2, \varphi_3 \in \Phi_3$ are auxiliary logic sets. It is therefore only one step before the result in Proposition 7: by substituting the definition of $\varphi_2, \varphi_3$ into $\varphi_1$ and further substituting the definition of $\varphi_1$ into $\varphi$, we can write down the definition of $\varphi$ into one line:

$$\varphi(x, y) := \varphi'(x, y) \mid \exists^{\geq N} z \left( \varphi'(x, z) \wedge E(z, y) \right) \mid \exists^{\geq N} z \left( E(x, z) \wedge \varphi'(z, y) \right).$$

Removing the redundant term $\varphi(x, y) := \varphi'(x, y)$ directly yields the result in Proposition 7. $\square$

*Proof.* By utilizing the results from Proposition 5 and further simplify the resulted equivalent logic sets, we can easily obtain these results. Note that for brevity we omit the terms $\varphi := \neg \varphi' \mid \varphi' \wedge \varphi''$.

**MPNN.**
$$\chi^{(l+1)}(x) = \text{COM}\left( \chi^{(l)}(x), \text{AGG}\left( \left\{\!\!\left\{ \chi^{(l)}(y) \mid y \in \mathcal{N}(x) \right\}\!\!\right\} \right) \right)$$
$$\Rightarrow \varphi(x) := \varphi'(x) \mid \exists^{\geq N} x \left( \varphi'(y) \wedge E(x, y) \right)$$
$$\Rightarrow \varphi(x) := \exists^{\geq N} x \left( \varphi'(y) \wedge E(x, y) \right).$$

**Subgraph GNN (weak).** The layers are given by
$$\chi^{(l+1)}(x) = \text{AGG}\left( \left\{\!\!\left\{ \chi^{(l+1)}(x, y) \mid y \in \mathcal{V} \right\}\!\!\right\} \right),$$
$$\chi^{(l+1)}(x, y) = \text{COM}\left( \chi^{(l)}(x, y), \text{AGG}\left( \chi^{(l)}(x, z) \mid z \in \mathcal{N}(y) \right) \right).$$

Therefore the equivalent logic sets are given by
$$\varphi(x) := \exists^{\geq N} y \left( \psi(x, y) \right),$$
$$\psi(x, y) := \psi'(x, y) \mid \exists^{\geq N} z \left( \psi'(x, z) \wedge E(z, y) \right),$$

which can be directly simplified as
$$\varphi(x) := \exists^{\geq N} y \left( \psi(x, y) \right),$$
$$\psi(x, y) := \exists^{\geq N} z \left( \psi'(x, z) \wedge E(z, y) \right),$$

**Subgraph GNN (strong).**  The layers are given by

$$\chi^{(l+1)}(x,y) = \mathrm{COM}\left(\chi^{(l)}(x,y), \mathrm{AGG}\left(\left\{\!\!\left\{\chi^{(l)}(x,z) \mid z \in \mathcal{N}(y)\right\}\!\!\right\}\right), \chi^{(l)}(y), \mathrm{AGG}\left(\left\{\!\!\left\{\chi^{(l)}(z) \mid z \in \mathcal{N}(y)\right\}\!\!\right\}\right)\right),$$
$$\chi^{(l+1)}(x) = \mathrm{AGG}\left(\chi^{(l+1)}(y,x) \mid y \in \mathcal{V}\right).$$

Therefore the equivalent logic sets are given by

$$\psi(x,y) := \psi'(x,y) \mid \exists^{\geq N} z \left(\psi(x,z) \wedge E(z,y)\right) \mid \varphi(y) \mid \exists^{\geq N} z \left(\varphi(z) \wedge E(z,y)\right),$$
$$\varphi(x) := \exists^{\geq N} y \left(\psi(y,x)\right).$$

Substituting $\psi(x,y) := (\varphi(z) \wedge E(z,y))$ to the second line leads to

$$\varphi(x) := \exists^{\geq N} z \left(\varphi'(z) \wedge E(z,x)\right).$$

Therefore, the above $\Phi$ can also be described by

$$\varphi(x) := \exists^{\geq N} y \left(\varphi'(y) \wedge \psi(y,x)\right),$$
$$\psi(x,y) := \exists^{\geq N} z \left(\psi(x,z) \wedge E(z,y)\right) \mid \varphi(y).$$

**NBFNet.**

$$\chi^{(l+1)}(x,y) = \mathrm{COM}\left(\chi^{(l)}(x,y), \mathrm{AGG}\left(\left\{\!\!\left\{\chi^{(l)}(x,z) \mid z \in \mathcal{N}(y)\right\}\!\!\right\}\right)\right)$$
$$\Rightarrow \varphi(x,y) := \varphi'(x,y) \mid \exists^{\geq N} z \left(\varphi'(x,z) \wedge E(z,y)\right)$$
$$\Rightarrow \varphi(x,y) := \exists^{\geq N} z \left(\varphi'(x,z) \wedge E(z,y)\right).$$

**Local 2-GNN.**

$$\chi^{(l+1)}(x,y) = \mathrm{COM}\left(\chi^{(l)}(x,y), \mathrm{AGG}\left(\left\{\!\!\left\{\chi^{(l)}(z,y) \mid z \in \mathcal{N}(x)\right\}\!\!\right\}\right), \mathrm{AGG}\left(\left\{\!\!\left\{\chi^{(l)}(x,z) \mid z \in \mathcal{N}(y)\right\}\!\!\right\}\right)\right)$$
$$\Rightarrow \varphi(x,y) := \varphi'(x,y) \mid \exists^{\geq N} z \left(\varphi'(x,z) \wedge E(z,y)\right) \mid \exists^{\geq N} z \left(E'(x,z) \wedge \varphi'(z,y)\right)$$
$$\Rightarrow \varphi(x,y) := \exists^{\geq N} z \left(\varphi'(x,z) \wedge E(z,y)\right) \mid \exists^{\geq N} z \left(E'(x,z) \wedge \varphi'(z,y)\right).$$

**2-FGNN.**

$$\chi^{(l+1)}(x,y) = \mathrm{COM}\left(\chi^{(l)}(x,y), \mathrm{AGG}\left(\left\{\!\!\left\{\mathrm{COM}\left(\chi^{(l)}(x,z), \chi^{(l)}(z,y)\right) \mid z \in \mathcal{V}\right\}\!\!\right\}\right)\right)$$
$$\Rightarrow \varphi(x,y) := \varphi'(x,y) \mid \exists^{\geq N} z \left(\psi(x,y,z)\right), \psi(x,y,z) := \varphi(x,y) \mid \varphi(y,z)$$
$$\Rightarrow \varphi(x,y) := \exists^{\geq N} z \left(\varphi'(x,z) \wedge \varphi''(z,y)\right).$$

The last line holds because $\mathbf{1}(x,y) \in \Phi$ where $\mathbf{1}(x,y) \equiv$ **true** for all $(x,y)$.

**SEAL (MPNN).**  The layers are given by

$$\chi^{(l+1)}(x,z,y) = \mathrm{COM}\left(\chi^{(l)}(x,z,y), \mathrm{AGG}\left(\left\{\!\!\left\{\chi^{(l)}(x,w,y) \mid w \in \mathcal{N}(z)\right\}\!\!\right\}\right)\right),$$
$$\chi^{(l+1)}(x,y) = \mathrm{AGG}\left(\left\{\!\!\left\{\chi^{(l+1)}(x,z,y) \mid z \in \mathcal{N}\right\}\!\!\right\}\right).$$

Therefore the equivalent logic sets are given by

$$\varphi(x,y) := \exists^{\geq N} z \left(\psi(x,z,y)\right),$$
$$\psi(x,z,y) := \exists^{\geq N} w \left(\psi(x,w,y) \wedge E(w,z)\right).$$

**2-GNN**

$$\chi^{(l+1)}(x,y) = \mathrm{COM}\left(\chi^{(l)}(x,y), \mathrm{AGG}\left(\left\{\!\!\left\{\chi^{(l)}(z,y) \mid z \in \mathcal{V}\right\}\!\!\right\}\right), \mathrm{AGG}\left(\left\{\!\!\left\{\chi^{(l)}(x,z) \mid z \in \mathcal{V}\right\}\!\!\right\}\right)\right)$$
$$\Rightarrow \varphi(x,y) := \varphi'(x,y) \mid \exists^{\geq N} z \left(\varphi'(x,z)\right) \mid \exists^{\geq N} z \left(\varphi'(z,y)\right)$$
$$\Rightarrow \varphi(x,y) := \exists^{\geq N} z \left(\varphi'(x,z)\right) \mid \exists^{\geq N} z \left(\varphi'(z,y)\right).$$

$$\square$$

## G.7 PROOF OF PROPOSITION 9

**Proposition 9.** *Suppose the equivalent logic set of a class of GNN models is $\Phi$. Then, the expressive power of the GNN models is bounded by $k$-WL,* iff *the number of variables of the logic formulas in $\Phi$ is at most $k$.*

*Proof.* Given any graphs $G, H$, Cai et al. (1992) states that the following statements are equivalent:

- $k$-WL distinguishes $G, H$;

- There is a $\text{FOC}_k$ formula that distinguishes $G, H$.

Recall that $\text{FOC}_k$ is a subset of first-order formula that allows quantifiers $\exists^{\geq N}$ but restricts the formulas to only possess $k$. Obviously, $\Phi$ in Proposition 9 is a subset of $\text{FOC}_k$, thus the expressive power of GNNs is bounded by $k$-WL. $\qquad\square$

## G.8 PROOF OF COROLLARY 10

**Corollary 10.** *The expressivity of GNN models satisfies: MPNNs = 1-WL < Subgraph GNNs (weak) = NBFNet < Subgraph GNNs (strong) < Local 2-FGNN < 2-FGNN = 3-WL, 1-WL < SEAL < 4-WL.*

*Proof.* By utilizing Proposition 5, Proposition 6, Proposition 9 and the known results (Xu et al., 2018; Qian et al., 2022; Huang et al., 2024; Cai et al., 1992), obviously Corollary 10 holds. $\quad\square$

## G.9 PROOF OF LEMMA 13

**Lemma 13.** *Let $\Phi_1$ be a set of logic formulas which is fully expressed by $\chi_1$. Let $\Phi$ be defined by*

$$\varphi(\mathbf{u}) := \exists^{\geq n}\mathbf{v}\left(\varphi_1(\mathbf{v}) \wedge \mathbf{1}_{\mathbf{v}\in\mathcal{N}(\mathbf{u})}\right) \mid \neg\varphi'(\mathbf{u}) \mid \varphi'(\mathbf{u}) \wedge \varphi''(\mathbf{u}),$$

*where $\varphi, \varphi', \varphi'' \in \Phi$, and we restrict $n < N$ where $N$ is an arbitrary positive integer. Then, there exists an aggregation function AGG such that $\chi(\mathbf{u}) = \text{AGG}(\{\{\chi_1(\mathbf{v}) \mid \mathbf{v} \in \mathcal{N}(\mathbf{u})\}\})$ fully expresses $\Phi$.*

*Proof.* We prove by manually constructing the AGG function. For each $\varphi \in \Phi$, let $\{\varphi^1, ..., \varphi^L\}$ where $\varphi^l \in \Phi$ for $l \in [L]$ be the series of sub-formulas of $\varphi$ such that if $\varphi^p$ is a sub-formula of $\varphi^q$ then $p < q$. Also, we denote $\chi[\varphi]$ to be the dimension of $\chi$ that corresponds to the formula $\varphi \in \Phi$. We prove by induction on the sub-formula series of $\varphi$.
1) At beginning for $\varphi^1$ we only have $\varphi^1(\mathbf{u}) := \exists^{\geq N}\mathbf{v}\left(\varphi_1(\mathbf{v}) \wedge \mathbf{1}_{\mathbf{v}\in\mathcal{N}(\mathbf{u})}\right)$. In this case we let

$$\chi(\mathbf{u})[\varphi^1] = \text{AGG}^1\left(\{\{\chi_1(\mathbf{v}) \mid \mathbf{v} \in \mathcal{N}(\mathbf{u})\}\}\right) = \mathbf{1}\{\text{There are no less than } N \text{ } \mathbf{v} \in \mathcal{N}(\mathbf{u}) \text{ such that } \chi_1(\mathbf{v})[\varphi_1] = \textbf{true}\}.$$

2) Suppose at iteration $l$, all $\varphi^p(\mathbf{u})$ for $p < l$ can be captured by some $\chi(\mathbf{u}) = \text{AGG}^p\left(\{\{\chi_1(\mathbf{v}) \mid \mathbf{v} \in \mathcal{N}(\mathbf{u})\}\}\right)$. We now need to show that $\varphi^l$ can also be captured by $\chi$. We show this by designing specific $\text{AGG}^l$ function there is also $\chi^l(\mathbf{u})[\varphi^l] = \text{AGG}^l\left(\{\{\chi_1(\mathbf{v}) \mid \mathbf{v} \in \mathcal{N}(\mathbf{u})\}\}\right)$ that captures $\varphi^l(\mathbf{u})$. It is also straightforward to prove: If $\varphi^l(\mathbf{u}) := \neg\varphi^q(\mathbf{u})$ for some $q \in [l-1]$, then

$$\text{AGG}^l\left(\{\{\chi_1(\mathbf{v}) \mid \mathbf{v} \in \mathcal{N}(\mathbf{u})\}\}\right) = \mathbf{1}\{\text{AGG}^q\left(\{\{\chi_1(\mathbf{v}) \mid \mathbf{v} \in \mathcal{N}(\mathbf{u})\}\}\right) = \textbf{false}\}.$$

If $\varphi^l(\mathbf{u}) := \varphi^p(\mathbf{u}) \wedge \varphi^q(\mathbf{u})$ then

$$\text{AGG}^l\left(\{\{\chi_1(\mathbf{v}) \mid \mathbf{v} \in \mathcal{N}(\mathbf{u})\}\}\right) = \mathbf{1}\{\text{AGG}^p\left(\{\{\chi_1(\mathbf{v}) \mid \mathbf{v} \in \mathcal{N}(\mathbf{u})\}\}\right) = \text{AGG}^q\left(\{\{\chi_1(\mathbf{v}) \mid \mathbf{v} \in \mathcal{N}(\mathbf{u})\}\}\right) = \textbf{true}\}.$$

Otherwise $\varphi^l(\mathbf{u}) := \exists^{\geq N}\mathbf{v}\left(\varphi^q(\mathbf{v}) \mid \mathbf{v} \in \mathcal{N}(\mathbf{u})\right)$ which is already proved in 1). $\qquad\square$

## G.10 PROOF OF LEMMA 14

**Lemma 14.** *Let $\Phi_1, \Phi_2$ be a set of logic formulas which is fully expressed by $\chi_1, \chi_2$. Let $\Phi$ be defined by*

$$\varphi(\mathbf{u}) := \varphi_1(\mathbf{u}) \mid \varphi_2(\mathbf{u}) \mid \neg\varphi_2'(\mathbf{u}) \mid \varphi_2'(\mathbf{u}) \wedge \varphi_2''(\mathbf{u}),$$

*where $\varphi, \varphi', \varphi'' \in \Phi$. Then, there exists a combination function* COM *such that $\chi(\mathbf{u}) = $* COM$(\chi_1(\mathbf{u}), \chi_2(\mathbf{u}))$ *fully expresses $\Phi$.*

*Proof.* Let $\{\varphi^1, ..., \varphi^L\}$ be the series of sub-formulas of $\varphi$ such that if $\varphi^p$ is a sub-formula of $\varphi_q$ then $p < q$. We prove by induction on the sub-formula series of $\Phi$.
1) At beginning for $\varphi^1$ we only have $\varphi^1(\mathbf{u}) := \varphi_1(\mathbf{u})$ or $\varphi(\mathbf{u}) := \varphi_2(\mathbf{u})$. In this case we let $\chi(\mathbf{u}) := \chi_1(\mathbf{u})$ or $\chi(\mathbf{u}) := \chi_2(\mathbf{u})$. Since $\Phi_1, \Phi_2$ are the equivalent sets of $\chi_1, \chi_2$ respectively, $\varphi^1$ is captured by $\chi$.
2) Suppose at iteration $l$, all $\varphi^p(\mathbf{u})$ for $p < l$ can be captured by some $\chi^p(\mathbf{u}) = $ COM$^p(\chi_1(\mathbf{u}), \chi_2(\mathbf{u}))$. We show that by designing specific COM$^l$ function there is also $\chi^l(\mathbf{u}) = $ COM$^l(\chi_1(\mathbf{u}), \chi_2(\mathbf{u}))$ that captures $\varphi^l(\mathbf{u})$. It is straightforward to prove: If $\varphi^l(\mathbf{u}) = \neg\varphi^q(\mathbf{u})$, then

$$\text{COM}^l(\chi_1(\mathbf{u}), \chi_2(\mathbf{u})) = \mathbf{1}\{\text{COM}^p(\chi_1(\mathbf{u}), \chi_2(\mathbf{u})) = \textbf{false}\}.$$

If $\varphi^l(\mathbf{u}) = \varphi^p(\mathbf{u}) \wedge \varphi^q(\mathbf{u})$ then

$$\text{COM}^l(\chi_1(\mathbf{u}), \chi_2(\mathbf{u})) = \mathbf{1}\{\text{COM}^p(\chi_1(\mathbf{u}), \chi_2(\mathbf{u})) = \text{COM}^q(\chi_1(\mathbf{u}), \chi_2(\mathbf{u})) = \textbf{true}\}.$$

Otherwise we have $\varphi^l(\mathbf{u}) = \varphi_1(\mathbf{u})$ or $\varphi^l(\mathbf{u}) = \varphi_2(\mathbf{u})$. In this case $\varphi^l$ can also be captured by $\chi$ as proven in 1). $\square$

## G.11 PROOF OF LEMMA 16

**Lemma 16.** *Given a series of functions $\{\chi_1, ..., \chi_K\}$ where each function $\chi_i$ maps a node tuple to a discrete color and is defined by either $\chi_i(\mathbf{u}) = \text{AGG}_i(\{\{\chi_j(\mathbf{v}) \mid \mathbf{v} \in \mathcal{N}_i(\mathbf{u})\}\}), \chi_i(\mathbf{u}) = \text{COM}_i(\chi_j(\mathbf{u}), \chi_k(\mathbf{u}))$ or $\chi_i(\mathbf{u}) = \text{INIT}_i(\mathbf{u})$. Let $\mathcal{X}_i$ be the set of all possible $\chi_i$ defined in the above manner for $i \in [K]$. Let $\Phi_i$ be the set specified by Theorem 3 of $\mathcal{X}_i$. Then, for arbitrary $i \in [K]$ and each possible color* **col** *of $\chi_i$, there exists a logic formula $\varphi_i^{\textbf{col}} \in \Phi_i$ satisfying:*

$$\varphi_i^{\textbf{col}}(\mathbf{u}) = \textbf{true} \iff \chi_i(\mathbf{u}) = \textbf{col}$$

*for arbitrary graph $G$ with no more than $N$ nodes and $\mathbf{u} \in \mathcal{V}_G^k$.*

*Proof.* For arbitrary $\chi_i$ and $\Phi_i$, if the statement in Lemma 16 holds, i.e. for each possible color **col** of $\chi_i$, there exists a logic formula $\varphi_i^{\textbf{col}} \in \Phi_i$ satisfying:

$$\varphi_i^{\textbf{col}}(\mathbf{u}) = \textbf{true} \iff \chi_i(\mathbf{u}) = \textbf{col}$$

for arbitrary graph $G$ with no more than $N$ nodes and $\mathbf{u} \in \mathcal{V}_G^k$, we say $\Phi_i$ $N$-captures $\chi_i$, and $\chi_i$ is $N$-captured by $\Phi_i$. We present the following lemmas.

**Lemma 21.** *Let $\chi_1$ be a function that maps a node tuple to a discrete color, which is $N$-captured by $\Phi_1$. Let $\chi(\mathbf{u}) = \text{AGG}(\{\{\chi_1(\mathbf{v}) \mid \mathbf{v} \in \mathcal{N}(\mathbf{u})\}\})$ where AGG is some aggregation function. Then by defining $\Phi$ as below:*

$$\varphi(\mathbf{u}) := \exists^{\geq n}\mathbf{v}\left(\varphi_1(\mathbf{v}) \wedge \mathbf{1}_{\mathbf{v} \in \mathcal{N}(\mathbf{u})}\right) \mid \neg\varphi'(\mathbf{u}) \mid \varphi'(\mathbf{u}) \wedge \varphi''(\mathbf{u}),$$

*where $\varphi, \varphi', \varphi'' \in \Phi$. $\Phi$ $N$-captures $\chi$.*

**Lemma 22.** *Let $\chi_1, \chi_2$ be functions that map a node tuple to a discrete color, which are $N$-captured by $\Phi_1, \Phi_2$ respectively. Let $\chi(\mathbf{u}) = \text{COM}(\{\{\chi_1(\mathbf{u}), \chi_2(\mathbf{u})\}\})$ where COM is some combination function. Then by defining $\Phi$ as below:*

$$\varphi(\mathbf{u}) := \varphi_1(\mathbf{u}) \mid \varphi_2(\mathbf{u}) \mid \neg\varphi_2'(\mathbf{u}) \mid \varphi_2'(\mathbf{u}) \wedge \varphi_2''(\mathbf{u}),$$

*where $\varphi, \varphi', \varphi'' \in \Phi$. $\Phi$ $N$-captures $\chi$.*

We are now ready to prove Lemma 16. At beginning for $\chi_1(\mathbf{u}) = \text{INIT}(\mathbf{u})$, we only have

$$\varphi_1(\mathbf{u}) := \textbf{atp}(\mathbf{u}) \mid \neg\varphi_1'(\mathbf{u}) \mid \varphi_1'(\mathbf{u}) \wedge \varphi_1''(\mathbf{u}),$$

where $\varphi_1, \varphi_1', \varphi_1'' \in \Phi_1$. It is evident that $\chi_1$ is $N$-captured by $\Phi_1$ and $\Phi_1$ is finite. Suppose at iteration $l$, $\chi_p$ is $N$-captured by $\Phi_p$ for all $p < l$. By directly utilizing Lemma 21, 21, it is obvious that $\Phi_l$ $N$-captures $\varphi_l$. Therefore the proof completes. $\square$

## G.12 PROOF OF LEMMA 21

**Lemma 21.** *Let $\chi_1$ be a function that maps a node tuple to a discrete color, which is $N$-captured by $\Phi_1$. Let $\chi(\mathbf{u}) = \mathrm{AGG}(\{\{\chi_1(\mathbf{v}) \mid \mathbf{v} \in \mathcal{N}(\mathbf{u})\}\})$ where $\mathrm{AGG}$ is some aggregation function. Then by defining $\Phi$ as below:*

$$\varphi(\mathbf{u}) := \exists^{\geq n} \mathbf{v} \left( \varphi_1(\mathbf{v}) \wedge \mathbf{1}_{\mathbf{v} \in \mathcal{N}(\mathbf{u})} \right) \mid \neg \varphi'(\mathbf{u}) \mid \varphi'(\mathbf{u}) \wedge \varphi''(\mathbf{u}),$$

*where $\varphi, \varphi', \varphi'' \in \Phi$. $\Phi$ $N$-captures $\chi$.*

*Proof.* For each possible color **col** of $\chi$, below we construct a logic formula $\varphi^{\mathbf{col}}$ satisfying $\varphi^{\mathbf{col}}(\mathbf{u}) = \mathbf{true} \iff \chi(\mathbf{u}) = \mathbf{col}$ for all graph $G$ with no more than $N$ nodes and $\mathbf{u} \in \mathcal{V}_G^k$ and show that $\varphi^{\mathbf{col}}$ is in $\Phi$.

For notation brevity we use $\mathbf{col}_1$ to refer to colors of $\chi_1$ and $\mathbf{col}$ to refer to colors of $\chi$. We first create an intermediate function $\chi'$ which is defined by

$$\chi'(\mathbf{u}) = \mathrm{hash}(\{\{\chi_1(\mathbf{v}) \mid \mathbf{v} \in \mathcal{N}(\mathbf{u})\}\}), \tag{10}$$

where $\mathrm{hash}$ is an injective hashing function that maps different inputs to different colors. Therefore, $\chi$ can be written in the form of: $\chi(\mathbf{u}) = f(\chi'(\mathbf{u}))$ with some function $f$. We first show that $\chi'$ is $N$-captured by $\Phi$ by constructing a $\varphi^{\mathbf{col}'}$ for each color $\mathbf{col}'$ of $\chi'$, then show that $\chi$ is also $N$-captured by $\Phi$.

First, recall that two multisets are different $\{\{\chi_1(\mathbf{w}) \mid \mathbf{w} \in \mathcal{N}(\mathbf{u})\}\} \neq \{\{\chi_1(\mathbf{w}) \mid \mathbf{w} \in \mathcal{N}(\mathbf{v})\}\}$ if there exists a color $\mathbf{col}_1$ of $\chi_1$ such that

$$\left| \{ \mathbf{w} \mid \mathbf{w} \in \mathcal{N}(\mathbf{u}), \chi_1(\mathbf{w}) = \mathbf{col}_1 \} \right| \neq \left| \{ \mathbf{w} \mid \mathbf{w} \in \mathcal{N}(\mathbf{v}), \chi_1(\mathbf{w}) = \mathbf{col}_1 \} \right|.$$

Therefore, $\chi'$ maps two tuples $\mathbf{u}, \mathbf{v}$ to different colors if there exists a color $\mathbf{col}_1$ of $\chi_1$ that the above equation holds. Note that since we only consider graphs with no more than $N$ nodes, $\chi_1$ has finite colors. It is evident that we can rewrite Eq. 11 into:

$$\chi'(\mathbf{u}) = \mathrm{hash}'( \left| \{ \mathbf{v} \mid \mathbf{v} \in \mathcal{N}(\mathbf{u}), \chi_1(\mathbf{v}) = \mathbf{col}_1^1 \} \right|,$$
$$\left| \{ \mathbf{v} \mid \mathbf{v} \in \mathcal{N}(\mathbf{u}), \chi_1(\mathbf{v}) = \mathbf{col}_1^2 \} \right|,$$
$$...$$
$$\left| \{ \mathbf{v} \mid \mathbf{v} \in \mathcal{N}(\mathbf{u}), \chi_1(\mathbf{v}) = \mathbf{col}_1^c \} \right|),$$

where $\mathbf{col}_1^1, ..., \mathbf{col}_1^c$ enumerates through all colors of $\chi_1$, and $\mathrm{hash}'$ is a perfect hashing function. We now show that the above definition of $\chi'$ can be fully reproduced using logic formulas.

Denote by $k$ the order of $\chi_1$. If we only consider graphs with no more than $N$ nodes, then it is obvious that $\mathrm{hash}'$ is a function with domain $[N^k] \times [N^k] \times ... \times [N^k]$ (repeated for $c$ times), which is finite. For each color $\mathbf{col}'$ of $\chi'$, we now construct the logic formula $\varphi^{\mathbf{col}'}$. Suppose the color $\mathbf{col}'$ is corresponded to the case below:

$$\left| \{ \mathbf{v} \mid \mathbf{v} \in \mathcal{N}(\mathbf{u}), \chi_1(\mathbf{v}) = \mathbf{col}_1^1 \} \right| = n_1,$$
$$\left| \{ \mathbf{v} \mid \mathbf{v} \in \mathcal{N}(\mathbf{u}), \chi_1(\mathbf{v}) = \mathbf{col}_1^2 \} \right| = n_2,$$
$$...$$
$$\left| \{ \mathbf{v} \mid \mathbf{v} \in \mathcal{N}(\mathbf{u}), \chi_1(\mathbf{v}) = \mathbf{col}_1^c \} \right| = n_c.$$

Then we define

$$\varphi^{\mathbf{col}'}(\mathbf{u}) := \left( \exists^{=n_1} \mathbf{v} \left( \varphi_1^{\mathbf{col}_1^1}(\mathbf{v}) \wedge \mathbf{1}_{\mathbf{v} \in \mathcal{N}(\mathbf{u})} \right) \right)$$
$$\wedge \left( \exists^{=n_2} \mathbf{v} \left( \varphi_1^{\mathbf{col}_1^2}(\mathbf{v}) \wedge \mathbf{1}_{\mathbf{v} \in \mathcal{N}(\mathbf{u})} \right) \right)$$
$$\wedge...$$
$$\wedge \left( \exists^{=n_c} \mathbf{v} \left( \varphi_1^{\mathbf{col}_1^c}(\mathbf{v}) \wedge \mathbf{1}_{\mathbf{v} \in \mathcal{N}(\mathbf{u})} \right) \right).$$

Therefore, $\varphi^{\mathbf{col}'}(\mathbf{u}) = \mathbf{true}$ *iff* $\chi'(\mathbf{u}) = \mathbf{col}'$. Also, $\varphi^{\mathbf{col}'}(\mathbf{u}) \in \Phi$. Next, we prove that for each color $\mathbf{col}$ of $\chi(\mathbf{u}) = f(\chi'(\mathbf{u}))$, there also exists $\varphi^{\mathbf{col}}$ that $\varphi^{\mathbf{col}}(\mathbf{u}) = \mathbf{true}$ *iff* $\chi(\mathbf{u}) = \mathbf{col}$. From the

above discussion, $f$ is a function that maps the colors $\textbf{col}'$ of $\chi'$ (which is finite if we only consider graphs with no more than $N$ nodes) to the colors $\textbf{col}$ of $\chi$ (which is again finite). Obviously, each color $\textbf{col}$ is corresponded to a subset $\mathcal{C}$ of the colors of $\chi'$ where $\chi(\textbf{u}) = \textbf{col}$ *iff* $\chi'(\textbf{u}) \in \mathcal{C}$. We define

$$\varphi^{\textbf{col}}(\textbf{u}) := \bigvee_{\textbf{col}' \in \mathcal{C}} \varphi^{\textbf{col}'}(\textbf{u}).$$

It is now evident that for all graphs with no more than $N$ nodes, $\varphi^{\textbf{col}}(\textbf{u}) = \textbf{true} \iff \chi(\textbf{u}) = \textbf{col}$. Also $\varphi^{\textbf{col}}(\textbf{u})$ is in $\Phi$. Thus the proof completes. $\qquad\square$

### G.13 PROOF OF LEMMA 22

**Lemma 22.** *Let $\chi_1, \chi_2$ be functions that map a node tuple to a discrete color, which are $N$-captured by $\Phi_1, \Phi_2$ respectively. Let $\chi(\textbf{u}) = \mathrm{COM}(\{\{\chi_1(\textbf{u}), \chi_2(\textbf{u})\}\})$ where $\mathrm{COM}$ is some combination function. Then by defining $\Phi$ as below:*

$$\varphi(\textbf{u}) := \varphi_1(\textbf{u}) \mid \varphi_2(\textbf{u}) \mid \neg\varphi_2'(\textbf{u}) \mid \varphi_2'(\textbf{u}) \wedge \varphi_2''(\textbf{u}),$$

*where $\varphi, \varphi', \varphi'' \in \Phi$. $\Phi$ $N$-captures $\chi$.*

*Proof.* The proof is similar to the proof of Lemma 21. For each possible color $\textbf{col}$ of $\chi$, below we construct a logic formula $\varphi^{\textbf{col}}$ satisfying $\varphi^{\textbf{col}}(\textbf{u}) = \textbf{true} \iff \chi(\textbf{u}) = \textbf{col}$ for all graph $G$ with no more than $N$ nodes and $\textbf{u} \in \mathcal{V}_G^k$ and show that $\varphi^{\textbf{col}}$ is in $\Phi$.

For notation brevity we use $\textbf{col}_1, \textbf{col}_2$ to refer to colors of $\chi_1, \chi_2$ respectively and use $\textbf{col}$ to refer to colors of $\chi$. We first create an intermediate function $\chi'$ which is defined by

$$\chi'(\textbf{u}) = \mathrm{hash}(\chi_1(\textbf{u}), \chi_2(\textbf{u})), \tag{11}$$

where $\mathrm{hash}$ is a perfect hashing function. Since we only consider graphs with no more than $N$ nodes, the colors of $\chi_1$ and $\chi_2$ are finite. We denote by $\mathcal{C}_1, \mathcal{C}_2$ the set of colors of $\chi_1, \chi_2$ respectively. The domain of $\mathrm{hash}$ is thus $\mathcal{C}_1 \times \mathcal{C}_2$. Suppose a color $\textbf{col}'$ of $\chi'$ is defined by $\textbf{col}' = \mathrm{hash}(\textbf{col}_1, \textbf{col}_2)$ where $\chi_1, \chi_2$ are some colors of $\chi_1, \chi_2$ respectively. Then, by letting

$$\varphi^{\textbf{col}'}(\textbf{u}) := \varphi_1^{\textbf{col}_1}(\textbf{u}) \wedge \varphi_2^{\textbf{col}_2}(\textbf{u}),$$

we have $\varphi^{\textbf{col}'}(\textbf{u}) = \textbf{true} \iff \chi'(\textbf{u}) = \textbf{col}'$ for graphs with no more than $N$ nodes. Also, obviously $\varphi^{\textbf{col}'}(\textbf{u})$ is in $\Phi$. We next show that for $\chi$ which can be expressed by $\chi(\textbf{u}) = f(\chi'(\textbf{u}))$ for some $f$, there still exists $\varphi^{\textbf{col}}$ for each color $\textbf{col}$ of $\chi$ such that $\chi(\textbf{u}) = \textbf{col} \iff \varphi^{\textbf{col}}(\textbf{u})$. This proof is exactly the same as in the proof of Lemma 21. Therefore, the proof completes. $\qquad\square$

### G.14 PROOF OF LEMMA 15

**Lemma 15.** *Given arbitrary $\mathcal{X}$, let $\Phi$ be constructed as in Theorem 3. Then, all logic formulas captured by a $\chi \in \mathcal{X}$ is in $\Phi$.*

*Proof.* We first define the concept of finite aggregation functions.

**Definition 23.** Recall that an aggregation function is a mapping from a multiset of colors to a color:

$$\mathrm{AGG}(\{\{\chi(\textbf{u}) \mid \textbf{u} \in \mathcal{N}(\textbf{v})\}\}) = \textbf{col},$$

where $\textbf{col}$ stands for the output color. From Appendix G.12 it is obvious that if the number of colors of $\chi$ is finite, we can write $\mathrm{AGG}$ equivalently in the form of

$$\mathrm{AGG}(\{\{\chi(\textbf{u}) \mid \textbf{u} \in \mathcal{N}(\textbf{v})\}\}) = f(\,\left|\left\{\textbf{v} \mid \textbf{v} \in \mathcal{N}(\textbf{u}), \chi(\textbf{v}) = \textbf{col}^1\right\}\right|,$$
$$\left|\left\{\textbf{v} \mid \textbf{v} \in \mathcal{N}(\textbf{u}), \chi(\textbf{v}) = \textbf{col}^2\right\}\right|,$$
$$...$$
$$\left|\left\{\textbf{v} \mid \textbf{v} \in \mathcal{N}(\textbf{u}), \chi(\textbf{v}) = \textbf{col}^c\right\}\right|),$$

where $\textbf{col}^1, ...\textbf{col}^c$ enumerate through all possible colors of $\chi$. It is obvious that $f : \mathbb{N} \times \mathbb{N} \times ... \times \mathbb{N} \to \mathcal{C}$ where $\mathcal{C}$ denotes the set of output colors. We say such an aggregation function $\mathrm{AGG}$ is finite, if the number of possible colors of $\chi$ is finite, and there exists a positive integer $N$ such that for all $l \in [c]$,

$$f(n_1, ..., n_l, ..., n_c) = f(n_1, ..., n_l + 1, ..., n_c) \text{ if } n_l > N.$$

In this case we say the aggregation function is bounded by $N$. The above definition implies:

- If the input colors of AGG are finite, then the output colors are also finite;

- $f(N_1, ..., N_c)$ is constant for sufficiently large $N_1, ..., N_c$.

Intuitively, finite aggregation functions does not distinguish the situations where there are more than $N$ items of each color. Extending from the definition of finite aggregation functions, we define the concept of finite $\chi \in \mathcal{X}$.

**Definition 24.** Given arbitrary $\mathcal{X}$ as defined in Theorem 3. We say a $\chi \in \mathcal{X}$ is finite, if all aggregation functions used for constructing $\chi$ are finite.

Note that, if $\chi$ is finite, the possible colors of $\chi$ are also finite (which is shown in Appendix G.12). We are now ready to prove the lemma.

We prove Lemma 15 by noticing the following fact.

**Lemma 25.** *If a FOC formula $\psi$ is expressed by some $\chi \in \mathcal{X}$, it is expressed by a finite $\chi \in \mathcal{X}$.*

Lemma 25 is intuitive, as given a specific formula $\psi$, the maximum counting number $N$ of the quantifiers $\exists^N$ is determined, and thus for each term in the form of $\exists^{\geq N} \mathbf{v}(\psi'(\mathbf{v}))$, $\psi$ also does not distinguish the situations where there are more than $N\mathbf{v}$ that satisfies $\psi'(\mathbf{v})$. Note that in Appendix G.12 we have provided a method for constructing a logic formula $\varphi \in \Phi$ that expresses such finite $\chi \in \mathcal{X}$. Hence, if a first-order logic formula $\psi$ is expressed by some $\chi \in \mathcal{X}$, it is expressed by a finite $\chi \in \mathcal{X}$, which is further expressed by a logic formula $\varphi \in \Phi$. This implies that $\psi \in \Phi$, thus concluding the proof. $\qquad\square$

### G.15 PROOF OF LEMMA 25

**Lemma 25.** *If a FOC formula $\psi$ is expressed by some $\chi \in \mathcal{X}$, it is expressed by a finite $\chi \in \mathcal{X}$.*

*Proof.* The proof is a direct generalization of Otto (2019) as follows.

**Lemma 26.** *(**Lemma 2.5 in Otto (2019)**). For $\varphi$ in first-order logic that is invariant under $\sim_\#$ over the class of all (or just all finite) pointed Kripke structures, there are $c, l \in \mathbb{N}$ s.t. $\varphi(x)$ is invariant under $\sim_\#^{c,l}$ over the class of all (or just all finite) pointed Kripke structures.*

In the above lemma, for two nodes $x, y$ in graphs, $x \sim_\# y$ indicates that two nodes $x, y$ are not distinguished by MPNNs (1-WL). $x \sim_\#^{c,l} y$ indicates that two nodes $x, y$ are not distinguished by MPNNs (1-WL) with $l$ layers and aggregation functions bounded by $c$. We generalize the above lemma to our setting as follows.

Suppose a logic formula $\psi$ is captured by some $\chi \in \mathcal{X}$, as stated in Lemma 25. As convention, suppose $\chi$ is defined by a series of functions $\{\chi_1, ..., \chi_K\}$ where we denote $\chi_K = \chi$ and each function $\chi_i$ maps a node tuple to a discrete color and is defined by either $\chi_i(\mathbf{u}) = \mathrm{AGG}_i(\{\{\chi_j(\mathbf{v}) \mid \mathbf{v} \in \mathcal{N}_i(\mathbf{u})\}\})$, $\chi_i(\mathbf{u}) = \mathrm{COM}_i(\chi_j(\mathbf{u}), \chi_k(\mathbf{u}))$ or $\chi_i(\mathbf{u}) = \mathrm{INIT}_i(\mathbf{u})$. We denote $\mathcal{X}_1, ..., \mathcal{X}_K$ to be the set of all $\chi_1, ..., \chi_K$. For each generalized neighbor $\mathcal{N}_i$, we denote its corresponding predicate $\mathbf{1}_{\mathbf{v} \in \mathcal{N}_i(\mathbf{u})}$ as $E_{\mathcal{N}_i}$ where $E_{\mathcal{N}_i}(\mathbf{v}, \mathbf{u})$ is **true** *iff* $\mathbf{v} \in \mathcal{N}_i(\mathbf{u})$. Given a graph $G$, we define the unrolling of $G$ by $\mathcal{X}$ at $\mathbf{u}$, denoted as $F = \mathrm{Unr}_\mathcal{X}(G, \mathbf{u})$, to be a structure for first-order logic described as follows.

- Denote **order**$(\mathcal{X}_i)$ to be the order of $\chi_i \in \mathcal{X}_i$. $F$ has a tuple of nodes $\boldsymbol{u}$ for $\mathbf{u}$.

- Repeat for $K$ times:

  - For each node tuple $\boldsymbol{u}$ existed in $F$ and each generalized neighbor $\mathcal{N}_i$ in $F$, suppose $\mathbf{v} \in \mathcal{N}_i(\mathbf{u})$ in the original graph, then add a tuple of nodes $\boldsymbol{u}$ in $F$ and let $E_{\mathcal{N}_i}(\boldsymbol{v}, \boldsymbol{u}) = $ **true** if $\boldsymbol{u}$ does not exists in $F$.

The above procedure generalize the rooted unfolding tree $T$ of graphs $G$ at $x$ in that:

- In the rooted unfolding tree $T$ two nodes are connected by edges $E$ if the corresponding nodes in $G$ are neighborhood. $T$ is a tree with root $x$.

- In the unrolling $F$ two node tuples are connected by generalized edges $E_{\mathcal{N}_i}$ if the corresponding node tuples in $G$ are generalized neighborhood $\mathcal{N}_i$. $F$ is a tree if we only consider the generalized edges $E_{\mathcal{N}_i}$ and consider each node tuple as a ensemble, with root $\mathbf{u}$.

Next, our proof step is similar to Otto (2019). For two node tuples $\mathbf{u}, \mathbf{v}$, we denote $\mathbf{u} \sim_{\mathcal{X}} \mathbf{v}$ if $\mathbf{u}$ and $\mathbf{v}$ are not separated by any $\chi \in \mathcal{X}$. First, we would like to mention that for any graphs $G, H$ and $\mathbf{u} \in \mathcal{V}_G^k, \mathbf{v} \in \mathcal{V}_H^k$, $\mathbf{u} \sim_{\mathcal{X}} \mathbf{v} \iff \boldsymbol{u} \sim_{\mathcal{X}} \boldsymbol{v}$ where $\boldsymbol{u}, \boldsymbol{v}$ are tuples in $\mathrm{Unr}_{\mathcal{X}}(G, \mathbf{u}), \mathrm{Unr}_{\mathcal{X}}(G, \mathbf{v})$ corresponding to $\mathbf{u}, \mathbf{v}$ respectively. It is straightforward: since the evaluation steps of any $\chi \in \mathcal{X}$ on $\mathbf{u} \in \mathcal{V}_G^k$ are exactly the same as $\chi$ applied on $\boldsymbol{u}$ in $\mathrm{Unr}_{\mathcal{X}}(G, \mathbf{u})$.

Next, we show that any first-order logic formula $\psi$ with quantifier rank $q$ that is invariant under $\sim_{\mathcal{X}}$, is generalized $l$-local for some $l = f(q)$ in restriction to all unrolling of graphs that are generalized tree (as mentioned above) to depth $l$, where we define a function is generalized $l$-local if it only considers node tuples up to distance $l$ by generalized edges. This corresponds to Lemma 2.3 in Otto (2019). The prove is straightforward: since $\chi \in \mathcal{X}$ itself is $l$-local, $\psi$ must also be $l$-local. Also, each unrolling is naturally $K$-local, since it is a tree with depth $K$.

Next, we need the following fact that for any $q \in \mathbb{N}$, there is a $c \in \mathbb{N}$ such that for any unrolling of graphs $F_1 = \mathrm{Unr}_{\mathcal{X}}(G, \mathbf{u}), F_2 = \mathrm{Unr}_{\mathcal{X}}(H, \mathbf{v})$:

$$\boldsymbol{u} \sim_{\mathcal{X}}^c \boldsymbol{v} \Rightarrow \phi(\boldsymbol{u}) = \phi(\boldsymbol{v}) \text{ for all first-order logic formula } \phi \text{ with variables no more than } q.$$

The above fact is a direct generalization of Lemma 2.4 in Otto (2019). Since $\phi$ consists of no more than $q$ variables, it can only at most consider $q$ distinct nodes and thus $c = q$ is enough for $\mathcal{X}$ to capture them. Given $\psi$ captured by $\chi \in \mathcal{X}$, we now show that $\psi$ is captured by a finite $\chi \in \mathcal{X}$ bounded by some $N$. This is equivalent to showing that, for any graphs $G, H$ and $\mathbf{u} \in \mathcal{V}_G^k, \mathbf{v} \in \mathcal{V}_H^k$ that are not separated by any finite $\chi \in \mathcal{X}$ bounded by $N$, $\psi(\mathbf{u}) = \psi(\mathbf{v})$. Without loss of generality we assume $\psi(\mathbf{u}) = \textbf{true}$. Let $\boldsymbol{u}, \boldsymbol{v}$ be the tuples in $\mathrm{Unr}_{\mathcal{X}}(G, \mathbf{u}), \mathrm{Unr}_{\mathcal{X}}(H, \mathbf{v})$ corresponding to $\mathbf{u}, \mathbf{v}$ respectively. Then we have:

$$
\begin{aligned}
\psi(\mathbf{u}) = \textbf{true} &\iff \psi(\boldsymbol{u}) = \textbf{true} && \text{(by } \mathbf{u} \sim_{\mathcal{X}} \mathbf{v} \iff \boldsymbol{u} \sim_{\mathcal{X}} \boldsymbol{v}) \\
&\iff \psi(\boldsymbol{v}) = \textbf{true} && \text{(by } l\text{-locality implied in } \mathrm{Unr}_{\mathcal{X}}(G, \mathbf{u}) \text{ and } \mathrm{Unr}_{\mathcal{X}}(H, \mathbf{v})) \\
&\iff \psi(\mathbf{v}) = \textbf{true} && \text{(by } \mathbf{u} \sim_{\mathcal{X}} \mathbf{v} \iff \boldsymbol{u} \sim_{\mathcal{X}} \boldsymbol{v}).
\end{aligned}
$$

$\square$

## G.16 PROOF OF LEMMA 17

**Lemma 17.** *If $n_k > N$ for some $k \in [K]$, then $\psi_{n_1 n_2 \dots n_K} = \textbf{false}$ on $G$.*

*Proof.* Obviously if $n_k > N$ for some $k \in [K]$, we have

$$
\begin{aligned}
&\textbf{Hom}(F, G) \\
&\geq \prod_{k \in [K]} n_k \\
&> N,
\end{aligned}
$$

which contradicts with the fact that $\textbf{Hom}(F, G) = N$. $\square$

## G.17 PROOF OF LEMMA 18

**Lemma 18.** *For $\psi_{n_1 n_2 \dots n_K}$ and $\psi_{m_1 m_2 \dots m_K}$, if $n_k \neq m_k$ for some $k \in [K]$, then then grounding results of $\psi_{n_1 n_2 \dots n_K}$ and $\psi_{m_1 m_2 \dots m_K}$ are different, i.e. there exists no $\mathbf{V}$ that is both a grounding result of $\psi_{n_1 n_2 \dots n_K}$ and $\psi_{m_1 m_2 \dots m_K}$.*

*Proof.* Since $n_k \neq m_k$ for some $k \in [K]$, we assume that $n_l \neq m_l$ while $n_k = m_k$ for $k \in [l+1, K]$. We prove by contradiction. Suppose $\mathbf{V} = (\mathbf{v}_1, ..., \mathbf{v}_K)$ is both a grounding result of $\psi_{n_1 n_2 ... n_K}$ and $\psi_{m_1 m_2 ... m_K}$. This indicates that for $\psi_{n_1 n_2 ... n_K}$, by fixing its variables to $\mathbf{x}_1 := \mathbf{v}_1, \mathbf{x}_2 := \mathbf{v}_2, ..., \mathbf{x}_{l-1} := \mathbf{v}_{l-1}$, there exists exactly $n_l$ different groundings of $\mathbf{x}_l$ satisfying

$$\exists^{=n_{l+1}} \mathbf{x}_{l+1} \exists^{=n_{l+2}} \mathbf{x}_{l+2} ... \exists^{=n_K} \mathbf{x}_K \left( \bigwedge_{m \in [M]} E(x_{i_m}, x_{j_m}) \right)$$

in $G$. However, for $\psi_{m_1 m_2 ... m_K}$ by fixing its variables to $\mathbf{x}_1 := \mathbf{v}_1, \mathbf{x}_2 := \mathbf{v}_2, ..., \mathbf{x}_{l-1} := \mathbf{v}_{l-1}$, there exists exactly $m_l$ different groundings of $\mathbf{x}_l$ satisfying

$$\exists^{=m_{l+1}} \mathbf{x}_{l+1} \exists^{=m_{l+2}} \mathbf{x}_{l+2} ... \exists^{=m_K} \mathbf{x}_K \left( \bigwedge_{m \in [M]} E(x_{i_m}, x_{j_m}) \right)$$

in $G$. Since $m_k = n_k$ for $k \in [l+1, K]$ and $m_l \neq n_l$, this yields a contradiction. $\square$

### G.18 PROOF OF LEMMA 19

**Lemma 19.** *If there is $\psi \in \Phi$ such that $\psi$ evaluates to different values on $G$ and $H$, then there exists $\varphi$ in the form of $\varphi := \exists^{=n_1} \mathbf{x}_1 \exists^{=n_2} \mathbf{x}_2 ... \exists^{=n_K} \mathbf{x}_K \left( \bigwedge_{p \in [P]} E(x_{i_p}, x_{j_p}) \bigwedge_{q \in [Q]} \neg E(x_{i_q}, x_{j_q}) \right)$ that also evaluates to different values on $G$ and $H$.*

*Proof.* We prove this by constructing $\varphi$ of the form

$$\varphi := \exists^{=n_1} \mathbf{x}_1 ... \left( \bigwedge_{m \in [M]} E(x_{i_m}, x_{j_m}) \right)$$

that explicitly captures the colors of $\chi$. Concretely, similar as Theorem 3, suppose a series of functions $\{\chi_1, ..., \chi_L\}$ where $\chi_l$ is defined by

$$\chi_l(\mathbf{x}) = \text{hash}\left( \chi_p(\mathbf{x}), \chi_q(\mathbf{x}) \right),$$

$$\chi_l(\mathbf{x}) = \text{hash}\left( \{\{\chi_p(\mathbf{y}) \mid y \in \mathcal{N}(\mathbf{x})\}\} \right),$$

or

$$\chi_l(\mathbf{x}) = \mathbf{atp}(\mathbf{x}).$$

The difference between Theorem 3 and here is that we replace AGG and COM functions with injective hash function. Obviously the separation power of $(\chi_l)_{l \in [L]}$ here is no less than that in Theorem 3. The above procedure can be regarded as a general *color refinement* algorithm where the value of $\chi_l(\mathbf{x})$ is called the *color* of $\mathbf{x}$ computed by $\chi_l$. We define the *signature* logic set $\Psi_l$ of $chi_l$ to be the set that satisfiesfor each color $C$, there exists $\psi_C \in \Phi_l$ such that

$$\psi_C(\mathbf{x}) = \textbf{true} \iff \chi(\mathbf{x}) = C.$$

We now provide a method to construct the signature logic set $\Psi_l$. We define:

1.

$$\chi_l(\mathbf{x}) = \text{hash}\left( \chi_p(\mathbf{x}), \chi_q(\mathbf{x}) \right)$$
$$\Rightarrow \psi_l(\mathbf{x}) := \psi_p(\mathbf{x}) \wedge \psi_q(\mathbf{x})$$

2.

$$\chi_l(\mathbf{x}) = \text{hash}\left( \{\{\chi_p(\mathbf{y}) \mid y \in \mathcal{N}(\mathbf{x})\}\} \right)$$
$$\Rightarrow \psi_l(\mathbf{x}) := \exists^{=N} \mathbf{y} \left( \mathbf{1}_{\mathbf{y} \in \mathcal{N}(\mathbf{x})} \right) \wedge \exists^{=N_1} \mathbf{y}_1 \left( \psi_p(\mathbf{y}) \wedge \mathbf{1}_{\mathbf{y}_1 \in \mathcal{N}(\mathbf{x})} \right)$$
$$\wedge \exists^{=N_2} \mathbf{y}_2 \left( \psi_p(\mathbf{y}) \wedge \mathbf{1}_{\mathbf{y}_2 \in \mathcal{N}(\mathbf{x})} \right) \wedge ... \mid \exists^{=0} \mathbf{y} \mathbf{1}_{\mathbf{y} \in \mathcal{N}(\mathbf{x})},$$

  where $N_2 \geq N_1 \geq 1$.

3.

$$\chi_l(\mathbf{x}) = \mathbf{atp}(\mathbf{x})$$
$$\Rightarrow \psi_l(\mathbf{x}) := \mathbf{1}_{\mathbf{atp}(\mathbf{x})=C}$$

where for each possible structure of a $k$-node graph ($k$ is the order of $\mathbf{x}$; we consider node orders thus there are $2^k$ structures in total; suppose the nodes of the $k$-node graph are $v_1, ..., v_k$), there is a corresponding $\psi_l$ that evaluates to **true** *iff* there is an isomorphism from the subgraph induced by $\mathbf{x}$ to the structure of the corresponding $k$-node graph that maps $v_i$ to $x_i$ for $i \in [k]$ where $\mathbf{x} = (x_1, ..., x_k)$.

We next prove that the above $\Psi_l$ indeed is the signature logic set of $\chi_l$. We denote $\psi_l^C(\mathbf{x})$ as the logic formula that evaluates to **true** *iff* $\chi_l(\mathbf{x}) = C$ where $C$ is the color of $\mathbf{x}$ evaluated by $\chi_l$. For situation 3 the statement obviously holds. For situation 1, suppose $\Psi_p, \Psi_q$ are the signature logic sets of $\chi_p, \chi_q$ respectively. For each color $C_l$ of $\chi_l$ where $C_l = \text{hash}(C_p, C_q)$, we have

$$\psi_l^{C_l}(\mathbf{x}) := \psi_p^{C_p}(\mathbf{x}) \wedge \psi_q^{C_q}(\mathbf{x})$$

which is **true** *iff* $\chi_l(\mathbf{x}) = C_l$. Thus the statement still holds.

For situation 2, suppose $\Psi_p$ is the signature logic set of $\chi_p$. Each color $C_l$ of $\chi_l$ is defined by

$$C_l = \text{hash}\left(\{\{C_p^1, C_p^2, ...\}\}\right) = \text{hash}\left(\{(C_p^1, N_1), (C_p^2, N_2), ...\}\right)$$

where $C_p^1, C_p^2, ...$ are colors produced by $\chi_p$, and $N_1, N_2, ... \geq 1$ are the numbers of the colors $C_p^1, C_p^2, ...$ emerged in the multiset, We then have

$$\psi_l^{C_l}(\mathbf{x}) := \exists^{=N}\mathbf{y}\mathbf{1}_{\mathbf{y}\in\mathcal{N}(\mathbf{x})}\exists^{=N_1}\mathbf{y}_1\left(\psi_p^{C_p^1}(\mathbf{y}_1) \wedge \mathbf{1}_{\mathbf{y}_1\in\mathcal{N}(\mathbf{x})}\right)\wedge\exists^{=N_2}\mathbf{y}_2\left(\psi_p^{C_p^2}(\mathbf{y}_2) \wedge \mathbf{1}_{\mathbf{y}_2\in\mathcal{N}(\mathbf{x})}\right)\wedge....$$

Specially, if the multiset is empty, we have

$$\psi_l^{C_l} := \exists^{=0}\mathbf{y}\mathbf{1}_{\mathbf{y}\in\mathcal{N}(\mathbf{x})}.$$

Then, $\psi_l^{C_l}(\mathbf{x})$ is **true** *iff* $\chi_l(\mathbf{x}) = C_l$. $\psi_l^{C_l}$ is also in $\Psi_l$. Therefore, we have constructed the signature logic set of $\chi_l$. Obviously, all $\psi_l \in \Psi_l$ can be written in the form of

$$\psi_l(\mathbf{x}) := \exists^{=N_1}\mathbf{x}_1...\exists^{=N_K}\mathbf{x}_K\left(\bigwedge_{p\in[P]} E(x_{i_p}, x_{j_p}) \bigwedge_{q\in[Q]} (\neg E(x_{s_q}, x_{t_q}))\right).$$

For two graphs $G, H$, if there exists $\psi$ that distinguishes them, then obviously the corresponding $\chi$ also distinguishes them. Without loss of generality, suppose the color of $\chi$ applied on $G$ is $C$. Let $\psi^C$ be the logic formula that evaluates **true** *iff* $\chi = C$. Then, we have

$$\psi^C \text{ evaluates to } \mathbf{true} \text{ on } G \text{ and } \mathbf{false} \text{ on } H.$$

Recall that since $\psi^C$ can be written in the form of

$$\psi^C(\mathbf{x}) := \exists^{=N_1}\mathbf{x}_1...\exists^{=N_K}\mathbf{x}_K\left(\bigwedge_{p\in[P]} E(x_{i_p}, x_{j_p}) \bigwedge_{q\in[Q]} (\neg E(x_{s_q}, x_{t_q}))\right),$$

the proof completes.

$\square$

### G.19   PROOF OF LEMMA 20

**Lemma 20.** *If* $\mathbf{Hom}(F, G) = \mathbf{Hom}(F, H)$ *for all* $F \in \mathcal{F}$, *then every* $\phi \in \Phi$ *of the above form satisfies:*

$$\mathbf{grd}(\phi, G) = \mathbf{grd}(\phi, H).$$

*Proof.* We prove by contradiction and assume $\textbf{Hom}(F, G) = \textbf{Hom}(F, H)$ for all $F \in \mathcal{F}$ but there exists $\phi \in \Phi$ given by

$$\phi := \exists \mathbf{x}_1 \exists \mathbf{x}_2 ... \exists \mathbf{x}_K \left( \bigwedge_{p \in [P]} E(x_{i_p}, x_{j_p}) \bigwedge_{q \in [Q]} \neg E(x_{i_q}, x_{j_q}) \right)$$

that classifies $G$ and $H$ differently. Without loss of generality we assume $\phi$ evaluates to **true** in $G$. By the definition of **grd**, obviously if $\textbf{Hom}(F, G) = \textbf{Hom}(F, H)$ for all $F \in \mathcal{F}$, we have $\textbf{grd}(\psi, G) = \textbf{grd}(\psi, H)$ for all $\psi \in \Psi \subseteq \Phi$ where $\psi \in \Psi$ is of the form

$$\psi := \exists \mathbf{x}_1 \exists \mathbf{x}_2 ... \exists \mathbf{x}_K \left( \bigwedge_{p \in [P]} E(x_{i_p}, x_{j_p}) \right),$$

i.e. $\psi$ contains no negative edges $\neg E$. We next show that we can use $\textbf{grd}(\psi, \cdot)$ for $\psi \in \Psi$ to infer $\textbf{grd}(\phi, \cdot)$. We denote

$$\phi_q := \exists \mathbf{x}_1 ... \exists \mathbf{x}_K \left( \bigwedge_{p \in [P]} E(x_{i_p}, x_{j_p}) \bigwedge_{r \in [q]} (\neg E(x_{s_r}, x_{t_r})) \right)$$

We now show that we can use the results $\textbf{grd}(\psi, \cdot)$ for $\psi \in \Psi$ to infer $\textbf{grd}(\phi^q, \cdot)$ for $q = 0, 1, ..., Q$. We denote $\mathbf{X} = (\mathbf{x}_1, ..., \mathbf{x}_K)$. For $q = 0$, obviously $\phi^0 \in \Psi$ thus the statement naturally holds. Since the groundings of $\phi^0$ consist of two parts:

- $\mathbf{X}$ that satisfy $\bigwedge_{p \in [P]} E(x_{i_p}, x_{j_p}) \wedge E(x_{s_1}, x_{t_1})$;

- $\mathbf{X}$ that satisfy $\bigwedge_{p \in [P]} E(x_{i_p}, x_{j_p}) \wedge \neg E(x_{s_1}, x_{t_1})$, corresponding to $\phi_1$.

Obviously the two part do not intersect. Therefore, $\textbf{grd}(\phi_1, \cdot) = \textbf{grd}(\phi_0, \cdot) - \textbf{grd}(\varphi^{(1)}, \cdot)$ where

$$\varphi^{(1)} := \exists \mathbf{x}_1 ... \exists \mathbf{x}_K \left( \bigwedge_{p \in [P]} E(x_{i_p}, x_{j_p}) \wedge E(x_{s_1}, x_{t_1}) \right).$$

Similarly, to infer $\textbf{grd}(\phi_2, \cdot)$, the set of $\mathbf{X}$ that satisfy $\left( \bigwedge_{p \in [P]} E(x_{i_p}, x_{j_p}) \right)$ consists of four non-intersect parts:

- $\bigwedge_{p \in [P]} E(x_{i_p}, x_{j_p}) \wedge E(x_{s_1}, x_{t_1}) \wedge E(x_{s_2}, x_{t_2})$,

- $\bigwedge_{p \in [P]} E(x_{i_p}, x_{j_p}) \wedge E(x_{s_1}, x_{t_1}) \wedge \neg E(x_{s_2}, x_{t_2})$,

- $\bigwedge_{p \in [P]} E(x_{i_p}, x_{j_p}) \wedge \neg E(x_{s_1}, x_{t_1}) \wedge E(x_{s_2}, x_{t_2})$,

- $\bigwedge_{p \in [P]} E(x_{i_p}, x_{j_p}) \wedge \neg E(x_{s_1}, x_{t_1}) \wedge \neg E(x_{s_2}, x_{t_2})$,

According to our assumption the first part is known. For the second part, since $\left( \bigwedge_{p \in [P]} E(x_{i_p}, x_{j_p}) \wedge E(x_{s_1}, x_{t_1}) \right)$ consists of two non-intersect parts:

- $\bigwedge_{p \in [P]} E(x_{i_p}, x_{j_p}) \wedge E(x_{s_1}, x_{t_1}) \wedge E(x_{s_2}, x_{t_2})$,

- $\bigwedge_{p \in [P]} E(x_{i_p}, x_{j_p}) \wedge E(x_{s_1}, x_{t_1}) \wedge \neg E(x_{s_2}, x_{t_2})$.

Thus the second part can also be inferred, and so does the third part. Therefore, the set of nodes that satisfy $\left( \bigwedge_{p \in [P]} E(x_{i_p}, x_{j_p}) \right)$ can be inferred, and thus also $\textbf{grd}(\phi_2, G) = \textbf{grd}(\phi_2, H)$ Using the same strategy one can show that

$$\textbf{grd}(\phi_q, G) = \textbf{grd}(\phi_q, H)$$

for any $q$. Therefore, this yields the contradiction.

$\square$

