# OpenReview forum: "Towards a Complete Logical Framework for GNN Expressiveness"
_ICLR.cc/2025/Conference — ICLR 2025 Oral_

### Official Review · Reviewer_pXBi · 2024-10-19

**Soundness:** 3
**Presentation:** 3
**Contribution:** 4
**Rating:** 8
**Confidence:** 3

**Summary:**

The paper first proposes GACNNs, a generalisation of several GNN architectures, using sequences of combination and aggregation functions over node tuples. This allows them to subsume several existing GNNs models. They then prove the logical expressivity of GACNNs and apply those results to find the equivalent logic sets for 8 GNN variants. Finally, they show how the logical expressivity can be used to also find the homomorphism expressivity and k-WL expressivity of the model.

**Strengths:**

1. The contribution is significant and will be of interest to the ICLR community. It unifies existing work on logical expressivity of GNNs, as well as showing the consequences that this has for two other important methods of analysing GNN expressivity.
2. The paper makes good use of space, sticking to the recommended 9 pages and rarely making redundant statements.
3. The quality of the work appears to be good, although I haven't been able to properly verify the soundness of the proofs.
4. The authors have made it clear where their work fits into the existing literature, and what their contribution is.

**Weaknesses:**

Major:
1. The main section of the paper lacks intuitions as to why the theorems are true, as well as proof sketches. These would greatly benefit the reader, particularly for a central result such as Theorem 3.
2. Due to a mixture of typos and imprecision, it is sometimes unclear what the authors are intending to stay from a technical perspective. This is particularly problematic in a paper that consists solely of theory. For example: on L262, the "logic sets at the bottom" have so far only been presented to be able to be atp(u), so what else could this refer to? And on L171, what does 'u' refer to? One might reasonably assume a node, but further down it also appears to refer to node tuples.
3. I have many other smaller concerns with the paper, which altogether come together into a major concern.

Medium:
1. L187 the example is far too simple and doesn't do enough to explain the idea of having a sequence of operations within each layer. It is not obvious how other real GNNs can be made from these building blocks. Using a more complex example like Local 2-GNN would be more beneficial.
2. In Theorem 3, equivalent logic sets are being used for each x_i, but they have so far only been defined for GNNs, not these building blocks. A more general definition of equivalent logic sets is needed.
3. L327 "auxiliary logic set". One might assume that this relates to global readouts, given that these greek letters were just used above. However, that interpretation does not seem to fit with the proposition. It is not clear, and should be stated explicitly what this logic set is for, as well as using different notation to separate it from global readouts if that is not what it intends to represent.
4. L331 the jump from the GNN models to their expressivity is not clear. It would be helpful to see an example of the whole pipeline in effect: GNN model -> GACNN -> equivalent logic -> simplification of the logic
5. L406 the construction is presented in a somewhat clunky manner, and would benefit from being both more precise and concise. E.g. the enumeration (a-c) is just the definition of a graph that can be done inline

Minor:
1. L35 "many studies" needs a citation
2. L38 it is unclear why k-WL lacks interpretability
3. L49 "although provides"
4. L55 MPNNs are already a generalisation of several GNNs models, so it is misleading to say they analyse different GNN models separately.
5. L76 "is denoted"
6. L97 "leave"
7. L105 "predicate", "corresponds"
8. In section 3.1, GNNs are referred to without having been defined yet.
9. L140 "district"
10. L152 "the elements" -> "each element"
11. L166 no need for "etc", since you already said "including"
12. L172 should be clearer that it's a sequence of two operations, not just two operations
13. L182 "variants", but only one is described
14. L187 the appendix should be referenced here for the construction of the other GNN models within this framework
15. L254 "existed"
16. L255 "edges" -> "neighbourhoods", since we're generalising to node tuples
17. L270 "function", "Similar as previous"
18. L281 "is" = "are"
19. L298 "its layers" -> "a layer", since all layers have been assumed to have the same structure
20. L309 "provides"
21. L323 "prediction"
22. L365 "although computes"
23. L376 "tasks"
24. L392 should this not be "homomorphism expressivity"? Homogenous has other connotations. Also "beautiful" is emotive language and should be avoided
25. L404 full stop should not be there
26. L464 - 466 is a bit verbose, and restating the obvious

**Questions:**

1. L134 the GNN outputs "true" for a node. Does this mean that GNNs are restricted to those that do binary classification on nodes? If so, this is not stated explicitly anywhere.
2. L140 why are logic sets "quantitative" but k-WL or homomorphism tests are not?
3. L171 - what does "u" refer to? This needs to be more explicit.
4. L179 I find it strange that the computation counts down (from K + 1) within a GACNN layer, but at the layer level it counts up from 0. Is there a good reason for this?
5. L180 "generalised neighbour" has not been defined. What does this precisely refer to?
6. L196 why not use an example of an existing GNN model here?
7. L262 "logic sets at the bottom" have so far only been presented to be able to be atp(u), so what else could this refer to?
8. L351 "more general v". In what sense is it more general? And why is it obviously more powerful?
9. L450 how useful is the construction, given its recursive nature?
10. L472 why is the WL bound on SEAL so weak, given proposition 9? Shouldn’t we be able to exactly specify it’s WL expressivity?
11. Is the paper claiming to have generalised all GNNs? L482 seems to suggest so.

---

> ### Author Response · Authors · 2024-11-18
> **Response (Part 1)**
>
> We sincerely thank the reviewer for the detailed and constructive comments! We admit that some parts of the paper were written in a rush and we didn't have enough time for checking. We have updated a revised paper to improve precision, correct typos and address the weaknesses. The modified parts in the main texts of the revised paper are colored blue.
>
> # Weakness
> ## Major
>
> ```
> 1. The main section of the paper lacks intuitions as to why the theorems are true, as well as proof sketches. These would greatly benefit the reader, particularly for a central result such as Theorem 3.
> ```
>
> We have added proof sketches for the central theorems and propositions, which also provide intuitions about why the theorems / propositions are true. Please search for "proof sketch" in the revised paper.
>
> ```
> 2. Due to a mixture of typos and imprecision, it is sometimes unclear what the authors are intending to stay from a technical perspective. This is particularly problematic in a paper that consists solely of theory.
> ```
>
> We have checked the paper again and corrected typos, as well as rewriting mathematical symbols, definitions and theorems to improve precision.
>
> ```
> 2. For example: on L262, the "logic sets at the bottom" have so far only been presented to be able to be atp(u), so what else could this refer to?
> ```
>
> L262 (original) -> L321 (new), the "logic sets at the bottom" in the original paper can only be atp(u). In the revised paper we have modified the figure to represent one layer of Local 2-GNN layer, where each logic set corresponds to a representation computed in Local 2-GNN layers. Please refer to Page 5, Figure 1 to check the difference.
>
> ```
> 2. And on L171, what does 'u' refer to? One might reasonably assume a node, but further down it also appears to refer to node tuples.
> ```
>
> L171 (original) -> L194 (new). $\mathbf{u}$ refer to node tuples. Recall that we use bold letters $\mathbf{u}$ to represent $k$-order node tuples (Section 2). As a special case, if $k=1$, $\mathbf{u}$ can also represent a node. \
> This is a trick for notation brevity, so that we can use the symbol $\mathbf{u}$ for both describing MPNNs which learn node representations and 2-GNNs which learn representations for node pairs.
>
> ## Medium:
>
> ```
> 1. L187 the example is far too simple and doesn't do enough to explain the idea of having a sequence of operations within each layer. It is not obvious how other real GNNs can be made from these building blocks. Using a more complex example like Local 2-GNN would be more beneficial.
> ```
>
> L187 (original) -> L211 (new). We have added a more complex example Local 2-GNN and showed how we decompose its layer into a series of AGG and COM functions. Please refer to L236 to check the details.
>
> ```
> 2. In Theorem 3, equivalent logic sets are being used for each x_i, but they have so far only been defined for GNNs, not these building blocks. A more general definition of equivalent logic sets is needed.
> ```
>
> We have added a generalized definition in L278.
>
> ```
> 3. L327 "auxiliary logic set". One might assume that this relates to global readouts, given that these greek letters were just used above. However, that interpretation does not seem to fit with the proposition. It is not clear, and should be stated explicitly what this logic set is for, as well as using different notation to separate it from global readouts if that is not what it intends to represent.
> ```
>
> L327 (original) -> L389 (new). In the original paper, the auxiliary logic sets are not global readouts. These auxiliary logic sets are used to help define the equivalent logic set of GNNs. For example, if $\varphi(x):=\exists y\psi(y)\land E(x,y),\psi(y):=\exists z Red(z)\land E(z,y)$, then $\psi$ is the auxiliary logic formula used to help define $\varphi$.
>
> In the revised paper we have corrected the notations and provided a short explanation for the auxiliary logic sets.
>
> ```
> 4. L331 the jump from the GNN models to their expressivity is not clear. It would be helpful to see an example of the whole pipeline in effect: GNN model -> GACNN -> equivalent logic -> simplification of the logic
> ```
>
> L331 (original) -> L394 (new).
> We have added a proof sketch for the proposition which explains the whole pipeline.
>
> ```
> 5. L406 the construction is presented in a somewhat clunky manner, and would benefit from being both more precise and concise. E.g. the enumeration (a-c) is just the definition of a graph that can be done inline
> ```
>
> L406 (original) -> L491 (new).
> We have rewritten the paragraph to make the presentations of the counstruction more precise and concise.
>
> ## Minor:
>
> We have checked and corrected minor issues.

---

> ### Author Response · Authors · 2024-11-18
> **Response (Part 2)**
>
> # Questions
>
> ```
> Q1: L134 the GNN outputs "true" for a node. Does this mean that GNNs are restricted to those that do binary classification on nodes? If so, this is not stated explicitly anywhere
> ```
>
> L134 (original) -> L150 (new).
> Yes. To study the relation between GNNs and logic formulas (which only output "true" or "false"), the GNNs are restricted to do binary classification.
>
> We have revised the paper and explicitly state this in paragraph "Graph neural networks", Section 2.
>
> ```
> Q2: L140 why are logic sets "quantitative" but k-WL or homomorphism tests are not?
> ```
>
> L140 (original) -> L160 (new).
> Homomorphism tests are also quantitative. The concept of quantitative is proposed in [1] (Section 3.1, page 4, "Second, homomorphism expressivity is a quantitative measure..."). According to their definition, given a class of GNNs $M$, a qualitative measure such as k-WL, only provide qualitative results such as "
> $M$ is k-WL-expressive" and "$M$ is not (k+1)-WL-expressive", which only contains true-or-false results. A quantitative measure is much finer and precisely describe the expressive power for GNN models, in this case precisely identifies all logic formulas that can be captured by $M$ (Section 3.1, page 4 in [1]).
>
> ```
> Q3: L171 - what does "u" refer to? This needs to be more explicit.
> ```
>
> L171 (original) -> L194 (new).
> $\mathbf{u}$ refer to node tuples. Recall that we use bold letters $\mathbf{u}$ to represent $k$-order node tuples (Section 2). As a special case, if $k=1$, $\mathbf{u}$ can also represent a node. \
> This is a trick for notation brevity, so that we can use the symbol $\mathbf{u}$ for both describing MPNNs which learn node representations and 2-GNNs which learn representations for node pairs.
>
> ```
> Q4: L179 I find it strange that the computation counts down (from K + 1) within a GACNN layer, but at the layer level it counts up from 0. Is there a good reason for this?
> ```
>
> L179 (original) -> L204 (new).
> We have revised the paper so that the computation within a GACNN layer also counts up from 0.
>
> ```
> Q5: L180 "generalised neighbour" has not been defined. What does this precisely refer to?
> ```
>
> L180 (original) -> L205 (new).
> MPNNs learn representations for nodes $x$. The neighbors of $x$ is defined by $N(x)=\{ y ~ | ~ E(x,y)\in\mathcal{E} \}$ where $\mathcal{E}$ is the set of edges. Consider NBFNet which learns representations for node pairs $(x,y)$. Its layers is defined as $\chi^{(l+1)}(x,y)=COM(\chi^{(l)}(x,y),AGG(\{\chi^{(l)}(x,z) ~ | ~ (z,y)\in\mathcal{E}\}))$. Therefore, in NBFNet's $AGG$ function the neighbor of $(x,y)$ is defined as $N(x,y)=\{(x,z) ~ | ~ (z,y)\in\mathcal{E}\}$. In this case, NBFNet has generalized the concept of neighbour, compared with MPNNs.
>
> We have revised the paper and used an example to clarify this.
>
> ```
> Q6: L196 why not use an example of an existing GNN model here?
> ```
>
> L196 (original) -> L253 (new).
> Here, we want to illustrate the idea using a relative complex GACNN structure. We have replaced the example with Local 2-GNN.
>
> ```
> Q7: L262 "logic sets at the bottom" have so far only been presented to be able to be atp(u), so what else could this refer to?
> ```
>
> L262 (original) -> L321 (new).
> It can only be atp(u) in this case. We have revised the relevant parts in the paper.

---

> ### Author Response · Authors · 2024-11-18
> **Response (Part 3)**
>
> ```
> Q8: L351 "more general v". In what sense is it more general? And why is it obviously more powerful?
> ```
>
> L351 (original) -> L431 (new).
> Consider the following two graphs, which are cycles of 3 and 4 nodes. It is evident that MPNNs cannot distinguish the nodes in (a) and the nodes in (b): MPNNs only use edges $E(x,y)$ to model the relation of nodes, and for each node (denoted by $x$) in both (a) and (b), there are exactly two nodes (denoted by $y$) that satisfy $E(x,y)$. \
> Subgraph GNNs (weak) utilize logic formulas $\psi(x,y)$ to model the relation between nodes. If we let $\psi(x,y):=\exists z(E(x,z)\land E(z,y))$ (which can be expressed by subgraph GNNs(weak)), each node $x$ in (a) has two nodes $y$ that satisfy $\psi(x,y)$, while for (b) there is only one. In this sence, $\psi$ is more general (as it can model more complex relation $\exists z(E(x,z)\land E(z,y))$ between nodes) and more powerful (distinguish nodes that are indistinguishable by MPNNs).
>
> $o       ~~~~~~~~~~~~ o-o$\
> | $~ \backslash  ~~~~~~~~ |~~~~~~|$\
> $o-o   ~~~~~     o-o\$ \
> $~~(a)~~~~~~~~~(b)$
>
> ```
> Q9: L450 how useful is the construction, given its recursive nature?
> ```
>
> L450 (original) -> L558 (new).
> Since the equivalent logic sets (and the logic expressivity in previous works [4]) are defined in a recursive manner, we also illustrate a method to recursive construct the homomorphism expressivity. The recursive construction of homomorphism expressivity is useful when trying to enumerate patterns that GNNs can capture, but is less useful if one only needs to test whether a certain subgraph structure can counted by GNNs. In this case, we can use the correspondence in L490 of the revised paper to judge whether the structure can be counted by GNNs.
>
> ```
> Q10: L472 why is the WL bound on SEAL so weak, given proposition 9? Shouldn’t we be able to exactly specify it’s WL expressivity?
> ```
>
> L472 (original) -> L560 (new).
> This is because that SEAL is indeed incomparable with 3-WL: there exists graph pairs that SEAL fails to distinguish but 3-WL distinguishes [2] and graph pairs such that SEAL distinguishes but 3-WL fails to distinguish (Shrikhande Graph and rook's 4x4 graph). Also, 2-WL is equivalently expressive in distinguishing graph pairs as 1-WL.
>
> ```
> Q11: Is the paper claiming to have generalised all GNNs? L482 seems to suggest so.
> ```
>
> L482 (original) -> L570 (new).
> We generalize all GNNs that can be described by GACNNs, i.e. GNNs whose layers are built upon aggregation and combination layers. For example, Graphormer-GD [3] which injects distance information into node pairs are not covered here, because the procedure of injecting distance information cannot be expressed by aggregation or combination layers. It is possible to find the upper bound of Graphormer-GD, i.e. to find a logic set that is more expressive than Graphormer-GD, as well as finding the lower bound of Graphormer-GD, i.e. finding a logic set that can be captured by Graphormer-GD. However, we do not know whether Graphormer-GD has an equivalent logic set $\Phi$, i.e. $\Phi$ is equivalently expressive with Graphormer-GD. Although we highly suspect the answer is negative, the results in this paper cannot generalize to such situation.
>
> [1] Zhang et al. BEYOND WEISFEILER-LEHMAN: A QUANTITATIVE FRAMEWORK FOR GNN EXPRESSIVENESS. ICLR 2024.
>
> [2] Qian et al. Ordered Subgraph Aggregation Networks.
>
> [3] Zhang et al. Rethinking the Expressive Power of GNNs via Graph Biconnectivity. ICLR 2023.
>
> [4] Barcelo et al. THE LOGICAL EXPRESSIVENESS OF GRAPH NEURAL NETWORKS. ICLR 2020.

---

> ### Comment · Reviewer_pXBi · 2024-11-19
> **Response to author rebuttal**
>
> I would like to thank the authors for so thoroughly engaging with my feedback.
> The edits made to the paper and clarifications provided have largely addressed my concerns. I am still not quite able to verify that the proofs are correct, but the theory is certainly presented more clearly now.
>
> All line number references from here on will refer to the new revision (V2). I hope to have a couple more queries answered before I update my score.
>
> Some remaining questions / points I have:
> 1. For bullet point 3 of Definition 2, 153 - the iff seems to imply that there it a one-to-one equivalence between the GNN expressivity and the logical expressivity. This does not correspond to my understanding of the existing work: Barcelo [1], for example, show that "Theorem 4.2. A logical classifier is captured by AC-GNNs if and only if it can be expressed in graded modal logic.". This is different to saying that graded modal logic is exactly equivalent to AC-GNNs. The claim is rather that any FOC classifier that is captured is one expressible in graded modal logic. There do exist particular MPNNs without an equivalent logical formula. However, your statement on L182 calls this result an "equivalent logic set".
>
> Could you please clarify my understanding here: is your iff showing an exact equivalence between the class of GNN models and the class of logical formulas?
>
> 2. I see that the third bullet point of definition 2 has been changed from before to talk about nodes in the graphs, rather than about the graphs. This is good, and has corrected an error that I failed to catch in the first draft. However, given that Proposition 7 defines an equivalent logic set for a GNN with readout, should there not be a brief mention here for how you can adapt the standard definition of logical equivalence to work for GNNs that give graph outputs, instead of node tuple outputs?
>
> 3. `To study the relation between GNNs and logic formulas (which only output "true" or "false"), the GNNs are restricted to do binary classification.` - this is mostly true in the existing literature, but there are ways to handle multi-class classification. See this recent work at KR, for example: https://proceedings.kr.org/2024/84/. However, given that other existing work approaches it as a binary classification problem, and the different approaches can be easily compared, I do not have a problem with this paper using only binary classification.
>
> 4. `This is because that SEAL is indeed incomparable with 3-WL` - I see that, and your explanation makes sense to me. However, I still am not seeing how Proposition 10 does not always give you an exact WL-bound, given the iff in the theorem. This then seems to imply that Proposition 10 is somehow wrong? Could you please clarify why Proposition 10 cannot give an exact WL-bound in the SEAL case?
>
> A couple of minor points on the new revision:
> 1. L107 - the node should be Red
> 2. L330 Main Results is perhaps a strange name for this section, given that Theorem 4 is defined above.
> 3. Small nitpick: the end of proof sketches should be more clearly marked (e.g. with the standard end-of-proof box). It currently just appears as a paragraph and will be harder to see the boundary when the blue text is removed.
> 4. In the background, please define generalised neighbourhoods instead of just regular neighbourhoods (or at least mention that you will use a generalised definition of neighbourhood).
> 5. If you need to save space, definition 2 and 3 might be a bit redundant, although I appreciating you adding definition 3 to be technically sound. Could you not unify both under a single more general definition? (I don't feel strongly on this, just a suggestion)
> 6. L568 "sole" -> "solely", "are not" -> "is not"
>
> [1] Pablo Barceló, Egor V Kostylev, Mikaël Monet, Jorge Pérez, Juan Reutter, et al.. THE LOGICAL
> EXPRESSIVENESS OF GRAPH NEURAL NETWORKS. 8th International Conference on Learning
> Representations (ICLR 2020), Apr 2020, Virtual conference, Ethiopia. ffhal-03356968f

---

> > ### Author Response · Authors · 2024-11-20
> > **Response (part 2)**
> >
> > There are also other works that study the equivanlence between logic and Weisfeiler-Lehman tests. Cai et al. [2] have discovered the following theorem:
> >
> > Theorem 5.2. Let $G$,$H$ be a pair of colored graphs and let $(u,v)$ be a k-configuration (which corresponds to k-order node tuples in our setting) on $G,H$, where $k ≥ 1$. Then the following are equivalent:
> > 1. $W(u) = W(v)$
> > 2. $G,u ≡_{Ck+1} H,v$,\
> > where $W$ is the stable color computed by (k-1)-FWL and $G,u ≡_{Ck+1} H,v$ indicates that all logic formulas in $FOC_{k+1}$ cannot distinguish $(G,u)$ and $H,v$.
> >
> > The above theorem describes the equivalence of WL and logic in a similar manner with ours.
> >
> > ```
> > I see that the third bullet point of definition 2 has been changed from before to talk about nodes in the graphs, rather than about the graphs. This is good, and has corrected an error that I failed to catch in the first draft. However, given that Proposition 7 defines an equivalent logic set for a GNN with readout, should there not be a brief mention here for how you can adapt the standard definition of logical equivalence to work for GNNs that give graph outputs, instead of node tuple outputs?
> > ```
> >
> > In fact we already use $\mathbf{u}$ to express graph-level outputs. In our definition in Section 2, we defined that if the order $k$ of $\mathbf{u}$ is 0, then $\mathbf{u}$ represents $\text{None}$. Concretely, if a logic formula $\varphi$ discribes graph-level outputs, the the order of $\varphi$ is $0$. An example would be \
> > $\varphi:=\exists x(Red(x))$. \
> > In this paper we discuss about logic formulas with different orders via a unified framework, therefore we also need a unified symbol to express logic formulas with different orders. For example we can abbreviate the term $\varphi_1(x_1,y_1)$ into $\varphi_1(\mathbf{u}_1)$ where $\mathbf{u}_1:=(x_1,y_1)$ and similarly $\varphi_1(x_2,y_2,z_2)$ into $\varphi_1(\mathbf{u}_2)$ where $\mathbf{u}_2:=(x_2,y_2,z_2)$. Specially, we abbreviate $\varphi$ into $\varphi(\mathbf{u})$ where $\mathbf{u}:=\text{None}$ and the order of $\mathbf{u}$ is 0.
> >
> > ```
> > To study the relation between GNNs and logic formulas (which only output "true" or "false"), the GNNs are restricted to do binary classification. - this is mostly true in the existing literature, but there are ways to handle multi-class classification. See this recent work at KR, for example: https://proceedings.kr.org/2024/84/. However, given that other existing work approaches it as a binary classification problem, and the different approaches can be easily compared, I do not have a problem with this paper using only binary classification.
> > ```
> >
> > Thanks for pointing it out. We have checked the recent work at KR. We also believe that it is possible to extend our work to multi-class classification GNNs, and we plan to add a section (probably in Appendix) discussing about this. Hopefully, this will be done before discussion period ends.
> >
> > ```
> > This is because that SEAL is indeed incomparable with 3-WL - I see that, and your explanation makes sense to me. However, I still am not seeing how Proposition 10 does not always give you an exact WL-bound, given the iff in the theorem. This then seems to imply that Proposition 10 is somehow wrong? Could you please clarify why Proposition 10 cannot give an exact WL-bound in the SEAL case?
> > ```
> >
> > Proposition 10 only gives the upper bound of GNNs. We have rewrite Proposition to make it more precise as below:
> >
> > Proposition 10. Suppose the equivalent logic set of a class of GNN models is $\Phi$. Then, the expressive power of the GNN models is bounded by $k$-WL, iff all logic formulas in $\Phi$ can be expressed with at most $k$ variables.
> >
> > In fact, Proposition 10 is a simple corollary of Theorem 5.2 in [2] (presented above): all formulas in the equivalent logic set $\Phi$ of a class of GNNs can be described with at most $k$ variables $\iff$ $\Phi$ is a subset of $FOC_{k}$ $\iff$ $\Phi$ is bounded by (k-1)-FWL $\iff$ $\Phi$ is bounded by k-WL $iff$ the class of GNNs is bounded by k-WL.
> >
> > Therefore, for SEAL, Proposition 10 only gives half of the results. Let $\Phi_{SEAL},\Phi_{MPNN}$ be the equivalent logic sets of SEAL and MPNNs respectively. We also need to prove that SEAL>1-WL by showing that $\Phi_{MPNN}\subset\Phi_{SEAL}$.
> >
> > # Minor points
> >
> > ```
> > L107 - the node should be Red
> > ```
> >
> > We have corrected this issue.
> >
> >
> > ```
> > L330 Main Results is perhaps a strange name for this section, given that Theorem 4 is defined above.
> > ```
> >
> > We have changed the name into "Equivalent Logic Sets for GACNNs".
> >
> > ```
> > Small nitpick: the end of proof sketches should be more clearly marked (e.g. with the standard end-of-proof box). It currently just appears as a paragraph and will be harder to see the boundary when the blue text is removed.
> > ```
> >
> > We have added a latex symbol \qed at the end of each proof sketch.

---

> > ### Author Response · Authors · 2024-11-20
> > **Response (part 3)**
> >
> > ```
> > In the background, please define generalised neighbourhoods instead of just regular neighbourhoods (or at least mention that you will use a generalised definition of neighbourhood)
> > ```
> >
> > We have added a definition of generalized neighborhoods in Section 2.
> >
> > ```
> > If you need to save space, definition 2 and 3 might be a bit redundant, although I appreciating you adding definition 3 to be technically sound. Could you not unify both under a single more general definition? (I don't feel strongly on this, just a suggestion)
> > ```
> >
> > We have combined Definition 2 and 3 into Definition 2 in Section 3, which defines the equivalent logic set for a family of functions $\chi\in\mathcal{X}$ that maps node tuples to $\{true,false\}$. This covers both GNNs and the computation units in Section 5.1.
> >
> > ```
> > L568 "sole" -> "solely", "are not" -> "is not"
> > ```
> >
> > We have corrected this.
> >
> >
> > #
> >
> > Overall, we sincerely thanks the reviewer for the constructive comments. Please feel free to comment if there are concerns / questions that are not properly addressed. We are also working on :(1) adjusting the proofs in Appendix to follow the changes in main text, and (2) conducting numerical experiments to validate the results in this paper. We will finish these works before the discussion period ends.
> >
> >
> >
> >
> > [1] Barcelo et al. THE LOGICAL EXPRESSIVENESS OF GRAPH NEURAL NETWORKS. ICLR 2020.
> >
> > [2] Cai et al. An Optimal Lower Bound on the Number of Variables for Graph Identification. Combinatorica, (12:4) (1992),389-410.

---

> > ### Comment · Reviewer_pXBi · 2024-11-21
> > **Response**
> >
> > Thanks again to the authors for replying: this is a most engaging discussion. Please see my response below.
> >
> > >Thanks for pointing this out! We have thoroughly investigated [1]. In short, (1) our defintion of equivalent logic set differs from Theorem 4.2 in [1] and (2) yes, our iff shows an exact equivalence between the class of GNN models and the class of logical formulas.
> >
> > This is substantial, and to my knowledge represents entirely novel work for the class of unrestricted combine-aggregate GNNs. Such results exist for restricted classes, e.g. monotonic max-sum GNNs: https://arxiv.org/abs/2305.18015. It's also a little concerning that this (crucially important) subtlety slipped past in the initial writing of the paper. As such, I am concerned that there may be some issues in the theory, that I may not end up catching. However, I will do my best to engage.
> >
> > > An interesting problem is, whether the equivalent logic sets of GNNs are always maximal.
> >
> > What precisely do you mean by "maximal"? If your GNNs and logical formulas are always exactly equivalent, that it seems like they're always maximal and minimal, under a natural understanding of those terms.
> >
> > > We are uncertain about the sentence "There do exist particular MPNNs without an equivalent logical formula."
> >
> > I was imprecise there, sorry. I meant that the latter: there exists an MPNN with output $\chi$ such that no FOL formula $\gamma$ satisfied the following: for any graph $G$ and nodes $u, v \in G$, $\gamma(u) = \gamma(v) \iff \chi(u) = \chi(v)$.
> >
> > The counterexample you raised is the exact one I was about to raise, so I appreciate that. However, I do not understand how your response solves this issue. You need up ending to define the logic formula with respect to a particular node $z$, and not the GNN in general. So it seems like you end up with a different logical formula for each node, rather than one logical formula which captures the GNN as a whole.
> >
> > Could you please unpack and explain this further? (using the above example about an even number of neighbours). I was following your explanation and agreeing with you every step of the way until the line "Instead, our theory still holds".
> > This will be a crucial point for me understanding and accepting or rejecting the paper, as this seems to get down to the very essence of what you mean by "capturing" a model.
> >
> > >Thanks for pointing it out. We have checked the recent work at KR. We also believe that it is possible to extend our work to multi-class classification GNNs, and we plan to add a section (probably in Appendix) discussing about this.
> >
> > You're most welcome. I think just a very brief mention of this would more than suffice, since multi-class classification can be covered by simply have a different equivalent logical formula for each output class. Then it's binary again, and your theory holds.
> >
> > > Proposition 10 only gives the upper bound of GNNs.
> > Thank you for your clarifications here. I was missing that it's a strict upper bound, which leads to the results you present.
> >
> > **Overall**: thank you for your continued engagement. I hope we can continue to figure this out and clear up confusion. Glad to hear about the ongoing experiments and adaptations to the proofs. I do not care about the experiments much (if at all), given that the contribution of this paper is theoretical, but would appreciate more clarity in the proofs. I'm still finding them quite difficult to follow.

---

> ### Author Response · Authors · 2024-11-20
> **Response (part 1)**
>
> We sincerely thank the reviewer for the detailed and constructive comments. We will address the questions below.
>
> ```
> For bullet point 3 of Definition 2, 153 - the iff seems to imply that there it a one-to-one equivalence between the GNN expressivity and the logical expressivity. This does not correspond to my understanding of the existing work: Barcelo [1], for example, show that "Theorem 4.2. A logical classifier is captured by AC-GNNs if and only if it can be expressed in graded modal logic.". This is different to saying that graded modal logic is exactly equivalent to AC-GNNs. The claim is rather that any FOC classifier that is captured is one expressible in graded modal logic. There do exist particular MPNNs without an equivalent logical formula. However, your statement on L182 calls this result an "equivalent logic set".
> ```
>
> Thanks for pointing this out! We have thoroughly investigated [1]. In short, (1) our defintion of equivalent logic set differs from Theorem 4.2 in [1] and (2) yes, our $iff$ shows an exact equivalence between the class of GNN models and the class of logical formulas. We now explan in detail.
>
> (1) Our defintion of equivalent logic set differs from Theorem 4.2. We have added a sentence in L182 to clarify this. The idea in our paper is to find a set of logic formulas which not only are captured by GNNs, but also capture the GNNs. Therefore, intuitively, for a class of GNNs $M$, we want to find a set of logic formulas $\Phi$ satisfying:
> - all $\varphi\in\Phi$ are captured by $M$.
> - $\Phi$'s ability of distinguishing nodes / nodes tuples / graphs is equivalent to that of $M$.
>
> The intuition in [1] is rather to find a set of logic formulas $\Phi$ for a class of GNNs $M$ that satisfies:
> - all $\varphi\in\Phi$ are captured by $M$ (which also exists in our setting).
> - $\Phi$ is maximal: all logic formulas that can be captured by $M$ is expressed in $\Phi$.
>
> In order to fully describe the logical expressivity of GNNs, we believe it is necessary for $\Phi$ to share the same distinguishing power with $M$.
>
> An interesting problem is, whether the equivalent logic sets of GNNs are always *maximal*. According to [1], the equivalent logic set of MPNNs is maximal. For other GNNs and more generally, arbitrary GACNN, we conjecture that the equivalent logic sets are also maximal. However, we are still working on verifying the conjecture, and cannot guarantee its correctness.
>
> (2) Our $iff$ shows an exact equivalence between the class of GNN models and the class of logical formulas. We are uncertain about the sentence "There do exist particular MPNNs without an equivalent logical formula." Does this indecate:
> - There exists a class of GNNs which is a subset of MPNNs that does not have an equivalent logic set, or
> - There exists a MPNN (with specified structure and parameters) whose output is $\chi$ such that there exists no logic formula $\varphi$ satisfying $\varphi(x)=\varphi(y)\iff\chi(x)=\chi(y)$ for arbitrary nodes $x$,$y$?
>
> For the first statement, we view all MPNNs as an ensemble, and we do not further divide MPNNs into different parts.\
> For the second statement, we would like to point out that our setting is different. Even if the second statement is true, this doesn't affect the correctness of our theory: consider the example below. Suppose a MPNN which only has one layer and is defined by
>
> $\chi(x)=AGG(\{y|y\in N(x)\})=1(|N(x)|\text{ is even})$,
>
> i.e. $\chi(x)$ is true iff $x$ has even number of neighbors. A logic formula that does the similar thing would be: $\varphi(x):=\exists^{=0}y(E(x,y))\lor\exists^{=2}y(E(x,y))\lor...$. However, we can always construct a counterexample: let $N$ be the maximal number emerged in the superscript of the quantifiers, then there always exists a node $z$ with $N+2$ neighbors and $\varphi(z)=false$ but $\chi(z)=true$. Instead, our theory still holds: for any nodes $z,w$ satisfying $\chi(z)\neq\chi(w)$ (without loss of generality we assume $\chi(x)=true$), there always exists a logic formula $\varphi'(x):=\exists^{=|N(z)|}(y)(E(x,y))$ such that $\varphi'(z)=true$ and $\varphi'(w)=false$, where $|N(z)|$ is the number of the neighbors of $z$. Therefore, our theory still holds.

---

> ### Author Response · Authors · 2024-11-23
> **Response (part 1)**
>
> Thanks again to the reviewer for the constructive comments. Below we would like to clear up the confusion about the theory in this paper.
>
> # About equivalent logic sets.
>
> According to the the response, we believe the main confusion lies in the discussion of the equivalent logic sets.
>
> ## In the intial writting of the paper:
>
> To summarize, in the initial version of the paper, given a family of functions $\chi\in\mathcal{X}$ that maps node tuples to $\{true, false\}$, we define its equivalent logic set $\Phi$ to be the set satisfying:
> 1. Each $\varphi\in\Phi$ is captured by some $\chi\in\mathcal{X}$:$\chi(\mathbf{u})=\varphi(\mathbf{u})$ for arbitrary $\mathbf{u}$.
> 2. For arbitrary graphs $G,H$ and $\mathbf{u}\in\mathcal{V}_G^k,\mathbf{v}\in\mathcal{V}_H^k$: there exists $\chi\in\mathcal{X}$ such that $\chi(\mathbf{u})\neq\chi(\mathbf{v})$ iff there exists $\varphi\in\Phi$ such that $\varphi(\mathbf{u})\neq \varphi(\mathbf{v})$.
>
> Note that this definition follows [1]. The statement 1 expresses that each $\varphi\in\Phi$ can be expressed by some $\chi\in\mathcal{X}$, which is clear. Now let's take a close look at the statement 2. If we stick to the statement 2, the equivalent logic set $\Phi$ does not restrict that for each $\chi\in\mathcal{X}$, there is a corresponded $\varphi\in\Phi$ that $\varphi(\mathbf{u})=\chi(\mathbf{u})$ always holds; nor does it restrict that for each $\chi\in\mathcal{X}$, there is a corresponded $\varphi\in\Phi$ that:
> - For arbitrary graphs $G,H$, $\mathbf{u}\in\mathcal{V}_G^k,\mathbf{v}\in\mathcal{V}_H^k$ and $\chi\in\mathcal{X}$, there exists $\varphi\in\Phi$ satisfying: $\chi(\mathbf{u})\neq\chi(\mathbf{v})$ iff $\varphi(\mathbf{u})\neq \varphi(\mathbf{v})$.
>
> In other words, the above definition of equivalent logic sets does not restrict that there has to be a one-to-one correspondence between $\chi\in\mathcal{X}$ and $\varphi\in\Phi$. In fact, it only defines an "all-to-all" correspondence: all $\chi\in\mathcal{X}$ cannot distinguish $\mathbf{u},\mathbf{v}$ iff all $\varphi\in\mathcal{X}$ cannot distinguish $\mathbf{u},\mathbf{v}$. Therefore, it is possibly needed to define a logic formula with respect to the particular $\mathbf{u}$, rather than only to $\chi$.
>
> **Example.** Using the example about an even number of neighbors: $$\chi(x)=AGG(\{y|y\in N(x)\})=1(|N(x)|\text{ is even}).$$
>
> $~~~~~~~~~~~~~~~~~~~~~~~~~~~~~$ o \
> $~~~~~~~~~~~~~~~~~~~~~~~~~~~~~$ | \
> o--$\bullet$--o $~~~~~~~~~~~~$ o--$\bullet$--o \
> $~~~~~~~~~~~~~~~~~~~~~~~~~~~~~$ | \
> $~~~~~~~~~~~~~~~~~~~~~~~~~~~~~$ o \
> $~~~~$ (a) $~~~~~~~~~~~~~~~~~~~$ (b)
>
> Consider distinguish the bullet between circles. Obviously $\chi$ is able to distinguish the the bullet between other circles in both graph (a) and (b). Acording to the definition of equivalent logic set, there exists $\varphi_a$ that distinguishes the bullet between circles in (a); there also exists $\varphi_b$ that distinguishes the bullet between circles in (b). However, we do not restrict $\varphi_a$ to be the same with $\varphi_b$: for example, $\varphi_a(x):=\exists^{=2}y(E(x,y)),\varphi_b(x)=\exists^{=4}y(E(x,y))$.
>
> [1] Cai et al. An Optimal Lower Bound on the Number of Variables for Graph Identification. Combinatorica, (12:4) (1992),389-410.

---

> > ### Comment · Reviewer_pXBi · 2024-11-25
> > **Response r.e. Equivalent Logic Sets**
> >
> > Thanks again to the authors for another thorough response to my queries.
> > Please see my response below:
> >
> > Your response is clear to me, thank you. I believe I now understand your definition of equivalence.
> >
> > > Given that statement (1) does not hold, we believe statement (2) and (3) appropriately describes the one-to-one correspondence between GNNs and logic.
> >
> > I think that this is not quite accurate. Statement (1) has been shown to hold in some restricted cases: in [1] for example. And the solution of [2] was to say that *when* they are equivalent, then they can be expressed in a particular fragment of FOL.
> > I would be happy, however, if you simply presented your theory as another (weaker) way to look at equivalence, given that statement (1) does not hold for unrestricted classes of GNNs.
> >
> > > The reason why we cannot find $\varphi$ such that $\varphi(x) = \chi(x)$ for arbitrary graphs and nodes $x$
> >  is that we may end up with a logic formula with infinite length
> >
> > Agreed, this is the exact problem.
> >
> > >We would like to point that besides the latest change (where we added a statement to the definition of equivalent logic sets as discussed above), there was no change in our theory.
> >
> > My point about the missed subtlety is that in comparison to existing work on the logical expressivity of GNNs, your method seems to be using a different definition of logical equivalence. This crucial point did not appear in the original paper, and it should be made very clear in the final version, in my opinion.
> >
> > For example, on L161 of the original version, you state "Eq. 1 discovered by Barcelo ́ et al. (2020) describes the equivalent logic set...". But in fact, under your definition of "equivalent logic set", it does not (or rather, the Barcelo [2] result is making a different claim to the one you are making in this paper).
> >
> > One of the main established notions of equivalence appears to be the Barcelo one (in their definition 3.1). If you are presenting an alternative definition of logical equivalence to the literature, then it must please be very clear that you are doing so, and this should be motivated for directly in the paper (even if it is the one used by Cai et al.).
> >
> > [1] Cucala, David Tena, et al. "On the Correspondence Between Monotonic Max-Sum GNNs and Datalog." Proceedings of the International Conference on Principles of Knowledge Representation and Reasoning. Vol. 19. No. 1. 2023.
> >
> > [2] Barceló, Pablo, et al. "The logical expressiveness of graph neural networks." 8th International Conference on Learning Representations (ICLR 2020). 2020.

---

> > > ### Author Response · Authors · 2024-11-26
> > > **Response**
> > >
> > > We sincerely thank the reviewer again for the detailed and constructive comments. We will address the concerns below.
> > >
> > > ## About the equivalent logic set.
> > >
> > > We believe the major concern lies in the difference between our definition of the equivalent logic set and that of [1]. We have been working on checking whether the statement in [1] still holds in this paper. In the latest revision of the paper, we add a statement to the definition of equivalent logic set $\Phi$ of $\mathcal{X}$, which is corresponded to the statement in [1]:
> > > - A FOC formula is captured by $\mathcal{X}$ iff it is in $\Phi$.
> > >
> > > We now provide the intuition behind the proof of the statement. Consider $\chi(\mathbf{u})=AGG( \lbrace \lbrace\chi_1(\mathbf{v}) | \mathbf{v}\in N(\mathbf{u}) \rbrace \rbrace)$. Here, the aggregation function $AGG$ can be written in the form of
> > > $$AGG( \lbrace \lbrace\chi(\mathbf{v}) | \mathbf{v}\in N(\mathbf{u}) \rbrace \rbrace) = f(  \vert \lbrace \mathbf{v} | \mathbf{v}\in N(\mathbf{u}),\chi(\mathbf{v})=col^1  \rbrace \vert, \vert \lbrace \mathbf{v} | \mathbf{v}\in N(\mathbf{u}),\chi(\mathbf{v})=col^2  \rbrace \vert,..., \vert \lbrace \mathbf{v} | \mathbf{v}\in N(\mathbf{u}),\chi(\mathbf{v})=col^c  \rbrace \vert)$$
> > > if the input colors of $\chi$ are finite, and $col^1,...,col^c$ enumurates through all colors of $\chi$. $f$ can be regarded as counting the number of the emergence of each color. Also, $f$ is a function that maps a tuple of $c$ integers to the output colors. We say $AGG$ is finite, if there exists a positive number $n$, such that
> > > $$f(n_1,...,n_l,...,n_c)=f(n_1,...,n_l+1,...,n_c)$$
> > > holds for all $n_l>n$. In this case we say $AGG$ is bounded by $n$. In other words, $f(n_1,...,n_c)$ is constant for sufficiently large $n_1,...,n_c$. Furthermore, we define $\chi$ is finite if all aggregations for constructing $\chi$ are finite. Then, the statement is proved as follows. Suppose a FOC formula $\psi$ is captured by $\chi\in\mathcal{X}$. Let $r$ be the quantifier rank of $\psi$ and $N$ be the maximum counting number of the quantifers in $\psi$ (that is, for each term $\exists^{\geq n}$ in $\psi$, we have $N\geq n$). Then, we show that $\psi$ is captured by a finite $\chi\in\mathcal{X}$ whose aggregation functions are bounded by $rN$. After that, we show that there exists $\varphi\in\Phi$ that captures any finite $\chi\in\mathcal{X}$, thus concluding the proof.
> > >
> > > ## Other issues.
> > >
> > > ```
> > > I think that this is not quite accurate. Statement (1) has been shown to hold in some restricted cases: in [1] for example.
> > > ```
> > > By "Given that statement (1) does not hold", we mean that there exists a GNN model which is not captured by any logic formula.
> > >
> > > ```
> > > And the solution of [1] was to say that when they are equivalent, then they can be expressed in a particular fragment of FOL. I would be happy, however, if you simply presented your theory as another (weaker) way to look at equivalence, given that statement (1) does not hold for unrestricted classes of GNNs.
> > > ```
> > > About the definition in [1], please refer to the above section.
> > >
> > > ```
> > > My point about the missed subtlety is that in comparison to existing work on the logical expressivity of GNNs, your method seems to be using a different definition of logical equivalence. This crucial point did not appear in the original paper, and it should be made very clear in the final version, in my opinion.
> > >
> > > One of the main established notions of equivalence appears to be the Barcelo one (in their definition 3.1). If you are presenting an alternative definition of logical equivalence to the literature, then it must please be very clear that you are doing so, and this should be motivated for directly in the paper (even if it is the one used by Cai et al.).
> > > ```
> > >
> > > Thanks for pointing it out. Above we have shown that the definition of existsing work[1] also holds for our work. Nevertheless, we have added a paragraph introducing our motivation for choose the definition of logical equivalence.
> > >
> > > ```
> > > For example, on L161 of the original version, you state "Eq. 1 discovered by Barcelo ́ et al. (2020) describes the equivalent logic set...". But in fact, under your definition of "equivalent logic set", it does not (or rather, the Barcelo [2] result is making a different claim to the one you are making in this paper).
> > > ```
> > > We have corrected this in the revised paper.
> > >
> > > Overall, we sincerely thanks the reviewer for the constructive comments, and we hope our response have addressed your concerns.
> > >
> > > [1] Barceló, Pablo, et al. "The logical expressiveness of graph neural networks." 8th International Conference on Learning Representations (ICLR 2020). 2020.

---

> > > > ### Comment · Reviewer_pXBi · 2024-11-29
> > > > **Final Comments**
> > > >
> > > > Thank you to the authors for the last bit of engagement and the edits made to the paper. I am satisfied with the theory now.
> > > >
> > > > I am particularly happy with the definition of equivalence being clarified to "for any graphs $G$ with no more than $N$ nodes...". I find that while this is still a weaker definition of equivalence than the correspondence holding for any graph $G$, it is still a very useful one, and the strongest I can conceive of given that we know that some GNNs do not have an equivalent FOL formula.
> > > >
> > > > I trust the authors to include all the clarifications they have provided here during the review process in their final paper, and would like to encourage them to spend some time making it as concise and as readable as possible, as it offers some substantial theoretical contributions that will be of great interest to the ICLR community.
> > > >
> > > > I have adjusted my score, and strongly recommend acceptance of the paper into ICLR.
> > > > A final thanks from me to the authors for a very engaging review process. I look forward to seeing more papers from them on this topic in the future.

---

> > > > > ### Author Response · Authors · 2024-12-03
> > > > > **Thank you**
> > > > >
> > > > > Thank you for your positive feedback and great suggestion. Thanks very much for the active engagement, which helped us to improve the paper a lot. We will try our best to polish the paper to make it more concise and readable, as well as sticking to the 10-page limitation of the conference. Thanks again for the engaging review process.

---

> ### Author Response · Authors · 2024-11-23
> **Response (Part 2)**
>
> ## Extending equivalent logic sets:
>
> The above discussion only focus on our definition of equivalent logic sets in the initial writting of the paper. Below we would like to discuss the possibility to extend the concept to:
>
> (1) For arbitrary $\chi\in\mathcal{X}$, there exists $\varphi\in\Phi$ such that for arbitrary $G,H$ and $\mathbf{u}\in\mathcal{V}_G,\mathbf{v}\in\mathcal{V}_H$: $\chi(\mathbf{u})\neq\chi(\mathbf{v})\iff\varphi(\mathbf{u})\neq\varphi(\mathbf{v})$.
>
> Unfortunately, in the previous discussion it is known that there exists $\chi$ for which we cannot find the corresponding $\varphi$ that satisfies the above statement.(i.e. the example about an even number of neighbors). Does this mean that we have to define different logic formulas for each node, rather than one logical formula which captures each GNN?
>
> Fortunately, in our proof of Theorem 3, we have already shown that:
>
> (2) For arbitrary $\chi\in\mathcal{X}$ and positive integer $N$, there exists $\varphi\in\Phi$ such that for arbitrary $G,H$ **with no more than $N$ nodes** and $\mathbf{u}\in\mathcal{V}_G,\mathbf{v}\in\mathcal{V}_H$: $\chi(\mathbf{u})\neq\chi(\mathbf{v})\iff\varphi(\mathbf{u})\neq\varphi(\mathbf{v})$.
>
> To better reflect this we have also adjusted the proof of Theorem 3 in the paper to make it more clear. In fact, we also have the following result:
>
> (3) For arbitrary $\chi\in\mathcal{X}$ and positive integer $N$, there exists $\varphi\in\Phi$ such that for arbitrary $G$ **with no more than $N$ nodes** and $\mathbf{u}\in\mathcal{V}_G$: $\chi(\mathbf{u})=\varphi(\mathbf{u})$ (recall that $\chi$ do binary classification).
>
> Given that statement (1) does not hold, we believe statement (2) and (3) appropriately describes the one-to-one correspondence between GNNs and logic. We have modified Section 3.1 the definition of equivalent logic sets to add this statement.
>
> **Example.** We now illustrate how statement (3) holds. For the formal proof, please refer to Appendix E.1 in the revised paper.
>
> Let us first also consider the example: $$\chi(x)=AGG(\{y|y\in N(x)\})=1(|N(x)|\text{ is even}).$$
>
> The reason why we cannot find $\varphi$ such that $\varphi(x)=\chi(x)$ for arbitrary graphs and nodes $x$ is that we may end up with a logic formula with infinite length:
> $$\varphi(x):=\exists^{=2}y(E(x,y))\lor\exists^{=4}y(E(x,y))\lor...$$
> For arbitrary $\varphi$ in the above form, suppose the maximum superscript $n$ of $\exists^{=n}$ is $N$. Then for a node with $N+2$ neighbors, $\varphi$ cannot classify it to $true$.
>
> Instead, if we restrict the graph to have at most $N$ (an even number) nodes, then
> $$\varphi(x):=\exists^{=2}y(E(x,y))\lor...\lor\exists^{=N}y(E(x,y))$$
> always expresses $\chi$.
>
> **Proof sketch.** The statement (3) is already implicitly shown in the initial version of our paper. We have modified the proof section to make it more clear. In short, consider the following situation.
>
> Suppose $\chi$ maps a node tuple to discrete colors (similar to WL tests). Then a set of logic formulas $\Phi$ $N$-expresses $\chi$, if for each possible color $\mathbf{col}$ of $\chi$, there is a $\varphi^\mathbf{col}\in\Phi$ such that for arbitrary graphs $G$ with no more than $N$ nodes and $\mathbf{u}\in\mathcal{V}_G^k$, $\varphi^\mathbf{col}(\mathbf{u})=true$ iff $\chi(\mathbf{u})=\mathbf{col}$. The key is to show that:
> 1. For $\chi(\mathbf{u})=COM(\chi_1(\mathbf{u}),\chi_2(\mathbf{u}))$, if there exists $\Phi_1$ that $N$-expresses $\chi_1$ and $\Phi_2$ that $N$-expresses $\chi_2$, there is also $\Phi$ that $N$-expresses $\chi$.
> 2. For $\chi(\mathbf{u})=AGG(\{\{\chi_1(\mathbf{v})|\mathbf{v}\in\mathcal{N}(\mathbf{u})\}\})$, if there exists $\Phi_1$ that $N$-expresses $\chi_1$, there is also $\Phi$ that $N$-expresses $\chi$.
>
> We do this by explicitly constructing a logic formula for each color of $\chi$ and show that the logic formula is also in $\Phi$, thus completes the proof. For the details about how we construct such logic formula, please refer to Appendix E.12, E.13.
>
> # Other problems.
>
> ```
> It's also a little concerning that this (crucially important) subtlety slipped past in the initial writing of the paper. As such, I am concerned that there may be some issues in the theory, that I may not end up catching. However, I will do my best to engage.
> ```
>
> We sincerely thank the review's effort to engage in the discussion of the paper. We would like to point that besides the latest change (where we added a statement to the definition of equivalent logic sets as discussed above), there was no change in our theory. Also, in the latest revision of our paper, the newly added statement in Definition 2, Section 3.1 has already been proved in the initial writting of the paper (which serves as an intermediate result for proving other results). Moreover, we have rewritten the proof of Theorem 3 to make the proof more clear.
>
> Overall, we thank the reviewer for the constructive comments. Please feel let us know if there are concerns / confusion unsettled.

---

### Official Review · Reviewer_oDss · 2024-11-01

**Soundness:** 4
**Presentation:** 2
**Contribution:** 4
**Rating:** 8
**Confidence:** 2

**Summary:**

The authors introduce a framework to assess the logical expressivity of Graph Neural Network (GNN) models by analyzing their underlying combination and aggregation operations. By constructing sets of logical formulas that each GNN model can capture, this method provides insights into the types of logical properties these models can represent and distinguish. Intuitively, a GNN captures a formula $\phi$ in FO if, for any graph G, the results of $\phi$ applied on G are fully reproduced by the representations computed by the GNN.
By mapping GNN operations to logical formulas, the framework identifies the set of logical statements a GNN can inherently represent. This is a more fine-grained approach to understanding the expressivity of GNNs compared to the WL hierarchy. The paper explores GNN expressivity across different prediction tasks—graph-level, node-level, and link-level—highlighting the logical distinctions between models when applied to tasks at each of these levels.

**Strengths:**

- A new theoretical framework for assessing the expressive power of GNNs by relating it to fragments of FO
- Classification of existing MPNNs, subgraph GNNs
- Ability to assess the expressive power wrt to the ability to capture certain substructures (motifs) in graphs
- Affirms conjecture from prior work

**Weaknesses:**

- The presentation could be improved in several places. For instance, there could be more illustrations and examples throughout the paper.
- Nitpick: I would refrain from assigning very positive attributes, "simple," "elegant," etc., to your own results and proofs.

**Questions:**

- There should be a limitations section. For instance, do these results apply to graphs with node attributes as well? What other limitations has the proposed framework for assessing expressiveness of GNNs? What could be future work?

---

> ### Author Response · Authors · 2024-11-19
> **Response**
>
> We sincerely thank the reviewer for the detailed and constructive comments!
>
> # Weakness
>
> ```
> The presentation could be improved in several places. For instance, there could be more illustrations and examples throughout the paper.
> ```
>
> We have revised the paper to improve the presentation, as well as providing more examples and proof sketches. Please refer to the revised paper, where major modifications are colored blue.
>
> ```
> Nitpick: I would refrain from assigning very positive attributes, "simple," "elegant," etc., to your own results and proofs.
> ```
>
> We have refrained the usage of "elegant", "simple", etc. in the revised paper.
>
>
> # Questions
> There should be a limitations section. For instance, do these results apply to graphs with node attributes as well? What other limitations has the proposed framework for assessing expressiveness of GNNs? What could be future work?
>
> The results of this paper are applicable to GNNs that can be expressed by GACNNs (which is built using aggregation and combination operations). This includes most popular GNNs that learn graph-level, node-level or edge-level representations for both directed and undirected graphs with node and edge features. However, our framework are not applicable for GNNs which do not consist sole of aggregation and combination operations. For example, Graphormer-GD [1] which injects distance information into node pairs. It is possible to find the upper bound of Graphormer-GD, i.e. to find a logic set $\Phi_{upper}$ that is more expressive than Graphormer-GD, as well as finding the lower bound of Graphormer-GD, i.e. finding a logic set $\Phi_{lower}$ that can be captured by Graphormer-GD. However, we do not know whether Graphormer-GD has an equivalent logic set $\Phi$, i.e.  $\Phi$ is equivalently expressive with Graphormer-GD. Although we highly suspect the answer is negative, it is beyond the scope of this paper. Therefore, a possible future direction is to discuss when a GNN has its equivalent logic set $\Phi$ for more general cases, and whether it is possible to find $\Phi$ for it.
>
> We have added a paragraph discussing the limitations and furture work in Section 7 in the revised paper.
>
> [1] Zhang et al. Rethinking the Expressive Power of GNNs via Graph Biconnectivity. ICLR 2023.

---

> > ### Comment · Reviewer_oDss · 2024-11-25
> >
> > I have read the responses of the authors and will keep my score. However, I did not check all the proofs in detail, hence my low confidence. Besides my uncertainty regarding the soundness of the proofs, the paper would be a nice contribution to the theory of GNN literature and the conference.

---

> > > ### Author Response · Authors · 2024-12-03
> > > **Thank you**
> > >
> > > Thank you for your positive feedback and great suggestion. We will try our best to polish the paper to make it more concise and readable.

---

### Official Review · Reviewer_ar9b · 2024-11-02

**Soundness:** 3
**Presentation:** 2
**Contribution:** 2
**Rating:** 8
**Confidence:** 3

**Summary:**

The paper presents a novel framework aimed at systematically assessing the logical expressiveness of GNNs. It proposes a comprehensive relationship between GNNs and logic, offering a method to construct logical formulas equivalent to arbitrary GNN architectures. The paper also discusses the implications of this framework for understanding existing models, comparing expressiveness across models, and estimating WL test bounds.

**Strengths:**

**Solid Theory**: The paper constructs a solid theoretical foundation through extensive mathematical proofs and derivations.

**Unified Framework**: By establishing a connection between GNNs and first-order logic, the paper proposes a theoretical framework that can unify different GNN models. This framework allows for a systematic comparison of the expressiveness of different GNN models.

**Novel Perspective**: The paper analyzes and explains the expressiveness of GNN models from the perspective of first-order logic, providing a new perspective for understanding these models.

**Weaknesses:**

**Limited Comparison and Analysis**: The paper does not specifically analyze what each operation in first-order logic implies about what the model has learned, nor does it directly compare with other existing expressiveness assessment frameworks, such as the work mentioned by Zhang et al.

**Practical Guidance for Model Development**: Although the paper provides an assessment framework, it lacks specific guidance on how to use this framework to guide the development of high-expressiveness GNN models.

**Insufficient Experimental Validation (Not necessarily a con)**: The paper lacks an experimental section to validate the effectiveness, rationality, and correctness of its framework.

**Questions:**

1. Early work also explained the expressiveness of GNNs from a logical perspective. What new contributions have your contributions made compared to their work? Specifically, what are the differences between the second-order logic they used and the first-order logic used in your work? What are the respective advantages and disadvantages?

2. Please further elaborate and supplement what each operation in first-order logic specifically corresponds to in terms of what the model learns from the graph?

3. Zhang et al. proposed a refined framework for assessing the expressiveness of GNNs in "Beyond Weisfeiler-Lehman: A Quantitative Framework for GNN Expressiveness." Compared to the assessment method proposed in the paper, what are the respective strengths and weaknesses? Could you provide a detailed comparative analysis?

4. The paper compares several typical GNN models using a unified framework. Could you further explain how the differences in logical expressions of different GNN models correspond to the actual capabilities of the models in practice?

5. Please supplement experiments to prove the completeness, rationality, and correctness of the theoretical aspects of the paper. This would be very helpful for understanding the differences between models and selecting models suitable for specific tasks.

---

> ### Author Response · Authors · 2024-11-22
> **Response (Part 1)**
>
> We sincerely thank the reviewer for the detailed and constructive comments. We are sorry for the late response, because we had to design and perform numerical experiments from scratch. We will address the questions below.
>
> # Weakness
>
> ```
> Limited Comparison and Analysis: The paper does not specifically analyze what each operation in first-order logic implies about what the model has learned, nor does it directly compare with other existing expressiveness assessment frameworks, such as the work mentioned by Zhang et al.
> ```
>
> Thanks for pointing it out. We tackle the weakness in the discussion of Question 2 below. To summarize, We have added a discussion in Section 5.1 to analyze what the operations $\exists^N,\neg,\land$ implies for the models. The text is colored blue, at the bottom of Page 6. We have already compared our theory with existing expressiveness assessment frameworks in paragraph "Expressivity of GNNs", Appendix A. Moreover, we have revised the paragraph to highlight the difference between our framework and existing ones. The modified parts are colored blue.
>
> ```
> Practical Guidance for Model Development: Although the paper provides an assessment framework, it lacks specific guidance on how to use this framework to guide the development of high-expressiveness GNN models.
> ```
>
> We tackle the weakness in the discussion of Question 4 below.
>
> ```
> Insufficient Experimental Validation (Not necessarily a con): The paper lacks an experimental section to validate the effectiveness, rationality, and correctness of its framework
> ```
>
> We are trying our best to design and perform numerical experiments to validate our theory. We have already finished major parts of the experiment and will add an experiment section in the paper soon.

---

> ### Author Response · Authors · 2024-11-22
> **Response (Part 2)**
>
> # Questions
>
> ```
> 1. Early work also explained the expressiveness of GNNs from a logical perspective. What new contributions have your contributions made compared to their work? Specifically, what are the differences between the second-order logic they used and the first-order logic used in your work? What are the respective advantages and disadvantages?
> ```
>
> We are sorry we don't fully understand "the second-order logic they used". To the best of our knowledge, an early work (Barcelo et al [1]) also uses first-order logic to model GNNs. Specially, they discover a subset of first-order logic, namely graded modal logic, which corresponds to MPNNs.
>
> We now compare our work with existing ones. The major advantages of our work includes: \
> (1) Our study establishes a deeper connection between GNNs and logic. Briefly speaking, previous works such as [1] only study a set $\Phi$ of logic formulas for MPNNs such that:
> - All logic formulas $\varphi\in\Phi$ can be captured by MPNNs.
> - A logic formula can be captured by a MPNN iff it is in $\Phi$.
>
> It is worth noting that the above statements only guarantees that the logic formulas can be captured by MPNNs. Intuitively, this does not establish an equivalence between logic and GNNs. Instead, in our work, we study a set $\Phi$ of logic formulas for GNNs such that:
> - All logic formulas $\varphi\in\Phi$ can be captured by GNNs.
> - The GNNs are equivalently expressive with $\Phi$ in distinguishing non-isomorphic graphs.
>
> Therefore, our theory reveals deeper connection between GNNs and logic (i.e. the equivalence between GNNs and logic.) For details, please refer to Definition 2, Section 3.1.
>
> (2) Our study specifies the logical expressiveness of *arbitrary* GNNs, provided that their layers are built using aggregation and combination functions. This differs with existing works, as existing works only investigate specific models (MPNNs), and to the best of our knowledge there's no existsing work that provide a general framework for assessing the logical expressiveness of arbitrary aggregation-combination GNNs.
>
> (3) In this paper, we establish a connection between logic and the homomorphism expressivity of GNNs, which is novel and to the best of our knowledge has not been discussed by previous works. (See answer of Question 3 below.)
>
> ```
> 2. Please further elaborate and supplement what each operation in first-order logic specifically corresponds to in terms of what the model learns from the graph?
> ```
> We have discussed in L318. The logic formulas are described using operators $\exists^{\geq N},\neg$ and $\land$, which follows the settings in previous works [1]. Each operator is corresponded to a specific property of GNNs. First, note that the major characteristic of GNNs as neural networks is that GNNs make predictions based on not only the current instance (e.g. a node in graphs), but also its relation with others (which is expressed by edges in graphs). This is realized via the aggregation of information among nodes. The logic operation $\exists^{\geq N}$, which expresses ``exists at least $N$'', exactly corresponds to this property: for example, $\mathrm{Blue}(x):=\exists^{\geq 2}y(\mathrm{Red}(y)\land E(x,y))$ directly explores the neighbors of the current node $x$ and decides whether it is $\mathrm{Blue}$ by aggregating the colors of its neighbors. Thus we can understand the aggregation procedures in GNNs via the logical operation $\exists^{\geq N}$. Therefore, the operator $\exists^{\geq N}$ corresponds to the model learning how to integrate the information of neighbors into the current node.
>
> Second, the operators $\neg$ and $\land$ are used to recombine existing logic formulas to obtain more complex ones. This implies the ability of GNNs to recombine existing information and make complex predictions. Therefore, these operators $\neg,\land$ correspond to the model learning how to process the existing information of current node and make prediction based on it.

---

> ### Author Response · Authors · 2024-11-22
> **Response (Part 3)**
>
> ```
> 3. Zhang et al. proposed a refined framework for assessing the expressiveness of GNNs in "Beyond Weisfeiler-Lehman: A Quantitative Framework for GNN Expressiveness." Compared to the assessment method proposed in the paper, what are the respective strengths and weaknesses? Could you provide a detailed comparative analysis?
> ```
>
> Compared with their method, our method:
>
> (1) Proposes a method for constructing the homomorphism expressivity for *arbitrary* GNNs, provided that the layers are build using aggregation and combination functions. Zhang et al. identified the homomorphism expressivity of several popular GNNs, but did not propose a method for identifying the homomorphism expressivity of arbitrary aggregation-combination GNNs.
>
> (2) Our work also discovers the connection between GNNs and logic (See answer of Question 1 above).
>
> ```
> The paper compares several typical GNN models using a unified framework. Could you further explain how the differences in logical expressions of different GNN models correspond to the actual capabilities of the models in practice?
> ```
> We have discussed the differences of GNN models in Section 6.1. Furthermore, we can illustrate the differences with some examples below.
>
> First, we would like to discuss about the **node classification capabilities** using MPNNs and Subgraph GNNs (weak) as examples. Recall that the logical expressiveness of MPNNs is:
> $$\varphi(x):=\exists^{\geq N}x(\varphi'(y)\land E(x,y)),$$
> and for Subgraph GNNs:
> $$\varphi(x):=\exists^{\geq N}y(\psi(x,y)).$$
>
> Consider the following two graphs, which are cycles of 3 and 4 nodes. It is evident that MPNNs cannot distinguish the nodes in (a) and the nodes in (b): MPNNs only use edges $E(x,y)$ to model the relation of nodes, and for each node (denoted by $x$) in both (a) and (b), there are exactly two nodes (denoted by $y$) that satisfy $E(x,y)$. \
> Subgraph GNNs (weak) utilize logic formulas $\psi(x,y)$ to model the relation between nodes. If we let $\psi(x,y):=\exists z(E(x,z)\land E(z,y))$ (which can be expressed by subgraph GNNs(weak)), each node $x$ in (a) has two nodes $y$ that satisfy $E(x,y)$, while for (b) there is only one. In this sence, $\psi$ is more general (as it can model more complex relation $\exists z(E(x,z)\land E(z,y))$ between nodes) and more powerful (distinguish nodes that are indistinguishable by MPNNs). This aligns with the fact that Subgraph GNNs are more powerful that MPNNs and often perform better in complex node classification tasks [2].
>
> o       $~~~~~~~~~~$ o---o\
> |$~~~$\\  $~~~~$ |$~~~~~~$|\
> o---o       $~~~~$ o---o\
> $~~(a)~~~~~~~~(b)$
>
> Next, we would like to discuss about the **link prediction capabilities** using MPNNs and NBFNet as examples. Consider to predict whether a person is another person's grandparent in a kinship graph. The procedure can be described by a simple logic formula
> $$grandparent(x,y):=\exists z(parent(x,z)\land parent(z,y)).$$
> Unfortunately, MPNNs cannot express the logic formula. Such a pattern of logic formulas is crucial for link prediction tasks. For example, deciding whether two nodes are connected:
> $$connect(x,y):=\exists z(connect(x,z)\land E(z,y)).$$
> Different from MPNNs, NBFNet is able to capture the above logic formulas. This aligns with the fact that MPNNs are relative weak on link prediction tasks and models like NBFNet often achieve much better performance in link prediction and knowledge graph completion tasks. [3]
>
> ```
> Please supplement experiments to prove the completeness, rationality, and correctness of the theoretical aspects of the paper. This would be very helpful for understanding the differences between models and selecting models suitable for specific tasks.
> ```
>
> We are trying our best to design and perform experiments, and will update the experiment results before discussion period ends. We select several representative logic formulas for accessing the node classification, link prediction and graph classification capabilities of models. We validate our theory by applying differernt GNN models on both manually designed graphs and randomly generated ones.
>
>
> [1] Barcelo et al. THE LOGICAL EXPRESSIVENESS OF GRAPH NEURAL NETWORKS. ICLR 2020.
>
> [2] Zhang et al. Beyond Weisfeiler-Lehman: A Quantitative Framework for GNN Expressiveness. ICLR 2024.
>
> [3] Zhu et al. Neural Bellman-Ford Networks: A General Graph Neural Network Framework for Link Prediction. NeurIPS 2021.

---

> > ### Comment · Reviewer_ar9b · 2024-11-26
> >
> > Upon carefully review of your thoughtful responses and subsequent revisions to the manuscript, I am pleased to see that my feedback has been considered and integrated into your work. Your revisions could address the majority of the concerns raised in the initial review.
> >
> > You have clearly explained the correlation between first-order logic operations and GNN behaviors. To further enhance the reader's comprehension of the practical utility of these logic expressions, I would encourage you to incorporate additional illustrative examples. These examples should ideally demonstrate the interplay between logic operation behaviors, model behaviors, and real-world applications, thereby providing a more tangible connection to the theoretical constructs.
> >
> > The examples you have presented, which elucidate the link between logical expressions and model capabilities, particularly in the link prediction, are good. Nevertheless, there is room for further expansion upon this topic. Prior researches on logical expressivity have not been extensively connected to real-world scenarios, and the focus on link prediction tasks within the GNN expressivity has been relatively sparse. A more detailed exploration of these aspects could significantly enrich the paper's contribution to the field.
> >
> > In addition, I note your commitment to supplementing your theory with experiments, which is very important. I look forward to seeing the results of these experiments to further validate the effectiveness of your approach.
> >
> > Overall, I found your response to be adequate and that your revisions could improve the quality of the paper. I look forward to seeing the results of your experiments included in the final version.

---

> > > ### Author Response · Authors · 2024-11-27
> > > **Response**
> > >
> > > We sincerely thank the reviewer again for the constructive comments.
> > >
> > > ```
> > > I would encourage you to incorporate additional illustrative examples. These examples should ideally demonstrate the interplay between logic operation behaviors, model behaviors, and real-world applications, thereby providing a more tangible connection to the theoretical constructs.
> > > ```
> > >
> > > We have added discussions and examples in L338. Please refer to the revised paper to see this.
> > >
> > > ```
> > > Prior researches on logical expressivity have not been extensively connected to real-world scenarios, and the focus on link prediction tasks within the GNN expressivity has been relatively sparse. A more detailed exploration of these aspects could significantly enrich the paper's contribution to the field.
> > > ```
> > >
> > > Thanks for the advice. We have added a section dedicated to discussing link prediction and logic. Please refer to Appendix F in the revised paper.
> > >
> > > ### About experiments.
> > >
> > > We have supplemented experiments in Appendix D. Please refer to the revised paper to check the results.
> > >
> > > Overall, we thank the review for the constructive comments, and we hope that our response have addressed your concerns

---

> > > > ### Comment · Reviewer_ar9b · 2024-11-29
> > > >
> > > > After checking the revision, I believe it is a good work at its current form.
> > > > I have adjusted my score accordingly. Good luck!

---

> > > > > ### Author Response · Authors · 2024-12-03
> > > > > **Thank you**
> > > > >
> > > > > Thank you for your positive feedback and great suggestion. We will try our best to polish the paper to make it more concise and readable, as well as sticking to the 10-page limitation of the conference.

---

### Author Response · Authors · 2024-11-29
**General Response: New results & Updates**

We sincerely thank all the reviewers for their great efforts in reviewing our paper. Below we would like to summarize major updates during discussion period and highlight that we have enhanced two key theoretical results originally presented in the initial paper, which we believe can further strengthen this work.

## The Definition of Equivalent Logic Set

In the inital submission, we define the equivalent logic set $\Phi$ of a family of graph function $\mathcal{X}$ (e.g. a class of GNNs) is:
- The order of each $\varphi\in\Phi$ is $k$;
- For all $\varphi\in\Phi$, there exists $\chi\in\mathcal{X}$ such that for arbitrary graphs $G$ and $\mathbf{u}\in\mathcal{V}^k_G$,  $\varphi(\mathbf{u})=true$ iff $\chi(\mathbf{u})=true$, and we say $\chi$ captures $\varphi$.
- Given any graphs $G,H$ and $\mathbf{u}\in\mathcal{V}_G^k,\mathbf{v}\in\mathcal{V}_H^k$, all $\chi\in\mathcal{X}$ cannot distinguish $\mathbf{u},\mathbf{v}$ iff all logic formulas $\varphi\in\Phi$ classify $\mathbf{u},\mathbf{v}$ the same.

During the discussion period, we found this definition limited in that:
1. The definition does not state that whether there exists a one-to-one correspondence between one GNN model $\chi\in\mathcal{X}$ and one logic formula $\varphi\in\Phi$. It only states that all GNNs are equivalent to all logic formulas.
2. The definition does not state the completeness of $\Phi$, i.e. whether all logic formulas captured by $\mathcal{X}$ are in $\Phi$. This property has been discussed by Barcelo et al. [1] for MPNNs.

We have now successfully addressed the above limits and extended our results (see Definition 2 in the paper). We believe that the equivalent logic sets can now more completely characterize the logical expressivity of different models. We added two statements in the defintion of equivalent logic sets:
### About one-to-one correspondence between $\chi\in\mathcal{X}$ and $\varphi\in\Phi$.
We have shown that there exists a GNN model which is not captured by any logic formula. Therefore, to disscuss this property, we defined a weaker version of one-to-one correspondence:
- Given arbitrary positive integer $N$ and $\chi\in\mathcal{X}$, there exists $\varphi\in\Phi$ satisfying: for any graphs $G$ with no more than $N$ nodes and $\mathbf{u}\in\mathcal{V}_G^k$, $\varphi(\mathbf{u})=true$ iff $\chi(\mathbf{u})=true$.

This is weaker than saying $\varphi$ is always captured by $\chi$, as it only considers finite situations. However, using this defintion we are able to give a one-to-one correspondence between GNNs and logic formulas. This property is already proved in the initial paper.
### About the completeness of $\Phi$.
We have now successfully proved the following results for equivalent logic sets:
- A FOC formula is captured by $\mathcal{X}$ iff it is in $\Phi$.

The proof is a direct generalization of the proof of Theorem 2.1 in [2], in which we generalized the discussion of first-order logic and $\sim_\sharp $-invariance (which is invariance under 1-WL where node colors are refined by neighbors) into the discussion of first-order logic and GACNNs(where node tuple colors are refined by generalized neighbors). One can see that the above results establish stronger relationship between GNNs and logic.

## Experiments

We have conducted experiments to empirically test the logical expressivity of GNNs validate our theory, which is presented in Appendix D.

## About writting / clarity of the paper

We have revised the main body of the paper to improve the writting and the clarity of the presentation. This includes:

- A more precise definition of the symbols / terms used in this paper (Section 2).
- A more precise definition of GACNNs (Section 3, 4).
- Examples about the structure of GACNNs (Section 4), the relation between logic operators and GNNs (Section 5.1).
- A more concise description of the construction of homomorphism expressivity (Section 6).
- Proof sketches for main theorems in the main text.
- Extended related works (Appendix A).
- Discussion of link prediction and logic (Appendix F).
- Rewrite of proofs.
- Other minor changes, including fixing typos, etc.






[1] Barcelo et al. THE LOGICAL EXPRESSIVENESS OF GRAPH NEURAL NETWORKS. ICLR 2020. \
[2] Martin Otto, Graded modal logic and counting bisimulation, 2023.

---

### Meta-Review · Area_Chair_eNT5 · 2024-12-21

**Metareview:**

This paper provides a novel, extensive lens on the logical expressiveness of GNN models. Its strengths are recognised unanimously by all three reviewers, who all recommend strong acceptance. A talk feature at ICLR is recommended!

**Additional Comments On Reviewer Discussion:**

I was especially impressed by how well the Authors handled the rebuttals, and would like to thank Reviewer pXBi for the very long chain of discussions with them to reach a unanimous acceptance decision. This is the review system exactly working-as-intended to produce a stellar paper out of an initially borderline contribution.

---

### Decision · Program_Chairs · 2025-01-22

Accept (Oral)